# Edge Representation Learning with Hypergraphs

**Jaehyeong Jo**[1*], **Jinheon Baek**[1*], **Seul Lee**[1*],
**Dongki Kim**[1], **Minki Kang**[1], **Sung Ju Hwang**[1,2]
KAIST[1], AITRICS[2], South Korea
harryjo97@kaist.ac.kr, jinheon.baek@kaist.ac.kr,
ellenlee7890@gmail.com, cleverki@kaist.ac.kr,
zzxc1133@kaist.ac.kr, sjhwang82@kaist.ac.kr

## Abstract

Graph neural networks have recently achieved remarkable success in representing
graph-structured data, with rapid progress in both the node embedding and graph
pooling methods. Yet, they mostly focus on capturing information from the nodes
considering their connectivity, and not much work has been done in representing
the *edges*, which are essential components of a graph. However, for tasks such
as graph reconstruction and generation, as well as graph classification tasks for
which the edges are important for discrimination, accurately representing edges
of a given graph is crucial to the success of the graph representation learning.
To this end, we propose a novel edge representation learning framework based
on *Dual Hypergraph Transformation* (DHT), which transforms the edges of a
graph into the nodes of a *hypergraph*. This dual hypergraph construction allows
us to apply message-passing techniques for node representations to edges. After
obtaining edge representations from the hypergraphs, we then cluster or drop
edges to obtain holistic graph-level edge representations. We validate our edge
representation learning method with hypergraphs on diverse graph datasets for
graph representation and generation performance, on which our method largely
outperforms existing graph representation learning methods. Moreover, our edge
representation learning and pooling method also largely outperforms state-of-the-
art graph pooling methods on graph classification, not only because of its accurate
edge representation learning, but also due to its lossless compression of the nodes
and removal of irrelevant edges for effective message-passing.[1]

## 1 Introduction

The recent demand in representing graph-structured data, such as molecular, social, and knowledge
graphs, has brought remarkable progress in the *Graph Neural Networks* (GNNs) [44, 37]. Early
works on GNNs [23, 15, 39] aim to accurately represent each node to reflect the graph topology, by
transforming, propagating, and aggregating information from their neighborhoods based on message-
passing schemes [12]. More recent works focus on learning holistic graph-level representations,
by proposing graph pooling techniques that condense the node-level representations into a smaller
graph or a single vector. While such state-of-the-art node embedding or graph pooling methods
have achieved impressive performances on graph-related tasks (e.g., node classification and graph
classification), they have largely overlooked the *edges*, which are essential components of a graph.

Most existing GNNs, including ones that consider categorical edge features [32, 12], only implicitly
capture the edge information in the learned node/graph representations when updating them. While a

---

[*]Equal contribution

[1]Code is available at https://github.com/harryjo97/EHGNN

35th Conference on Neural Information Processing Systems (NeurIPS 2021).

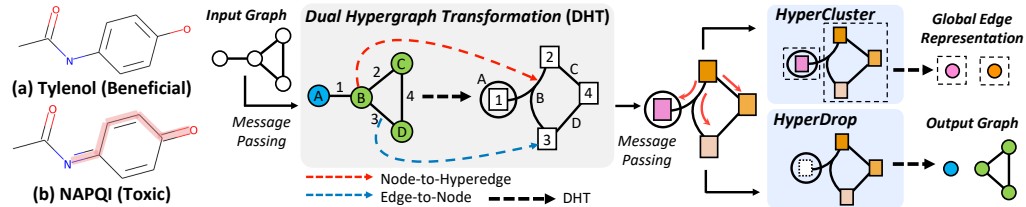

Figure 1: **(Left):** The two molecular graphs[2] have the identical set of nodes, but possess completely different properties due to the difference in edges. **(Right):** An illustration of the proposed edge representation learning framework with two novel edge pooling schemes. The grey box in the center describes the proposed **Dual Hypergraph Transformation**, where the numbers (letters) denote the corresponding edges (nodes) in the graph and nodes (hyperedges) in the hypergraph. The two blue boxes in the right illustrate the proposed edge pooling methods, **HyperCluster** which clusters similar edges, and **HyperDrop** which drops unnecessary edges.

few of them aim to obtain explicit representations for edges [20, 13, 40], they mostly use them only to augment the node-level representations, and thus suboptimally capture the edge information. This is partly because many benchmark tasks for GNN performance evaluation, such as graph classification, do not require the edge information to be accurately preserved. Thus, on this classification task, simple MLPs without any connectivity information can sometimes outperform GNNs [9, 18]. However, for tasks such as graph reconstruction and generation, accurately representing the edges of a graph is crucial to success, as incorrectly reconstructing/generating edges may result in complete failure of the tasks. For example, the two molecules (a) and (b) in Figure 1 have exactly the same set of nodes and are only different in their edges (bond types), but exhibit extremely different properties.

To overcome such limitations of existing GNN methods in edge representation learning, we propose a simple yet effective scheme to represent the edges. The main challenge of handling edges is the absence or suboptimality of the message-passing scheme for edges. We tackle this challenge by representing the edges as nodes in a *hypergraph*, which is a generalization of a graph that can model higher-order interactions among nodes as one hyperedge can connect an arbitrary number of nodes. Specifically, we propose *Dual Hypergraph Transformation* (DHT) to transform edges of the original graph to nodes of a *hypergraph* (Figure 1), and nodes to hyperedges. This hypergraph-based approach is effective since it allows us to apply any off-the-shelf message-passing schemes designed for node-level representation learning, for learning the representation of the edges of a graph.

However, representing each edge well alone is insufficient in obtaining an accurate representation of the entire graph. Thus we propose two novel graph pooling methods for the hypergraph to obtain compact graph-level edge representations, namely *HyperCluster* and *HyperDrop*. Specifically, for obtaining global edge representations for an entire graph, HyperCluster coarsens similar edges into a single edge under the global graph pooling scheme (see HyperCluster in Figure 1). On the other hand, HyperDrop drops unnecessary edges from the original graph by calculating pruning scores on the hypergraph (see HyperDrop in Figure 1). HyperCluster is more useful for graph reconstruction and generation as it does not result in the removal of any edges, while HyperDrop is more useful for classification as it learns to remove edges that are less useful for graph discrimination.

We first experimentally validate the effectiveness of the DHT with *HyperCluster*, on the reconstruction of synthetic and molecular graphs. Our method obtains extremely high performance on these tasks, largely outperforming baselines, which shows its effectiveness in accurately representing the edges. Then, we validate our method on molecular graph generation tasks, and show that it largely outperforms base generation methods, as it allows us to generate molecules with more correct bonds (edges). Further, we validate *HyperDrop* on 10 benchmark datasets for graph classification, on which *HyperDrop* outperforms all hierarchical pooling baselines, with larger gains on social graphs, for which the edge features are important. Our main contributions are summarized as follows:

- We introduce a **novel edge representation learning scheme** using *Dual Hypergraph Transformation*, which exploits the dual hypergraph whose nodes are edges of the original graph, on which we can apply off-the-shelf message-passing schemes designed for node-level representation learning.
- We propose **novel edge pooling methods** for graph-level representation learning, namely *HyperCluster* and *HyperDrop*, to overcome the limitations of existing node-based pooling methods.
- We validate our methods on **graph reconstruction, generation, and classification tasks**, on which they largely outperform existing graph representation learning methods.

---

[2]We depict only the heavy atoms, as conventional preprocessing of molecular graphs drops hydrogen atoms.

## 2 Related Work

**Graph neural networks**  Graph neural networks (GNNs) mostly use the message-passing scheme [12] to aggregate features from their neighbors. Particularly, Graph Convolutional Network (GCN) [23] generalizes the convolution operation in the spectral domain of graphs, and updates the representation of each node by applying the shared weights on it and its neighbors' representations. Similarly, GraphSAGE [15] propagates the features of each node's neighbors to itself, based on simple aggregation operations (e.g., mean). Graph Attention Network (GAT) [34] considers the relative importance on neighboring nodes with attention, to update each node's representation as the weighted combination of its neighbors'. Xu et al. [39] show that a simple sum on neighborhood aggregation makes GNNs as powerful as the Weisfeiler-Lehman (WL) test [36], which is effective for distinguishing different graphs. While GNNs have achieved impressive success on graph-related tasks, most of them only focus on learning node-level representations, with less focus on the edges.

**Edge-aware graph neural networks**  Some existing works on GNNs consider edge features while updating the node features [32, 33], however, they only use the edges as auxiliary information and restrict the representation of edges as the discrete features with categorical values. While a few methods [20, 12, 13, 40] explicitly represent edges by introducing edge-level GNN layers, they use the obtained edge features solely for enhancing node features. Also, existing message-passing schemes for nodes are not directly applicable to edge-level layers, as they are differently designed from the node-level layers, which makes it challenging to combine them with graph pooling methods [41] for graph-level representation learning. We overcome these limitations by proposing a dual hypergraph transformation scheme, to obtain a hypergraph whose nodes are edges of the original graph, which allows us to apply any message-passing layers designed for nodes to edges.

**Graph transformation**  Recently, some works [21, 27] propose to transform the original graph into a typical graph structure, to apply graph convolution for learning the edge features. Specifically, they construct a line graph [16], where the nodes of the line graph correspond to the edges of the original graph, and the nodes of the line graph are connected if the corresponding edges of the original graph share the same endpoint. However, the line graph transformation has obvious drawbacks: 1) the transformation is not injective, thus two different graphs may be transformed into the same line graph; 2) the transformation is not scalable; 3) node information in the original graph may be lost during the transformation. Instead of using such a graph structure, we use hypergraphs, which can model higher-order interactions among nodes by grouping multi-node relationships into a single hyperedge [3]. Using the hypergraph duality [31], edges of the original graph are regarded as the nodes of a hypergraph. For example, Lugo-Martinez and Radivojac [26] cast a hyperlink prediction task as an instance of node classification from the dual form of the original hypergraph. On the other hand, Kajino [22] uses the duality to extract useful rules from the hypergraph structures by transforming molecular graphs, for their generation. However, none of the existing works exploit the relation between the original graph and the dual hypergraph for edge representation learning.

**Graph pooling**  Graph pooling methods aim to learn accurate graph-level representation by compressing a graph into a smaller graph or a vector with pooling operations. The simplest pooling approaches are using `mean`, `sum` or `max` over all node representations [1, 39]. However, they treat all nodes equally, and cannot adaptively adjust the size of graphs for downstream tasks. More advanced methods, such as node clustering methods, coarsen the graph by clustering similar nodes based on their embeddings [41, 4], whereas the node pruning methods reduce the number of nodes from the graph by dropping unimportant nodes based on their scores [11, 24]. Ranjan et al. [29] combine both node pruning and clustering approaches, by dropping meaningless clusters after grouping nodes. Baek et al. [2] propose to use attention-based operations for considering relationships between clusters. Note that all of those pooling schemes not only ignore edge representations, but also alter the node set by dropping, clustering, or merging nodes, which result in an inevitable loss of node information.

## 3 Edge Representation Learning with Hypergraphs

In this section, we first introduce our novel edge representation learning framework with dual hypergraphs, which we refer to as *Edge HyperGraph Neural Network* (EHGNN), and then propose two novel edge pooling schemes for holistic graph-level representation learning: *HyperCluster* and *HyperDrop*. We begin with the descriptions of graph neural networks for node representation learning.

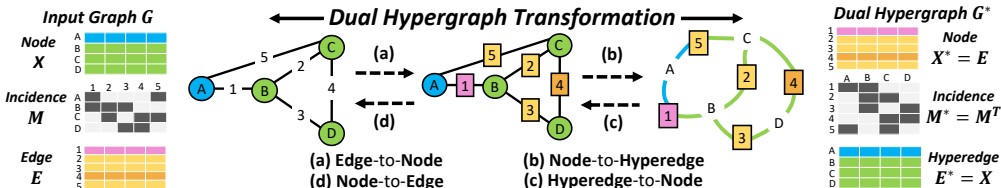

Figure 2: **Dual Hypergraph Transformation.** Illustration of the proposed graph-to-hypergraph transformation.

**Graph neural networks**  A graph $G$ with $n$ nodes and $m$ edges, is defined by its node features $\boldsymbol{X} \in \mathbb{R}^{n \times d}$, edge features $\boldsymbol{E} \in \mathbb{R}^{m \times d'}$, and the connectivity among the nodes represented by an adjacency matrix $\boldsymbol{A} \in \mathbb{R}^{n \times n}$. Here, $d$ and $d'$ are the dimensions of node and edge features, respectively. Then, given a graph, the goal of a *Graph Neural Network* (GNN) is to learn the node-level representation with message-passing between neighboring nodes [12] as follows:

$$\boldsymbol{X}_v^{(l+1)} = \text{UPDATE} \left( \boldsymbol{X}_v^{(l)}, \text{AGGREGATE} \left( \left\{ \boldsymbol{X}_u^{(l)} : \forall u \in \mathcal{N}(v; \boldsymbol{A}) \right\} \right) \right), \quad (1)$$

where $\boldsymbol{X}^{(l)}$ is the node features at $l$-th layer, AGGREGATE is the function that aggregates messages from a set of neighboring nodes of the node $v$, UPDATE is the function that updates the representation of the node $v$ from the aggregated messages, and $\mathcal{N}(v; \boldsymbol{A})$ is the set of neighboring nodes for the node $v$, obtained from the adjacency matrix $\boldsymbol{A}$. Such message-passing schemes can incorporate the graph topology into each node by updating its representation with the representation of its neighbors.

### 3.1 Edge representation learning with dual hypergraph transformation

**Edge representation learning**  To reflect the edge information on message-passing, some works on GNNs first obtain the categorical edge features between nodes, and then use them on the AGGRE-GATE function in equation 1, by adding or multiplying the edge features to the neighboring node's features [12, 32] (see Section A.1 of the supplementary file for more details). Similarly, few recent works aim to obtain explicit edge representations, but only to use them as auxiliary information to augment the node features, by adding or multiplying edge features to them [13, 40]. Thus, existing works only implicitly capture the edge information in the learned node representations. Although this might be sufficient for most benchmark graph classification tasks, many real-world tasks of graphs (e.g., graph reconstruction and generation) further require the edges to be accurately represented as the information on edges could largely affect the task performance.

Even worse, to define a message-passing function for edge representation learning, existing works propose to additionally create the adjacency matrix for edges, either by defining the edge neighborhood structure [40] or using the line graph transformation [20]. However, these are highly suboptimal as obtaining the adjacency of edges requires $\mathcal{O}(n^2)$ time complexity (see Section A.3 of the supplementary file for detailed descriptions), as shown in Table 1. This is

Table 1: Transformation and message-passing complexities of edge-aware GNNs, line graph, and our EHGNN for the star graph, in which one hub node is connected to $n$ other nodes.

| Models | Complexity | |
|---|---|---|
| | Transformation | Message-passing |
| Edge-aware GNNs | $\mathcal{O}(n^2)$ | $\mathcal{O}(n)$ |
| Line graph | $\mathcal{O}(n^2)$ | $\mathcal{O}(n^2)$ |
| EHGNN (Ours) | $\mathcal{O}(n)$ | $\mathcal{O}(n)$ |

the main obstacle for directly applying existing message-passing schemes for nodes to edges. To this end, we propose a simple yet effective method to represent the edges of a graph, using a hypergraph.

**Hypergraph**  A *hypergraph* is a generalization of a graph that can model graph-structured data with higher-order interactions among nodes, wherein a single hyperedge connects an arbitrary number of nodes, unlike in conventional graphs where an edge can only connect two nodes. For example, in Figure 2, the hyperedge $B$ defines the relation among three different nodes. To denote such higher-order relations among arbitrary number of nodes defined by a hyperedge, we use an *incidence matrix* $\boldsymbol{M} \in \{0,1\}^{n \times m}$, which represents the interaction between $n$ nodes and $m$ hyperedges, instead of using an adjacency matrix $\boldsymbol{A} \in \{0,1\}^{n \times n}$ that only considers interactions among $n$ nodes. Each entry in the incidence matrix indicates whether the node is incident to the hyperedge. We can formally define a hyperagraph $G^*$ with $n$ nodes and $m$ hyperedges, as a triplet of three components $G^* = (\boldsymbol{X}^*, \boldsymbol{M}^*, \boldsymbol{E}^*)$, where $\boldsymbol{X}^* \in \mathbb{R}^{n \times d}$ is the node features, $\boldsymbol{E}^* \in \mathbb{R}^{m \times d'}$ is the hyperedge features, and $\boldsymbol{M}^* \in \{0,1\}^{n \times m}$ is the incidence matrix of the hypergraph. We can also represent conventional graphs in the form of a hypergraph, $G = (\boldsymbol{X}, \boldsymbol{M}, \boldsymbol{E})$, in which a hyperedge in the

incidence matrix $M$ is associated with only two nodes. In the following paragraph, we will describe how to transform the edges of a graph into nodes of a hypergraph, for edge representation learning.

**Dual Hypergraph Transformation**   If we can change the role of the nodes and edges of the graph with a shared connectivity pattern across the nodes and edges, while accurately preserving their information, then we can use any node-based message-passing schemes for learning edges. To achieve this, inspired by the hypergraph duality [3, 31], we propose to transform an edge of the original graph into a node of a hypergraph, and a node of the original graph into a hyperedge of the same hypergraph. We refer to this graph-to-hypergraph transformation as *Dual Hypergraph Transformation* (DHT) (see Figure 2). To be more precise, during the transformation, we interchange the structural role of nodes and edges from the given graph, obtaining the incidence matrix for the new dual hypergraph simply by *transposing* the incidence matrix of the original graph (see the incidence matrix in Figure 2). Along with the structural transformation through the incidence matrix, the DHT naturally interchanges node and edge features across $G$ and $G^*$ (see the feature matrices in Figure 2). Formally, given a triplet representation of a graph, DHT is defined as the following transformation:

$$DHT \; : \; G = \big(X, M, E\big) \; \mapsto \; G^* = \big(E, M^T, X\big), \tag{2}$$

where we refer to the transformed $G^*$ as the *dual hypergraph* of the input graph $G$. Since the dual hypergraph $G^* = (E, M^T, X)$ retains all the information of the original graph, we can recover the original graph from the dual hypergraph with the same DHT operation as follows:

$$DHT \; : \; G^* = \big(E, M^T, X\big) \; \mapsto \; G = \big(X, M, E\big). \tag{3}$$

This implies that DHT is a *bijective transformation*. DHT is simple to implement, does not incur the loss of any features or topological information of the input graph, and does not require additional memory for feature representations. Moreover, DHT can be sparsely implemented using the edge list, which is the sparse form of the adjacency matrix, by only reshaping the edge list of the original graph into the hyperedge list of the dual hypergraph (see Section A.2 of the supplementary file for details), which is highly efficient in terms of time and memory. Thanks to DHT, we define the message-passing between edges of the original graph as the message-passing between nodes of its dual hypergraph.

**Message-passing on the dual hypergraph for edge representation learning**   After transforming the original graph into its corresponding dual hypergraph using DHT, we can perform the message-passing between edges of the input graph, by performing the message-passing between nodes of its dual hypergraph $G^* = (E, M^T, X)$, which is formally denoted as follows:

$$E_e^{(l+1)} = \text{UPDATE}\left(E_e^{(l)}, \text{AGGREGATE}\left(\left\{E_f^{(l)} : \forall f \in \mathcal{N}(e; M^T)\right\}\right)\right), \tag{4}$$

where $E^{(l)}$ is the node features of $G^*$ at $l$-th layer, the AGGREGATE function summarizes the neighboring messages of the node $e$ of the dual hypergraph $G^*$, and the UPDATE function updates the representation of the node $e$ from the aggregated messages. Here $\mathcal{N}(e; M^T)$ is the neighboring node set of the node $e$ in $G^*$, which we obtain using the incidence matrix $M^T$ of $G^*$. Furthermore, instead of using the dense incidence matrix, we can sparsely implement the message-passing on the dual hypergraph with the hyperedge list, from which the complexity of message-passing on the dual hypergraph reduces to $\mathcal{O}(m)$, which is equal to the complexity of message-passing between nodes on the original graph (See Section A.3 of the supplementary file for details). Note that, since the form of equation 4 is the same as the form of equation 1, we can use any graph neural networks which realize the message-passing operation in equation 1, such as GCN [23], GAT [34], GraphSAGE [15], and GIN [39], for equation 4. In other words, to learn the edge representations $E$ of the original graph, we do not require any specially designed layers, but simply need to perform DHT to directly apply existing off-the-shelf message-passing schemes to the transformed dual hypergraph.

To simplify, we summarize the equation 1 as follows: $X^{(l+1)} = \text{GNN}\big(X^{(l)}, M, E^{(l)}\big)$, and the equation 4 as follows: $E^{(l+1)} = \text{GNN}\big(E^{(l)}, M^T, X^{(l)}\big) = \text{EHGNN}\big(X^{(l)}, M, E^{(l)}\big)$, where EHGNN indicates our edge representation learning framework using DHT. After updating the edge features $E^{(L)}$ with EHGNN, $E^{(L)}$ is returned to the original graph by applying DHT on the dual hypergraph $G^*$. Then, the remaining step is how to make use of these edge-wise representations to accurately represent the edges of the entire graph, which we describe in the next subsection.

## 3.2 Graph-level edge representation learning with edge pooling

Existing graph pooling methods do not explicitly represent edges. To overcome this limitation, we propose two novel edge pooling schemes: *HyperCluster* and *HyperDrop*.

**Graph pooling**  The goal of graph pooling is to learn a holistic representation of the entire graph. The most straightforward approach for this is to aggregate all the node features with `mean` or `sum` operations [1, 39], but they treat all nodes equally without consideration of which nodes are important for the given task. To tackle this limitation, recent graph pooling methods propose to either cluster and coarsen nodes [41, 4] or drop unnecessary nodes [11, 24]. While they yield improved performances on graph classification tasks, they suffer from an obvious drawback: inevitable loss of both node and edge information. The node information is lost as nodes are dropped and coarsened, and the edge information is lost as edges for the dropped nodes or internal edges for the coarsened nodes are removed. To overcome this limitation, we propose a graph-level edge representation learning scheme.

**HyperCluster**  We first introduce *HyperCluster*, which is a novel edge clustering method to coarsen similar edges into a single edge, for obtaining the global edge representation. Generally, a clustering scheme for nodes of the graph [41, 4] is defined as follows:

$$\boldsymbol{X}^{pool} = \boldsymbol{C}^T \boldsymbol{X}', \quad \boldsymbol{M}^{pool} = \boldsymbol{C}^T \boldsymbol{M}, \tag{5}$$

where $\boldsymbol{X}^{pool} \in \mathbb{R}^{n_{pool} \times d}$ and $\boldsymbol{M}^{pool} \in \mathbb{R}^{n_{pool} \times m}$ denote the pooled representations, $\boldsymbol{X}' = \text{GNN}(\boldsymbol{X}, \boldsymbol{M}, \boldsymbol{E}) \in \mathbb{R}^{n \times d}$ is the updated node features, and $\boldsymbol{C} \in \mathbb{R}^{n \times n_{pool}}$ is the cluster assignment matrix that is generated from the $\boldsymbol{X}'$. Following this approach, the proposed HyperCluster clusters similar edges into a single edge, by clustering nodes of the dual hypergraph obtained from the original graph via DHT. In other words, we first obtain the node representation of the dual hypergraph $\boldsymbol{E}' = \text{EHGNN}(\boldsymbol{X}, \boldsymbol{M}, \boldsymbol{E}) \in \mathbb{R}^{m \times d'}$, and then cluster the nodes of the dual hypergraph as follows:

$$\boldsymbol{E}^{pool} = \boldsymbol{C}^T \boldsymbol{E}', \quad (\boldsymbol{M}^{pool})^T = \boldsymbol{C}^T \boldsymbol{M}^T \tag{6}$$

where $\boldsymbol{E}^{pool} \in \mathbb{R}^{m_{pool} \times d'}$ and $\boldsymbol{M}^{pool} \in \mathbb{R}^{n \times m_{pool}}$ denote the pooled edge representation and the incidence matrix of the input graph respectively, and $\boldsymbol{C} \in \mathbb{R}^{m \times m_{pool}}$ is the cluster assignment matrix generated from the input edge features $\boldsymbol{E}'$. Since HyperCluster coarsens the edges rather than dropping them, this edge pooling method is more appropriate for tasks such as graph reconstruction.

**HyperDrop**  We propose another edge pooling scheme, *HyperDrop*, which drops unnecessary edges to identify task-relevant edges, while performing lossless compression of nodes. Conventional node drop methods [11, 24] remove less relevant nodes based on their scores, as follows:

$$\boldsymbol{X}^{pool} = \boldsymbol{X}_{idx}, \quad \boldsymbol{M}^{pool} = \boldsymbol{M}_{idx} \; ; \; idx = \text{top}_k(\text{score}(\boldsymbol{X})), \tag{7}$$

where $idx$ is the row-wise (i.e., node-wise) indexing vector, $\text{score}(\cdot)$ computes the score of each node with learnable parameters, and $\text{top}_k(\cdot)$ selects the top $k$ elements in terms of the score. However, this approach results in the inevitable loss of node information, as it drops nodes. Thus, we propose to coarsen the graph by dropping edges instead of nodes, exploiting edge representations obtained from our EHGNN. *HyperDrop* selects the top-ranked edges of the original graph, by selecting the top-ranked nodes of the dual hypergraph. The pooling procedure for HyperDrop is as follows:

$$\boldsymbol{E}^{pool} = \boldsymbol{E}_{idx}, \quad (\boldsymbol{M}^{pool})^T = (\boldsymbol{M}^T)_{idx} \; ; \; idx = \text{top}_k(\text{score}(\boldsymbol{E})). \tag{8}$$

Then, we can obtain the pooled graph $G^{pool} = (\boldsymbol{X}, \boldsymbol{M}^{pool}, \boldsymbol{E}^{pool})$ by applying DHT to the pooled dual hypergraph. HyperDrop is most suitable for graph classification tasks, as it identifies discriminative edges for the given task. Since HyperDrop preserves the nodes intact, it can also be used for node-level classification tasks, which is impossible with exiting graph pooling methods that modify nodes. In another point of view, the proposed HyperDrop can be further considered as a learnable graph rewiring operation, which optimizes the graph for the given task by deciding whether to drop or maintain the nodes. Finally, a notable advantage of such HyperDrop is that it alleviates the over-smoothing problem in deep GNNs [25] (i.e., the features of all nodes converge to the same values when stacking a large number of GNN layers). As HyperDrop *learns* to remove unnecessary edges, the message-passing only happens across relevant nodes, which alleviates over-smoothing.

Figure 3: **Edge reconstruction results** on the ZINC molecule dataset by varying the pooling ratio. Solid lines denote the mean, and shaded areas denote the standard deviation of 5 runs.

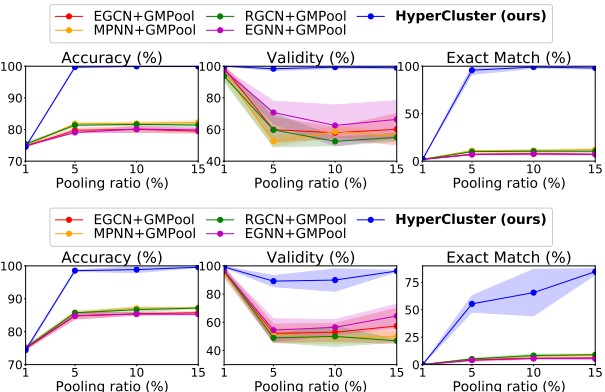

Figure 5: **Graph reconstruction results** on the ZINC molecule dataset by varying the pooling ratio. Solid lines denote the mean, and shaded areas denote the standard deviation of 5 runs.

Figure 4: **Edge reconstruction results** of the synthetic two-moon graph. The edge features are represented by colors.

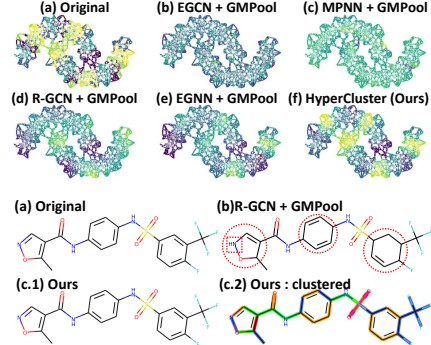

Figure 6: **Graph reconstruction examples.** Red dashed circles and squares indicate the incorrectly predicted edges and nodes, respectively. (c.2) shows the assigned clusters of edges as colors using our method.

## 4   Experiments

We experimentally validate the effectiveness of EHGNN coupled with either HyperCluster or Hyper-Drop on four different tasks: graph reconstruction, generation, classification, and node classification.

### 4.1   Graph reconstruction

Accurately reconstructing the edge features is crucial for graph reconstruction tasks, and thus we validate the efficacy of our method on graph reconstruction tasks first.

**Experimental setup**   We first validate our EHGNN with HyperCluster on the *edge reconstruction* tasks, where the goal is to reconstruct the edge features from their compressed representations. Then, we evaluate our method on the graph reconstruction tasks to validate the effectiveness of ours in holistic graph-level learning. We start with edge reconstruction of a synthetic two-moon graph, where node features (coordinates) are fixed and edge features are colors. For edge and graph reconstruction of real-world graphs, we use the ZINC dataset [19] that consists of 12K molecular graphs [7], where node features are atom types and edge features are bond types. We use accuracy, validity, and exact match as evaluation metrics. For more details, please see Section C.1 of the supplementary file.

**Implementation details and baselines**   We compare the proposed EHGNN framework against edge-aware GNNs, namely EGCN [17], MPNN [12], R-GCN [32], and EGNN [13], which use the edge features as auxiliary information for updating node features. We further combine them with an existing graph pooling method, namely GMPool [2], to obtain a graph-level edge representation for a given graph. In contrast, for our method, we first obtain edge representations with EHGNN, using GCN [23] as the message-passing function, and then coarsen the edge-wise representations using HyperCluster, whose cluster assignment matrices are obtained using GMPool [2]. For node reconstruction, we set message-passing to GCN and graph pooling to GMPool [2] for all models. We provide further details of the baselines and our model in Section C.1 of the supplementary file.

**Edge reconstruction results**   Figure 4 shows the original two-moon graph and edge-reconstructed graphs, where edge features are represented as colors, exhibiting clustered patterns. The baselines fail to reconstruct the edge colors, since they implicitly learn edge representations by using edge features as auxiliary information to update nodes, hence mixing the colors of the neighboring edges. On the other hand, our method distinguishes each edge cluster, which shows that our method can capture meaningful edge information by clustering similar edges. Moreover, as shown in Figure 3, our model obtains significantly higher performance over all baselines on the edge reconstruction task of molecular graphs, in all evaluation metrics. The performance gain of our method over baselines is notably large in exact match, which demonstrates that explicit learning of edge representation is essential for the accurate encoding of the edge information.

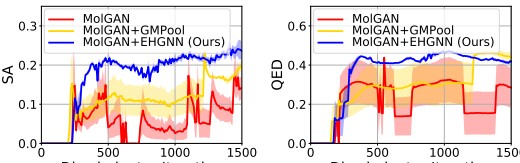

Figure 7: **Graph generation results on MolGAN.** Solid lines denote the mean, and shaded areas denote the standard deviation of 3 different runs.

| Dataset | Metrics | MARS [38] | MARS + EHGNN (Ours) |
|---------|---------|-----------|---------------------|
| ZINC15 | Success Rate | $59.53 \pm 2.11$ | $\mathbf{64.30} \pm 1.54$ |
| | QED ($\geq 0.67$) | $95.71 \pm 0.09$ | $\mathbf{96.36} \pm 0.49$ |
| | GSK3$\beta$ ($\geq 0.6$) | $86.52 \pm 1.67$ | $\mathbf{90.63} \pm 2.57$ |
| | JNK3 ($\geq 0.6$) | $71.52 \pm 4.15$ | $\mathbf{73.60} \pm 1.29$ |

Table 2: **Graph generation results on MARS.** The results are the mean and standard deviation of 3 different runs. Best performance and its comparable results ($p > 0.05$) from the t-test are highlighted in bold.

**Graph reconstruction results**   To verify the effectiveness of learning accurate edge representations for reconstructing both the node and edge features, we now validate our method on the molecular graphs in Figure 5. Combining our edge representation learning method (EHGNN + HyperCluster) with the existing node representation learning method (GCN + GMPool) yields incomparably high reconstruction performance compared to the baselines in exact match, which demonstrates that learning accurate edge representation, as well as node representation, is crucial to the success of the graph representation learning methods on graph reconstruction.

**Qualitative analysis**   We visualize the original and reconstructed molecular graphs in Figure 6. As shown in Figure 6 (b), the baseline cannot reconstruct the ring structures of the molecule, whereas our method perfectly reconstructs the rings as well as the atom types. The generated edge clusters in Figure 6 (c.2) further show that our method captures the detailed substructures of the molecule, as we can see the cluster patterns of hexagonal and pentagonal rings. More reconstruction examples of molecular graphs are shown in Figure 3 of the supplementary file.

**Graph compression**   To validate the effectiveness of our method in large and dense graph compression, we further apply EHGNN with HyperCluster to the Erdos-Renyi random graph [8] having six discrete edge features, where the number of nodes is fixed to $10^3$ while the number of edges increases from $10^3$ to $10^4$. In Figure 8, we report the relative memory size of the compressed graph after pooling the features, against the size of the original graph. We compare our method which compresses both the nodes and edges, against the node pooling method, namely GMT [2]. As the number of edges increases, we observe that compressing only the node features is insufficient for obtaining compact representations, whereas our method is able to obtain highly compact but accurate representation which can be assured from the sufficiently high edge

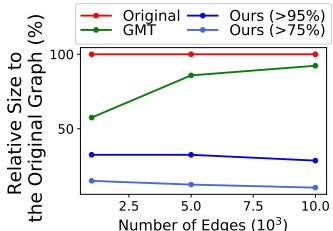

Figure 8: **Graph compression results.** For ours, we report the relative size to the original graph where the edge reconstruction accuracy is higher than 95% and 75%, respectively.

reconstruction accuracy. We believe that our proposed framework can not only learn accurate representations of nodes and edges, but also effectively compress their features, especially for large-scale real-world graphs, such as social networks or protein-protein interaction (PPI) graphs.

## 4.2   Graph generation

As shown in Figure 1 (left), graph generation depends heavily on the edge representations, as the model may generate incorrect graphs (e.g., toxic chemicals rather than drugs) if the edge information is inaccurate. Thus, we further validate our EHGNN on the graph generation tasks.

**Experimental setup**   We directly forward the edge representation from the EHGNN to molecule generation networks, namely MolGAN [5] and MArkov moleculaR Sampling (MARS) [38]. MolGAN uses the Generative Adversarial Network (GAN) [14], to generate the molecular graph by balancing weights between its generator and discriminator. MolGAN uses R-GCN [32] for node-level message-passing, whereas, for ours, we first obtain the edge representations using EHGNN, and use them with mean pooling in the graph encoder. For evaluation metrics, we use the Synthetic Accessibility (SA) and Druglikeness (QED) scores. We further apply EHGNN to MARS [38] that generates the molecule by sequentially adding or deleting its fragment, with MCMC sampling. While the original model uses MPNN [12] to implicitly obtain edge representations for adding and deleting actions, we use EHGNN to explicitly learn edge representations. We train models to maximize four

Table 3: **Graph classification results**. The results are the mean and standard deviation over 10 different runs. Best performance and its comparable results ($p > 0.05$) from the t-test are highlighted in bold. Hyphen (-) denotes out-of-resources that take more than 10 days. The results for the baselines are taken from Baek et al. [2].

| | | TU : Biochemical | | | TU : Social | | | OGB : Molecule | | | | Average |
|---|---|---|---|---|---|---|---|---|---|---|---|---|
| | | **D&D** | **PROTEINS** | **MUTAG** | **IMDB-B** | **IMDB-M** | **COLLAB** | **HIV** | **Tox21** | **ToxCast** | **BBBP** | |
| # graphs | | 1178 | 1113 | 188 | 1000 | 1500 | 5000 | 41127 | 7831 | 8576 | 2039 | |
| # classes | | 2 | 2 | 2 | 2 | 3 | 3 | 2 | 12 | 617 | 2 | |
| Avg # nodes | | 284.32 | 39.06 | 17.93 | 19.77 | 13.00 | 74.49 | 25.51 | 18.57 | 18.78 | 24.06 | |
| Avg # edges | | 715.66 | 72.82 | 19.79 | 96.53 | 65.94 | 2457.78 | 27.47 | 19.27 | 19.26 | 25.95 | |
| **Set** | DeepSet | $77.39 \pm 0.67$ | $68.95 \pm 0.92$ | $72.56 \pm 1.09$ | $72.42 \pm 0.36$ | $50.24 \pm 0.32$ | $75.27 \pm 0.21$ | $71.20 \pm 1.26$ | $72.25 \pm 0.23$ | $59.44 \pm 0.39$ | $63.64 \pm 0.62$ | 68.34 |
| **Naive GNN** | GCN | $72.05 \pm 0.55$ | $73.24 \pm 0.73$ | $69.50 \pm 1.78$ | $73.26 \pm 0.46$ | $50.39 \pm 0.41$ | $80.59 \pm 0.27$ | $76.81 \pm 1.01$ | $75.04 \pm 0.80$ | $60.63 \pm 0.51$ | $65.47 \pm 1.73$ | 69.70 |
| | GIN | $70.79 \pm 1.17$ | $71.46 \pm 1.66$ | $81.39 \pm 1.53$ | $72.78 \pm 0.86$ | $48.13 \pm 1.36$ | $78.19 \pm 0.63$ | $75.95 \pm 1.35$ | $73.27 \pm 0.84$ | $60.83 \pm 0.46$ | $\mathbf{67.65 \pm 3.00}$ | 70.04 |
| **Global** | SortPool | $75.58 \pm 0.72$ | $73.17 \pm 0.88$ | $71.94 \pm 3.55$ | $72.12 \pm 1.12$ | $48.18 \pm 0.83$ | $77.87 \pm 0.47$ | $71.82 \pm 1.63$ | $69.54 \pm 0.75$ | $58.69 \pm 1.71$ | $65.98 \pm 1.70$ | 68.49 |
| | GMT | $\mathbf{78.72 \pm 0.59}$ | $\mathbf{75.09 \pm 0.59}$ | $83.44 \pm 1.33$ | $73.48 \pm 0.76$ | $50.66 \pm 0.82$ | $80.74 \pm 0.54$ | $\mathbf{77.56 \pm 1.25}$ | $\mathbf{77.30 \pm 0.59}$ | $\mathbf{65.44 \pm 0.58}$ | $\mathbf{68.31 \pm 1.62}$ | 73.07 |
| **Hierarchical** | DiffPool | $77.56 \pm 0.41$ | $73.03 \pm 1.00$ | $79.22 \pm 1.02$ | $73.14 \pm 0.70$ | $\mathbf{51.31 \pm 0.72}$ | $78.68 \pm 0.43$ | $75.64 \pm 1.86$ | $74.88 \pm 0.81$ | $62.28 \pm 0.56$ | $\mathbf{68.25 \pm 0.96}$ | 71.40 |
| | SAGPool | $74.72 \pm 0.82$ | $71.56 \pm 1.49$ | $73.67 \pm 4.28$ | $72.55 \pm 1.28$ | $50.23 \pm 0.44$ | $78.03 \pm 0.31$ | $71.44 \pm 1.67$ | $69.81 \pm 1.75$ | $58.91 \pm 0.80$ | $63.94 \pm 2.59$ | 68.49 |
| | TopKPool | $73.63 \pm 0.55$ | $70.48 \pm 1.01$ | $67.61 \pm 3.36$ | $71.58 \pm 0.95$ | $48.59 \pm 0.72$ | $77.58 \pm 0.85$ | $72.27 \pm 0.91$ | $69.39 \pm 2.02$ | $58.42 \pm 0.91$ | $65.19 \pm 2.30$ | 67.47 |
| | MinCutPool | $\mathbf{78.22 \pm 0.54}$ | $74.72 \pm 0.48$ | $79.17 \pm 1.64$ | $72.65 \pm 0.75$ | $51.04 \pm 0.70$ | $80.87 \pm 0.34$ | $75.37 \pm 2.05$ | $75.11 \pm 0.69$ | $62.48 \pm 1.33$ | $65.97 \pm 1.13$ | 71.56 |
| | ASAP | $76.58 \pm 1.04$ | $73.92 \pm 0.63$ | $77.83 \pm 1.49$ | $72.81 \pm 0.50$ | $50.78 \pm 0.75$ | $78.64 \pm 0.50$ | $72.86 \pm 1.40$ | $72.24 \pm 1.66$ | $58.09 \pm 1.62$ | $63.50 \pm 2.47$ | 69.73 |
| | EdgePool | $75.85 \pm 0.58$ | $\mathbf{75.12 \pm 0.76}$ | $74.17 \pm 1.82$ | $72.46 \pm 0.74$ | $50.79 \pm 0.59$ | - | $72.66 \pm 1.70$ | $73.77 \pm 0.68$ | $60.70 \pm 0.92$ | $67.18 \pm 1.97$ | - |
| | HaarPool | - | - | $66.11 \pm 1.50$ | $73.29 \pm 0.34$ | $49.98 \pm 0.57$ | - | - | - | - | $66.11 \pm 0.82$ | - |
| **Ours** | HyperDrop | $\mathbf{78.74 \pm 0.68}$ | $\mathbf{75.30 \pm 0.45}$ | $84.00 \pm 0.69$ | $73.96 \pm 0.41$ | $\mathbf{51.68 \pm 0.41}$ | $\mathbf{81.29 \pm 0.25}$ | $76.79 \pm 0.86$ | $76.95 \pm 0.32$ | $64.21 \pm 0.70$ | $\mathbf{69.04 \pm 0.86}$ | 73.20 |
| | HyperDrop + GMT | $\mathbf{78.39 \pm 0.33}$ | $\mathbf{75.39 \pm 0.26}$ | $\mathbf{85.72 \pm 0.61}$ | $\mathbf{74.45 \pm 0.61}$ | $\mathbf{51.45 \pm 0.28}$ | $80.59 \pm 0.33$ | $\mathbf{77.84 \pm 0.37}$ | $\mathbf{77.58 \pm 0.43}$ | $65.15 \pm 0.65$ | $\mathbf{69.16 \pm 1.04}$ | 73.57 |

molecule properties: inhibition scores against two proteins, namely GSK3$\beta$ and JNK3 (biological); QED and SA scores (non-biological). Then we report the success rate at which the molecule satisfies all the properties. For more details, please see Section C.2 of the supplementary file.

**MolGAN results**  Figure 7 shows the SA and QED scores of the generated molecules, of the Mol-GAN architecture with different encoders. Our EHGNN framework obtains significantly improved generation performance, over the original MolGAN which uses the R-GCN encoder and the MolGAN with GMPool, a state-of-the-art global node pooling encoder. This is because EHGNN learns explicit edge representation which enhances the ability of the discriminator to distinguish between real and generated graphs. The improvement in the discriminator also leads to notably more stable results compared to the baselines, which show a large variance in the quality of the generated molecules.

**MARS results**  To perform correct editing actions to generate graphs with MARS, we need accurate edge representations, as edges determine the structure of the generated molecule. Table 2 shows that using our EHGNN achieves significantly higher generation performance over original MARS, which uses edges as auxiliary information only to enhance node representations. Notably, performance gain on the GSK3$\beta$ inhibition score for which structural binding is important, suggests that accurate learning of edges leads to generating more effective molecules that interact with the target protein.

### 4.3   Graph and node classification

Now, we validate the performance of our EHGNN with HyperDrop on classification tasks. Our approach is effective for the classification of graphs with or without edge features, since it allows lossless compression of nodes, and drops edges to allow message-passing only across relevant nodes.

**Experimental setup**  Following the experimental setting of Baek et al. [2], we use the GCN as the node-level message-passing layers for all models, and compare our edge pooling method against the existing graph pooling methods. For this experiment, our HyperDrop uses SAGPool [24] on the hypergraph, which is a node drop pooling method based on self-attention. We use 6 datasets from the TU datasets [28] including three from the biochemical domains (i.e., DD, PROTEINS, MUTAG) and the remaining half from the social domains (i.e., IMDB-BINARY, IMDB-MULTI, COLLAB). Also, we further use the 4 molecule datasets (i.e., HIV, Tox21, ToxCast, BBBP) from the recently released OGB datasets [17]. We evaluate the accuracy of each model with 10-fold cross validation [43] on the TU datasets, and use ROC-AUC as the evaluation metric for the OGB datasets. For both datasets, we follow the standard experimental settings, from the feature extraction to the dataset splitting. We provide additional details of the experiments in Section C.3 of the supplementary file.

**Baselines**  We compare our EHGNN with HyperDrop, against the set encoding (DeepSet [42]), GNNs with naive pooling baselines (GCN and GIN [23, 39]), and state-of-the-art hierarchical pooling methods (DiffPool [41], SAGPool [24], TopKPool [11], MinCutPool [4], ASAP [29], EdgePool [6], and HaarPool [35]) that drop or coarsen node representations. We also additionally compare or combine the state-of-the-art global node pooling methods (SortPool [43], GMT [2]) with our model, for example, HyperDrop + GMT. For more details, see Section C.3 of the supplementary file.

| Model | MUTAG | PROTEINS | Tox21 |
|---|---|---|---|
| HyperDrop | 84.00 ± 0.69 | **75.30** ± 0.45 | **76.95** ± 0.32 |
| HyperCluster | **84.50** ± 1.50 | 72.76 ± 1.12 | 76.68 ± 0.56 |
| w/ RandDrop | 83.06 ± 1.15 | 74.92 ± 0.51 | 76.39 ± 0.47 |
| w/o HyperDrop | 83.06 ± 1.20 | 75.08 ± 0.37 | 76.60 ± 0.45 |
| w/o EHGNN | 69.50 ± 1.78 | 73.24 ± 0.73 | 75.04 ± 0.80 |

Table 4: **Ablation study** of Hyper-Drop on the MUTAG, PROTEINS, and Tox21 datasets for classification.

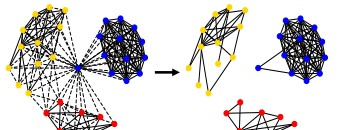

Figure 9: **Edge pooling results** on the COLLAB dataset. Colors denote connected components.

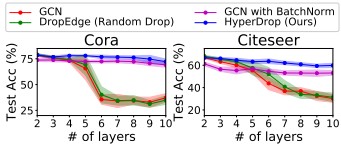

Figure 10: **Node classification results.** Lines denote means over 10 runs and shades denote variances.

**Classification results**  Table 3 shows that the proposed EHGNN with HyperDrop significantly outperforms all hierarchical pooling baselines. This is because HyperDrop not only retains nodes by removing edges that are less useful for graph discrimination, but also explicitly uses the edge representations for graph classification. Since HyperDrop does not remove any nodes on the graph, it can be jointly used with any node pooling methods, and thus, we pair HyperDrop with GMT. This model largely outperforms GMT, obtaining the best performance on most of the datasets, which demonstrates that accurate learning of both the nodes and edges is important for classifying graphs. We further visualize the edge pooling process of HyperDrop in Figure 9, which shows that our method accurately captures the substructures of the entire graph, which leads to dividing the large graph into several connected components, thus adjusting the graph topology for more effective message-passing. We provide more visual examples of edge drop procedures in Section D.3 of the supplementary file.

**Ablation study**  To see how much each component contributes to the performance gain, we conduct an ablation study on EHGNN with HyperDrop. Table 4 shows that, compared with a model that only uses node features (i.e., w/o EHGNN), learning explicit edge representations significantly improves performance. Our model EHGNN with HyperCluster, or without HyperDrop, or the model with random edge drop obtains decent performance, but substantially underperforms HyperDrop.

**Over-smoothing with deep GNNs**  Lastly, we demonstrate that our EHGNN with HyperDrop alleviates the over-smoothing problem of deep GNNs on semi-supervised node classification tasks, which is not possible for the existing node-based pooling methods. We follow the settings of existing works [23, 34, 10] and provide the experimental details in Section C.4 of the supplementary file. As shown in Figure 10, HyperDrop retains the accuracy as the number of layers increases, whereas the naive GCN or random drop [30] results in largely degraded performance, since HyperDrop identifies and preserves the task-relevant edges while the sampling-based methods randomly drop the edges. Further, our method outperforms BatchNorm which alleviates over-smoothing by yielding differently normalized feature distribution at each batch. This is because HyperDrop splits the given graph into smaller subgraphs that capture meaningful message-passing substructures as shown in Figure 9.

## 5  Conclusion

We tackled the problem of accurately representing the edges of a graph, which has been relatively overlooked over node representation learning. To this end, we proposed a novel edge representation learning framework using *Dual Hypergraph Transformation* (DHT), which transforms the edges of the original graph into nodes on a hypergraph. This allows us to apply a message-passing scheme for node representation learning, for edge representation learning. Further, we proposed two edge pooling methods to obtain a holistic edge representation for a given graph, where one clusters similar edges into a single edge for graph reconstruction and the other drops unnecessary edges for graph classification. We validated our edge representation learning framework on graph reconstruction, generation, and classification tasks, showing its effectiveness over relevant baselines.

## 6  Acknowledgements and Disclosure of Funding

We thank the anonymous reviewers for their constructive comments and suggestions. This work was supported by Institute of Information & communications Technology Planning & Evaluation (IITP) grant funded by the Korea government (MSIT) (No.2019-0-00075, Artificial Intelligence Graduate School Program (KAIST), and No.2021-0-02068, Artificial Intelligence Innovation Hub), and the Engineering Research Center Program through the National Research Foundation of Korea (NRF) funded by the Korean Government MSIT (NRF-2018R1A5A1059921).

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
