# [Supplementary File]
# Edge Representation Learning with Hypergraphs

**Organization** The supplementary file is organized as follows. In section A, we first describe the structural details of the proposed *Edge HyperGraph Neural Network* (EHGNN) framework using the *Dual Hypergraph Transformation* (DHT) in comparison to those of the existing edge-aware graph neural networks. Also, we describe the detailed components of the proposed edge pooling methods: *HyperCluster* and *HyperDrop* in section B. We provide the experimental setups in Section C, which include the detailed descriptions of the models and datasets, as well as the experimental details for each task. Then, we provide additional experimental results on graph reconstruction and generation tasks with visualization of examples in Section D. Finally, We discuss the limitations and potential societal impacts of our work in Section E, and provide the NeurIPS paper Checklist in Section F.

## A   Edge Representation Learning

In this section, we first describe the detailed formulations of existing edge-aware graph neural networks (GNNs), and compare them with our methods. Then, we introduce the time complexity of making connectivity patterns for edges using existing edge-aware GNNs and our *Edge HyperGraph Neural Network* (EHGNN). Finally, we discuss the sparse implementation of the proposed EHGNN along with our *Dual Hypergraph Transformation* (DHT) at this end of the section.

### A.1   Discussion on edge-aware graph neural networks

Here, we first formalize the existing edge-aware GNNs that we used as baselines [17, 14, 33, 15]. We begin by introducing the basic components of GNNs: $\boldsymbol{X}^{(l)}$ denotes the node features at $l$-th layer, $\boldsymbol{W}$ denotes the learnable weight matrix, $\boldsymbol{E}$ denotes the edge features, and $\mathcal{N}(v)$ denotes the neighboring node set of node $v$ in the given graph.

**EGCN** The node-wise formulation of edge-aware GCN [17] is defined as follows:

$$\boldsymbol{X}_v^{(l+1)} = \boldsymbol{W} \sum_{u \in \mathcal{N}(v) \cup \{v\}} n_{u,v} \cdot (\boldsymbol{X}_u^{(l)} + \boldsymbol{E}_{u,v}) \tag{1}$$

where $n_{u,v}$ is the normalizing coefficient for two adjacent nodes $u$ and $v$, and edge feature $\boldsymbol{E}$ is obtained from the categorical edge features without learning.

**MPNN** The node-wise formulation of MPNN [14] using edge conditioned convolution [35] is defined as follows:

$$\boldsymbol{X}_v^{(l+1)} = \boldsymbol{W} \boldsymbol{X}_v^{(l)} + \sum_{u \in \mathcal{N}(v)} \boldsymbol{X}_u^{(l)} \cdot \text{MLP}(\boldsymbol{E}_{u,v}) \tag{2}$$

where MLP is a linear layer for learning the edge representations to augment the node representations.

**R-GCN** The node-wise formulation of R-GCN [33] is defined as follows:

$$\boldsymbol{X}_v^{(l+1)} = \boldsymbol{W} \boldsymbol{X}_v^{(l)} + \sum_{r \in \mathcal{R}} \sum_{u \in \mathcal{N}_r(v)} \frac{1}{|\mathcal{N}_r(v)|} \boldsymbol{W}_r \boldsymbol{X}_u^{(l)} \tag{3}$$

where $\mathcal{R}$ is a set of categorical edge types, and $\mathcal{N}_r(v)$ denotes the neighboring node set of the node $v$, having the associated edge type $r \in \mathcal{R}$.

**EGNN** The node-wise formulation of convolution-based edge GNN [15] is defined as follows:

$$\boldsymbol{X}_v^{(l+1)} = \boldsymbol{W} \sum_{u \in \mathcal{N}(v) \cup \{v\}} \boldsymbol{E}_{u,v}^{(l)} \boldsymbol{X}_u^{(l)} \tag{4}$$

| # of edges ($10^3$) | 2 | 4 | 8 | 16 | 32 | 64 |
|---|---|---|---|---|---|---|
| Line graph | 32.78 | 65.93 | 131.36 | 260.92 | 527.44 | 1071.31 |
| DHT (ours) | **0.13** | **0.18** | **0.26** | **0.45** | **0.81** | **1.37** |

Table 1: **Transformation time(s) results** of the line graph and our dual hypergraph. The results are the average of over 100 runs.

| Graph | Message-passing | |
|---|---|---|
| | Nodes (GCN) | Edges (Ours) |
| Erdos-Renyi | 0.0031 | 0.0034 |
| Barabasi-Albert | 0.0029 | 0.0032 |

Table 2: **Message-passing time(s)** on the original graph and our dual hypergraph.

where the edge features at $l$-th layer $\boldsymbol{E}^{(l)}$ are obtained by edge-level layers which are differently designed from node-level layers. The features are used as the attention coefficients for nodes to enhance the node-level representations.

It is worthwhile to note that all baselines only implicitly capture edge information in the learned node representations rather than directly using it for downstream graph tasks, while our EHGNN framework explicitly learns and utilizes the learned edge representations.

## A.2 Sparse implementation of the dual hypergraph transformation

Since most graphs have relatively few connections per node, the number of non-zero elements in the adjacency matrix, which defines the connection among nodes, is smaller than the number of zero elements. Thus, using the adjacency matrix for message-passing is highly inefficient in terms of memory usage. To handle this issue, the most dominant approach is to use an edge list, which is a sparse representation of the adjacency matrix (or the incidence matrix) of the graph. Specifically, each column of the edge list $\boldsymbol{L} \in \mathbb{R}^{2 \times m}$ denotes an edge $e$, which has two incident nodes $(v_{start}, v_{end})$, where $v_{start}$ denotes the start node and $v_{end}$ denotes the end node of the edge $e$.

Similarly, the incidence matrix of a hypergraph can be represented as a sparse form using a hyperedge list $\boldsymbol{L}^* \in \mathbb{R}^{2 \times D}$, where $D$ is the sum of degrees of all nodes in the hypergraph. Each column of $L^*$ indicates a hyperedge $e^*$ with a (node, hyperedge) pair $(v^*, e^*)$, where $v^*$ is the node incident to the hyperedge $e^*$. If the hyperedge is incident to three nodes, then it will appear on three columns of $L^*$ paired with each incident node. Compared to this general hypergraph, the dual hypergraph obtained by DHT is 2-regular, which means each node in the hypergraph has a degree of two, since each edge in the original graph is incident to exactly two nodes. Thanks to this property, the hyperedge list of the dual hypergraph has the dimensionality of $2 \times 2m$ (i.e., $\boldsymbol{L}^* \in \mathbb{R}^{2 \times 2m}$).

Then, the concrete implementation of DHT with the sparse edge list of the original graph and the sparse hyperedge list of its dual hypergraph is formalized as follows:

$$\boldsymbol{DHT} \; : \; G = \big(\boldsymbol{X}, \boldsymbol{L}, \boldsymbol{E}\big) \; \mapsto \; G^* = \big(\boldsymbol{E}, \boldsymbol{L}^*, \boldsymbol{X}\big), \tag{5}$$

where the hyperedge list $\boldsymbol{L}^*$ is obtained by reshaping the edge list $\boldsymbol{L}$ as follows:

$$\boldsymbol{L}^*_{1,2i-1} = \boldsymbol{L}^*_{1,2i} = i, \tag{6}$$
$$\boldsymbol{L}^*_{2,2i-1} = \boldsymbol{L}_{1,i} \;, \quad \boldsymbol{L}^*_{2,2i} = \boldsymbol{L}_{2,i}, \tag{7}$$

for all $1 \leq i \leq m$.

## A.3 Complexity analysis

In this subsection, we provide the detailed complexity analysis of transformation and message-passing operations of existing edge-aware GNNs [21, 15, 42] and our DHT. We first introduce the transformation complexity, and then describe the message-passing complexity.

**Transformation complexity** To define the adjacency of edges to perform message-passing between edges, previous works either define the edge neighborhood structure [42], or use the line graph transformation [21]. Constructing edge neighborhood takes $\mathcal{O}(m^2)$ for transforming the node adjacency to the edge adjacency, as, for verifying two edges are adjacent, we need to first sample one edge among $m$ edges, and then find the other edge that shares the same node among the remaining $m-1$ edges. In a similar manner, the complexity of line graph transformation is quadratic to the number of edges Monti et al. [26], as, for each pair of edges, we need to verify whether they share the same node. However, with our sparse implementation of DHT explained in A.2, we can obtain the hyperedge list – a sparse data structure of the hypergraph – by simply reshaping the given edge list of the original graph, which takes at most $\mathcal{O}(m)$.

We further experimentally verify the transformation complexity of the line graph transformation [21] and the proposed DHT, on Erdos-Renyi graph [8] with 1000 nodes and the number of edges increasing from $2 \times 10^3$ to $64 \times 10^3$. As shown in Table 1, our DHT is highly efficient compared to the line graph transformation, especially for large and dense graphs, as the line graph transformation is quadratic to the number of edges, whereas ours only requires simple tensor-reshape operations.

**Message-passing complexity**   Note that the complexity of message-passing on the graphs depends only on the number of edges, thus it is enough to focus on the number of edges. When we transform the original graph into the line graph following Jiang et al. [21], the constructed line graph has $\mathcal{O}(m \cdot d_{max})$ edges, therefore the complexity of message-passing is $\mathcal{O}(m \cdot d_{max})$. For instance, when the input graph is a star graph having one hub node and $n$ other nodes (i.e., the number of edges is $n$), the line graph of the star graph has $n^2$ number of edges, thus the message-passing cost is $\mathcal{O}(n^2)$, as shown in Table 1 of the main paper. However, with our DHT implemented over the sparse hyperedge list, we only have $2m$ number of node-hyperedge pairs as explained in Section A.2, thus we can perform the message-passing between edges (nodes of the dual hypergraph) with complexity $\mathcal{O}(m)$. This complexity is equal to that of the message-passing between nodes of the original graph. In other words, the analytical complexity of message-passing between edges in equation 4 of the main paper is equivalent to the complexity of message-passing between nodes in equation 1 of the main paper.

We experimentally validate the message-passing complexity on the original graph (message-passing between nodes) and the dual hypergraph (message-passing between edges) in Table 2. We evaluate the message-passing time on both the Erdos-Renyi graph [8] and the scale-free (Barabasi-Albert) network [2], with 3000 nodes 11984 edges following the densification law (i.e. $m \propto n^{1.18}$ [25]) of the internet graph. Table 2 shows that message-passing time on the dual hypergraph is almost equal to the message-passing time on the original graph, which coincides with the previous analysis.

## B   Details for Edge Pooling Schemes

In this section, we describe the proposed two novel edge pooling schemes: *HyperCluster* that coarsens similar edges for global edge representations, and *HyperDrop* that drops unnecessary edges for hierarchical graph representations.

### B.1   HyperCluster

Our cluster-based edge pooling model, HyperCluster, consists of edge-level message-passing layers (i.e., EHGNN layers) and HyperCluster layers, which we describe below in detail. Before clustering edges, we first update the edge features using multiple EHGNN layers as follows:

$$\boldsymbol{E}^{(l+1)} = \text{EHGNN}(\boldsymbol{X}, \boldsymbol{M}, \boldsymbol{E}^{(l)}), \tag{8}$$

where $\boldsymbol{E}^{(l)}$ denotes the updated edge features at the $l$-th layer from the initial edge features $\boldsymbol{E}^{(0)} = \boldsymbol{E}$, and we finally obtain $\boldsymbol{E}' = \boldsymbol{E}^{(L)}$ after $L$ number of EHGNN layers. Then, to obtain the global edge representation of the entire graph, we cluster the nodes of its dual hypergraph using the node clustering method. While we can use any off-the-shelf node clustering methods [43, 3, 1], in this paper, we use the state-of-the-art pooling method, namely GMPool [1]. To apply GMPool on a hypergraph, we modify the graph multi-head attention block (GMH), which is used to construct key and value matrices using GNNs for the original graph structure in the GMPool paper [1], for the hypergraph structure by replacing the adjacency matrix to the incidence matrix. We compress $m$ nodes in the dual hypergraph into $k$ nodes with the modified $\text{GMPool}_k$, formalized as follows:

$$\boldsymbol{E}^{pool} = \text{GMPool}_k(\boldsymbol{E}', \boldsymbol{M}^T), \quad \boldsymbol{M}^{pool} = \boldsymbol{M}\boldsymbol{C}, \tag{9}$$

where $\boldsymbol{C}$ is the cluster assignment matrix generated by GMPool. The overall architecture can be either global or hierarchical, depending on the downstream task.

### B.2   HyperDrop

Our drop-based edge pooling model, HyperDrop, consists of EHGNN layers and HyperDrop layers, which we describe below in detail. Before dropping unnecessary edges, we first update the edge features using the proposed EHGNN layer as follows:

$$\boldsymbol{E}' = \text{EHGNN}(\boldsymbol{X}, \boldsymbol{M}, \boldsymbol{E}). \tag{10}$$

Then, we drop the nodes of the dual hypergraph based on a learnable score function. While we can use any off-the-shelf node drop methods [12, 24] with their score functions, in this paper, we use the self-attention score based node drop method proposed in Lee et al. [24] as follows:

$$\boldsymbol{Z} = \tanh(\text{GNN}(\boldsymbol{E'}, \boldsymbol{M}^T, \boldsymbol{X})) \tag{11}$$

Based on the output score vector $\boldsymbol{Z} \in \mathbb{R}^m$ for every $m$ nodes on the dual hypergraph, we select the top-ranked $k$ nodes to obtain the pooled edge features and the incidence matrix as follows:

$$\boldsymbol{E}^{pool} = \boldsymbol{E'}_{idx}, \quad \boldsymbol{M}^{pool} = ((\boldsymbol{M}^T)_{idx})^T \; ; \; idx = \text{top}_k(\boldsymbol{Z}). \tag{12}$$

Thus, we obtain the edge-pooled graph $G^{pool} = (\boldsymbol{X}, \boldsymbol{M}^{pool}, \boldsymbol{E}^{pool})$ without loss of node information of the original graph. Furthermore, we use the self-attention score vector $\boldsymbol{Z}$ as the edge weight for the node-level message-passing layer, to reflect the relative importance of the neighboring information. This can be formulated as follows:

$$\boldsymbol{X'} = \text{GNN}\left(\boldsymbol{X}, \boldsymbol{M}^{pool}, \boldsymbol{Z}_{idx}\right), \tag{13}$$

where we can use simple GCN [23] or edge-aware GNNs for the GNN function.

## C   Experimental Setup

In this section, we introduce baselines and proposed models that we used for verifying the effectiveness of our approaches, in two different paragraphs: one for message-passing methods and another for graph pooling methods, and then provide the information of the computing resources. After that, we describe the experimental details about four different tasks on which we validate our methods.

**Baselines and our model for graph neural networks**     Here, we describe a set encoding model that ignores connectivity between nodes, naive graph neural networks that only consider node features without edge information, edge-aware graph neural networks that use edge features as auxiliary information for updating node features, and our model that explicitly represents edges as follows:

1. **DeepSet.** This method [44] is the set encoding baseline that first represents each node with a linear function, and then aggregates all node representations with sum pooling, which does not consider connectivity patterns between nodes.
2. **GCN.** This method [23] is the naive graph neural network baseline that aggregates neighboring nodes' information using the mean operation, which does not consider edge information. Also, we obtain the entire graph representation using the mean pooling of all nodes.
3. **GIN.** This method [41] is the naive graph neural network baseline that aggregates neighboring node's information using the sum operation, which does not consider edge information. Also, we obtain the entire graph representation using the sum pooling of all nodes.
4. **EGCN.** This method [16] is the edge-aware graph neural network baseline that uses edges as auxiliary information only to augment node-level representations, by adding the edge features between a node and its neighborhood to the node features (see Section A.1 for detailed formulation).
5. **MPNN.** This method [14] is the edge-aware graph neural network baseline that uses edges as auxiliary information only to augment the node-level representations, by multiplying the edge features between a node and its neighborhood to the node feature (see Section A.1 for details).
6. **R-GCN.** This method [33] is the edge-aware graph neural network baseline that uses discrete edge features for considering relation types between nodes, by multiplying the categorical weights of edges to the node features (see Section A.1 for detailed formulation).
7. **EGNN.** This method [15] is the edge-aware graph neural network baseline that first obtains explicit edge representations using differently designed edge-level layer, and then uses them to augment node-level representations, by multiplying the edge representations to the node representations (see Section A.1 for detailed formulation).
8. **EHGNN.** This is our edge representation learning framework that first transforms the given original graph into its dual hypergraph with *Dual Hypergraph Transformation*, and then obtain the explicit edge representations with existing off-the-shelf message-passing schemes for nodes, which is further directly used for graph-level representation learning.

**Baselines and our model for graph pooling** Here, we explain the global node pooling baselines, as well as the hierarchical node pooling baselines. Then, we describe the proposed two novel edge pooling schemes: cluster-based and drop-based methods, for graph-level representation learning.

1. **DiffPool.** This method [43] is the hierarchical node pooling baseline that coarsens nodes with a clustering-based approach, where it generates a cluster-assignment matrix for nodes using a GNN.
2. **SAGPool.** This method [24] is the hierarchical node pooling baseline that drops unnecessary nodes with a drop-based approach, where it generates scores for nodes with a GNN.
3. **TopKPool.** This method [12] is the hierarchical node pooling baseline that drops unnecessary nodes with a drop-based approach, where it generates scores for nodes with MLPs.
4. **MinCutPool.** This method [3] is the hierarchical node pooling baseline that coarsens nodes with a clustering-based approach, where it generates a cluster-assignment matrix for nodes using MLPs.
5. **ASAP.** This method [31] is the hierarchical node pooling baseline that first clusters similar nodes, then drop unnecessary clusters to coarsen an entire graph.
6. **EdgePool.** This method [6] is the hierarchical node pooling baseline that computes the edge score between nodes, then contracts two adjacent nodes with the high edge score into a single node.
7. **HaarPool.** This method [39] is the hierarchical node pooling baseline that coarsens nodes with the Haar transformation, which is based on the Haar basis in the Haar wavelet domain [38].
8. **SortPool.** This method [45] is the global node pooling baseline that first sorts the obtained node representations at the end of graph convolution layers, then predicts an entire graph representation with sorted node features.
9. **GMPool.** This method [1] is the global node pooling baseline that uses self-attention based operations to compress multiple nodes into a few clusters with learnable cluster assignment vectors to obtain an entire graph representation.
10. **GMT.** This method [1] is the global node pooling baseline that stacks self-attention based layers not only to compress many nodes into a few clusters with learnable cluster assignment vectors, but also to consider the inter-node (or cluster) relationships to obtain an entire graph representation.
11. **HyperCluster.** This is our global edge representation learning scheme that coarsens similar edges into a single edge to obtain a holistic edge-level representation, where we can generate the cluster assignment matrix for edges using existing clustering-based methods, such as GMPool [1] (see Section B.1 for more details).
12. **HyperDrop.** This is our hierarchical edge representation learning scheme that drops unnecessary edges based on a learnable score function, such as MLPs or GNNs, thereby adjusting the graph topology for more effective message-passing. Notably, this scheme does not result in the removal of any nodes. (see Section B.2 for more details).

**Computing resources** For all experiments, we use PyTorch [29] and PyTorch geometric [11], and train each model on a single Titan XP, GeForce GTX Titan X, or GeForce RTX 2080 Ti GPU. A single experiment of each task takes less than 1 day, and for the classification tasks such as node or graph classification, the single runtime on most datasets of a relatively small size is less than 1 hour.

### C.1 Graph reconstruction

**Common implementation details** Given a set of graphs $\{G = (\boldsymbol{X}, \boldsymbol{M}, \boldsymbol{E})\}$, the goal of graph reconstruction is to reconstruct both node and edge features from the compressed representations, by training two separate autoencoders where one is trained for reconstructing node features and the other is trained for reconstructing edge features. Formally, we define the node and edge encoders as $\mathrm{ENC}_{node}$ and $\mathrm{ENC}_{edge}$, respectively, and the node and edge decoders as $\mathrm{DEC}_{node}$ and $\mathrm{DEC}_{edge}$, respectively. Then, following the standard architecture setting of graph reconstruction tasks of existing works [3, 1], the node-level autoencoder which is a pair of the node encoder and node decoder, $\mathrm{ENC}_{node}$ and $\mathrm{DEC}_{node}$, is defined as follows:

$$\mathrm{ENC}_{node}(\boldsymbol{X}, \boldsymbol{M}, \boldsymbol{E}) = \mathrm{GMPool}(\mathrm{GNN}(\mathrm{GNN}(\boldsymbol{X}, \boldsymbol{M}, \boldsymbol{E}))) = \boldsymbol{X}^{pool}, \tag{14}$$

$$\mathrm{DEC}_{node}(\boldsymbol{X}^{pool}, \boldsymbol{M}, \boldsymbol{E}) = \mathrm{GNN}(\mathrm{GNN}(\mathrm{GNN}(\mathrm{GMPool}^{-1}(\boldsymbol{X}^{pool}, \boldsymbol{M}, \boldsymbol{E})))) = \boldsymbol{X}^{rec}, \tag{15}$$

where we use the GMPool [1] for reconstructing node features, as it shows outstanding performance on node-level reconstruction tasks. GMPool denotes the pooling operation, and $\mathrm{GMPool}^{-1}$ denotes the unpooling operation following the setting of the original paper [1]. Also, $\boldsymbol{X}^{rec} \in \mathbb{R}^{n \times d}$ is the

reconstructed node features from the pooled node representations $\boldsymbol{X}^{pool} \in \mathbb{R}^{k \times d}$, where $k$ is the number of pooled nodes and $n$ is the number of all nodes. We omit the inputs of the GNN, which are the incidence matrix $\boldsymbol{M}$ and the edge feature matrix $\boldsymbol{E}$, for simplicity.

However, to reconstruct the entire graph which have both node and edge features, we further need to define a separate edge-level autoencoder. Thus, similarly to the node-level autoencoder, we define the edge-level reconstruction module as a pair of the edge encoder and edge decoder, $\text{ENC}_{edge}$ and $\text{DEC}_{edge}$, formalized as follows:

$$\text{ENC}_{edge}(\boldsymbol{X}, \boldsymbol{M}, \boldsymbol{E}) = \text{Pool}(\text{GNN}(\text{GNN}(\boldsymbol{X}, \boldsymbol{M}, \boldsymbol{E}))) = \boldsymbol{E}^{pool}, \tag{16}$$

$$\text{DEC}_{edge}(\boldsymbol{X}, \boldsymbol{M}, \boldsymbol{E}^{pool}) = \text{GNN}(\text{GNN}(\text{GNN}(\text{Pool}^{-1}(\boldsymbol{X}, \boldsymbol{M}, \boldsymbol{E}^{pool})))) = \boldsymbol{E}^{rec}, \tag{17}$$

where, for our models, we use the EHGNN with the GCN [23] for GNN operations, and HyperCluster for pooling and unpooling operations which is described in Section B.1 in detail. Meanwhile, for the baselines, we use the existing edge-aware GNNs [17, 14, 33, 15] for GNN operations, and GMPool [1] for pooling and unpooling operations, where we obtain the final edge representation by averaging the two representations of incident nodes for the edge. This is because the baselines only use edge features as auxiliary information for updating node features. $\boldsymbol{E}^{rec} \in \mathbb{R}^{m \times d'}$ is the reconstructed edge features from the pooled edge representations $\boldsymbol{E}^{pool} \in \mathbb{R}^{k' \times d'}$, where $k'$ is the number of pooled edges and $m$ is the number of all edges. Similar to the formulation of the node-level autoencoder, we omit the inputs of the GNN for simplicity.

Our reconstruction objective is to minimize the discrepancy between the original graph $G = (\boldsymbol{X}, \boldsymbol{M}, \boldsymbol{E})$ and the reconstructed graph $G^{rec} = (\boldsymbol{X}^{rec}, \boldsymbol{M}, \boldsymbol{E}^{rec})$, with a loss function such as mean squared error or cross-entropy loss for node and edge features. For the edge reconstruction task, we only use the edge autoencoder without using the node autoencoder. For all reconstruction experiments, the learning rate of the node autoencoder is set to $5 \times 10^{-3}$, and the learning rate of the edge autoencoder is set to $1 \times 10^{-3}$. We optimize the full network using an Adam optimizer [22].

**Implementation details on synthetic graphs**  For the edge reconstruction of a synthetic graph, we use the standard two-moon graph generated by the PyGSP library [5], with node features given by their coordinates and edge features given by RGB colors of which values range from 0 to 1. Then, the goal of the edge reconstruction task is to restore all edge colors from the compressed edge representations after edge pooling. To minimize the discrepancy between original and reconstructed edge features, we use the mean squared error loss as the learning objective. Also, we use the early stopping criterion, where we stop the training if there is no further improvement on the training loss during 1,000 epochs, and the maximum number of epochs is set to 5,000. We set the pooling ratio of all models as 1% with the hidden dimension of size 16.

**Implementation details on molecular graphs**  Following the experimental setting of the existing work [7, 1], we use the subset of the full ZINC dataset [20], which consists of 12K molecular graphs, where node features are atom types and edge features are bond types. The number of atom types is 28, and the number of bond types is 5. We follow the dataset splitting of training, validation, and test sets from Dwivedi et al. [7]. Then, the goal of the molecular graph reconstruction task is to restore both atom types and bond types of all nodes and edges from their compressed representations after pooling. To train the model, we use the cross-entropy loss for molecular graph reconstruction, since the initial features given for nodes and edges are discrete. We also use the early stopping criterion, where we stop the training if there is no further improvement on the validation loss during 200 epochs. For hyperparameters, the maximum number of epochs is set to 500, hidden dimension size is set to 32, and batch size is set to 128. We run five experiments with different random seeds, and report the average performance with its standard deviation. Following the evaluation setup of Baek et al. [1], we use the following three metrics: *accuracy* measures the classification accuracy of all nodes and edges, *validity* counts the number of reconstructed molecules which are chemically valid, and *exact match* counts the number of reconstructed molecules which are identical to the original molecules.

**Implementation details on graph compression**  We quantitatively compare the relative memory size of the compressed graph after pooling nodes and edges against the size of the original graph, which we use the Erdos-Renyi random graph model [8]. We compare our proposed method EHGNN with HyperCluster, with the node pooling baseline, GMT. The number of nodes is fixed to $10^3$, while the number of edges is selected from one of $10^3$, $5 \times 10^3$, and $10^4$. To obtain the features of nodes and edges, we first randomly assign one of three values to each node (i.e., one among $\{0, 1, 2\}$), and

then generate edge features using the values of two adjacent nodes for each edge. For example, if two nodes have the same 0 value for the incident edge, then we assign the zero value to the edge feature. Since the total number of pairs of node values is six for the undirected graph, the number of edge features is six. The node pooling ratio is equally fixed to 15% for both GMT and our model, and we report the relative size of the entire graph with the edge reconstruction accuracy higher than 95% or 75%, where the edge pooling ratio is decided according to its accuracy.

### C.2 Graph generation

**Implementation details on MolGAN architectures** We use the QM9 dataset [30] that contains 133,885 organic compounds, where each molecular graph consists of carbon (C), oxygen (O), nitrogen (N), and fluorine (F) with up to nine non-hydrogen atoms. To evaluate the generated molecular graphs, we use the normalized Synthetic Accessibility (SA) and Druglikeness (QED) scores following the evaluation setup of the original paper [4]. Also, we use the categorical re-parameterization trick with the Gumbel-softmax function during the discretization process of molecule generation, to train the model in an end-to-end fashion, which adapts the learning scheme of the original paper [4].

In the original MolGAN [4], R-GCNs [33] are used to encode feature representations of nodes for the discriminator and reward networks. Learning rates of the generator, the discriminator, and the reward network are equally set to $1 \times 10^{-3}$, and hidden sizes of the two-layer R-GCNs are 128 and 64. For the MolGAN with GMPool (MolGAN + GMPool) setting, the GMPool, which is the global node pooling baseline, is additionally used to obtain the compact node-level representations. The tanh activation function is used for GMPool. For the MolGAN with the proposed EHGNN (MolGAN + EHGNN) setting, we use two EHGNN layers to encode the feature representations of the edges, wherein we use the GCN as the edge-level message-passing function. The hidden sizes are set to 32 and 16. After obtaining the edge-level representations, we use mean pooling to obtain the global edge representation, which is forwarded to the discriminator and reward networks. We further combine the GMPool with the MolGAN + EHGNN combination (MolGAN + GMPool + EHGNN) to additionally enhance the global graph representation with both node and edge representations. The learning rate of the EHGNN parameters in the discriminator and reward networks is set to $1 \times 10^{-2}$. Also, all the models use Adam optimizer [22] for training. Regarding other settings, we strictly follow the original MolGAN paper [4], and use the available code[1].

**Implementation details on MARS architectures** For the experiments using the MARS architecture, we use the ZINC15 [36, 18] dataset, which contains 2 million molecules, and we use the available data[2] from Hu et al. [18]. Further, we provide additional experimental results on the ChEMBL [13] dataset, which consists of 1,488,640 molecules, in Section D.2. As the fragments of molecular graphs are the basic building blocks for molecular graph generation in the MARS [40], we build the fragment vo-

Table 3: Statistics of fragment vocabularies of ZINC15 and ChEMBL datasets on MARS experiments.

|  | ZINC15 | ChEMBL |
|---|---|---|
| # of node types | 9 | 9 |
| Avg # of nodes | 7.68 | 7.35 |
| # of edge types | 4 | 4 |
| Avg # of edges | 7.54 | 7.08 |

cabularies following the same procedure of the original MARS paper: fragments are built by breaking a single bond of molecules from the given dataset, limiting the size of fragments to 10 atoms (see the original paper [40] for more details on the generation process of fragment vocabularies). We report the statistics of generated fragments from each dataset in Table 3.

The MARS model sequentially generates molecules by taking one of the addition or deletion actions at each step, especially where this model uses the explicit edge representation on the deletion actions. For a set of given graphs $\{G = (\boldsymbol{X}, \boldsymbol{M}, \boldsymbol{E})\}$, the original MARS model obtains the edge representation for the deletion actions as follows:

$$
\begin{aligned}
\boldsymbol{X}' &= \text{MPNN}(\boldsymbol{X}, \boldsymbol{M}, \boldsymbol{E}) \\
\boldsymbol{E}'_e &= \text{Concat}(\boldsymbol{X}'_u, \boldsymbol{X}'_v, \text{MLP}(\boldsymbol{E}_e))
\end{aligned}
\tag{18}
$$

where MPNN is the edge-aware graph neural network described in the subsection C, an edge $e$ is incident to two nodes $u$ and $v$, and $\boldsymbol{E}'_e$ is the output edge representation of the edge $e$. Compared to this baseline that implicitly captures the edge representation on the learned node representation $\boldsymbol{X}'$ with the concatenated edge representation through the naive MLP layer, for our model, we replace

---

[1]https://github.com/yongqyu/MolGAN-pytorch
[2]http://snap.stanford.edu/gnn-pretrain/data/

the MLP layer with the proposed EHGNN to explicitly learn the edge representation via edge-level message-passing. For a fair comparison in terms of the number of parameters, we use the same number of layers and embedding size for both MLP and EHGNN.

Following the experimental setup of the original MARS paper [40], we train the models to maximize the sum of multiple scores: QED, SA, and target protein inhibition scores against GSK3$\beta$ and JNK3, respectively. For evaluation metrics, we measure the percentage of the generated molecules having scores above a certain threshold for each property: QED $\geq 0.67$, SA $\geq 0.67$, and the inhibition scores against GSK3$\beta \geq 0.6$ and JNK3 $\geq 0.6$. The success rate can measure the overall multi-objective score by calculating the percentage of the generated molecules satisfying all four objectives. We also report the suggested easier threshold from the original MARS paper [40]: QED $\geq 0.6$, SA $\geq 0.67$, and the inhibition scores against GSK3$\beta \geq 0.5$ and JNK3 $\geq 0.5$, in Section D.2, where we see the same tendency for the results of baseline and our model. For the experiment on ZINC15, we set the learning rate of EHGNN parameters to $5 \times 10^{-3}$ with a cosine scheduler for learning rate warmup. For the experiment on ChEMBL, we set the learning rate of EHGNN parameters to $3 \times 10^{-4}$. The learning rate of other parameters in MARS is set to $3 \times 10^{-4}$, following the original paper [40]. We use the available code[4] from the original MARS paper.

### C.3 Graph classification

**Datasets** We validate our models on ten different benchmark datasets including six from the TU datasets [27] and four from the OGB datasets [16]. For a fair comparison of baselines and our model, following the standard experimental setting of Errica et al. [9], we use the one-hot encoding of atom types as initial node features in TU bio-chemical datasets (D&D, PROTEINS, MUTAG) and one-hot encoding of node degrees as initial node features in TU social datasets (IMDB-B, IMDB-M, COLLAB), if initial node features are not given in advance. Furthermore, if the initial edge features are not given in advance, we set them to one uniformly. For the dataset splitting of the TU datasets, we follow the standard training/test splits from Niepert et al. [28], Zhang et al. [45], Baek et al. [1], and further divide the training set into training and validation sets by using the 10 percent of the training data as validation data, as suggested by the fair comparison setup of Errica et al. [9]. For the OGB datasets (HIV, Tox21, ToxCast), following the original dataset paper [17], we use the additional atom and bond features for each graph, and follow the performance evaluation and data split setting of Hu et al. [17]. The statistics of each dataset are provided in Table 3 of the main paper.

**Implementation details** We follow the standard experiment setting from Baek et al. [1] with the same base architectures and hyperparameters for all models on all datasets[3]. Notably, we stack three number of GCN layers as node-level message-passing for all pooling models, including ours. For our model, we use the GCN for the EHGNN layer, where we equally stack three number of EHGNN layers to obtain the explicit edge representations, in parallel with node-level layers. Also, from the explicitly learned edge representations, we drop edges with their scores at each edge-level layer, which is described in section B.2 in detail. For the model HyperDrop + GMT, we apply the global node pooling layer GMPool [1] after the HyperDrop layers to obtain the global representation. For the hyperparameters of our HyperDrop, we set the hidden dimension of edges as 128 except the COLLAB dataset, on which we set the hidden dimension as 16, since the COLLAB dataset has a large number of edges compared to other datasets. Also, we randomly search for the edge drop ratio by increasing the drop ratios from $5\%$ to $75\%$ with $5\%$ increments. We report the average performances and standard deviations of 10 runs with different random seeds on test datasets.

### C.4 Node classification

To demonstrate HyperDrop's effectiveness in alleviating the over-smoothing problem in deep GNNs, we validate it on the semi-supervised node classification tasks.

**Datasets** We experiment on two benchmark datasets [34], namely Cora and Citeseer, which is the citation network where nodes are documents and edges are citation links between documents. The goal of the node classification task is to predict the class of the documents (nodes). The Cora dataset consists of 2,708 nodes and 5,429 edges with 7 classes. Also, the Citesser dataset consists of 3,327 nodes and 4,732 edges with 6 classes. Node features for each dataset consist of bag-of-words for

---

[3]https://github.com/JinheonBaek/GMT

Figure 1: **Additional edge reconstruction results with TopKPool** on the ZINC dataset by varying the compression ratio. Along with the results of Figure 3 in the main paper, we additionally report the average performance of the baselines using TopKPool over 5 different runs with the standard deviation.

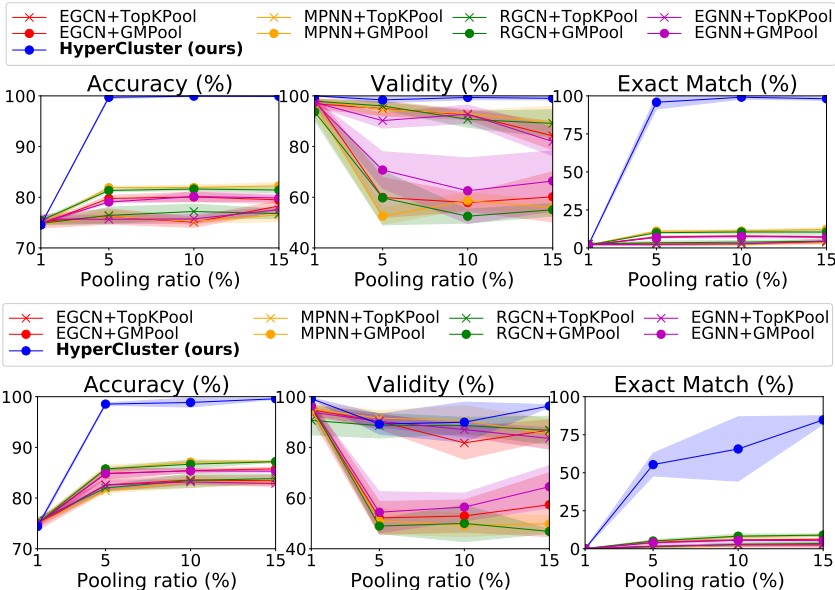

Figure 2: **Additional graph reconstruction results with TopKPool** on the ZINC dataset by varying the compression ratio. Along with the results of Figure 5 in the main paper, we additionally report the average performance of the baselines using TopKPool over 5 different runs with the standard deviation.

each document. As the initial edge features are not given, we set them by concatenating the features of two endpoints of the edge. We use the classification accuracy as an evaluation metric.

**Implementation details**    For a fair evaluation of the semi-supervised node classification task, we follow the standard experimental setting of existing works [23, 37, 10], from the node features to the dataset splitting. Regarding baselines, we use the naive GCN [23], GCN with batch normalization [19], and random edge drop scheme [32]. Specifically, for the GCN with batch normalization, we use the batch normalization layer between every GCN layer to normalize the features of nodes. Also, for the random edge drop baseline, we randomly drop the partial number of edges before the first layer of GNNs, following the setting of Rong et al. [32], where we do not use the batch normalization to directly see the effect of random drop on the over-smoothing problem. For our model, we use the HyperDrop with EHGNN (see section B.2 for detailed architectures), where we drop edges when passing through every four GNN layers starting from the second layer, and we do not use the batch normalization. Finally, we use the GCN as the node-level message-passing layers for all models, and also use it as the edge-level message-passing layers for our HyperDrop with EHGNN.

Following the hyperparameters of the existing semi-supervised node classification work [10], for the Cora dataset, we set the dropout rate as $0.5$, hidden size as $32$, and learning rate as $0.01$. Also, for the Citeseer dataset, we use the same setting from the Cora dataset except for the dropout rate which is set to $0.2$. For the random drop and our models, we drop $20\%$ of edges at each drop step.

## D    Additional Experimental Results

In this section, we provide the additional experimental results on graph reconstruction and generation tasks, with examples of reconstructed or generated molecules. Then, to further qualitatively evaluate the performances of our model, we visualize the edge pooling process of the proposed HyperDrop.

### D.1    Graph reconstruction

**Additional graph reconstruction results**    To see the effect of the pooling method on edge and graph reconstruction tasks, we additionally provide the performance of the TopKPool, a representative node drop method, with existing edge-aware GNN baselines as well as the performance of the GMPool, a node clustering method used in our main paper. For the comparison of the pooling methods,

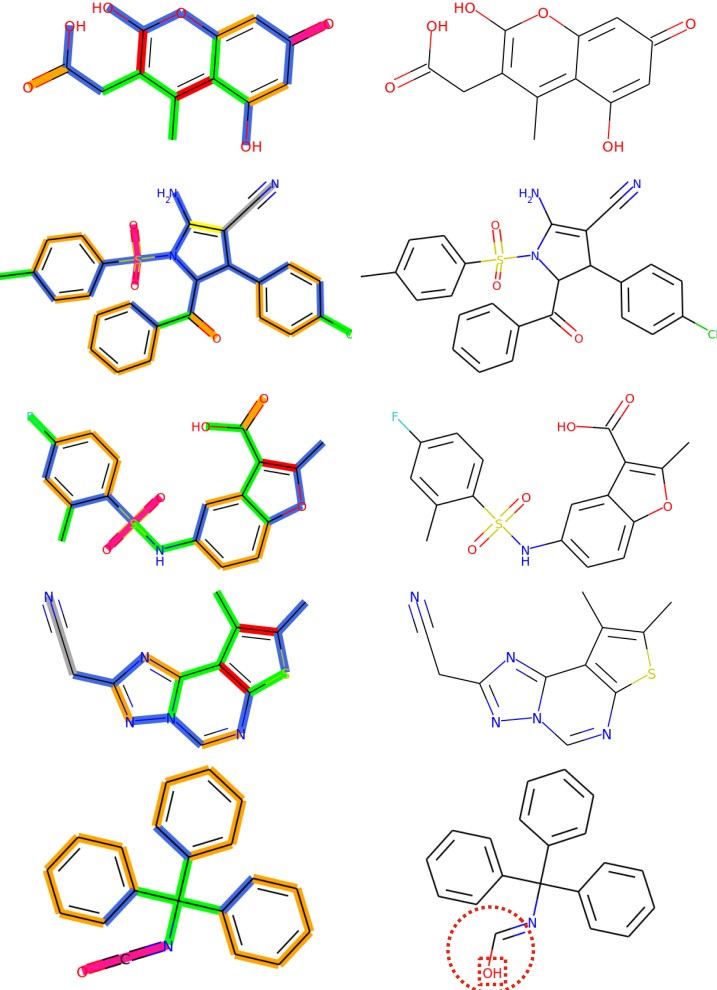

Figure 3: **Molecule reconstruction examples.** Molecules shown in the left column are the original molecules with an assigned cluster on each edge, where each cluster is represented as color. The clusters are generated by our method, HyperCluster. The molecules shown in the right column are the reconstructed molecules with our method, where red circles and squares indicate the incorrect prediction of edges and nodes, respectively.

we report the performances of both TopKPool and GMPool, in Figure 1 for edge reconstruction and in Figure 2 for graph reconstruction. As shown in Figure 1 and Figure 2, the proposed EHGNN with HyperCluster largely outperforms all the baselines, which suggests that accurately learning the edge representations is more important than choosing which pooling methods to use, in order to obtain the global graph-level representations. Moreover, we observe that the node drop method (TopKPool) for reconstruction is inferior to the node clustering method (GMPool) in terms of accuracy and exact match, since drop methods result in the removal of nodes and edges. The performance gain in validity with the TopKPool mostly comes from its reconstruction of a graph with a single bond, which makes them valid but far different from the desired reconstructed molecules.

**Additional examples of molecular graph reconstruction** We provide additional examples of reconstructed molecular graphs on the ZINC dataset in Figure 3. Molecules on the left side are the original molecules with each edge color indicating the assigned cluster, obtained by our HyperCluster. Molecules on the right side are the reconstructed molecules, where red circles and squares denote the incorrect predictions of edges and nodes, respectively. As shown in Figure 3, we can see that the clusters are meaningfully assigned with respect to the underlying substructures considering both edges and nodes. For example, edges in the hexagonal ring are assigned to orange and blue colors, where their color patterns are generally determined by the number of adjacent edges with their bond type. Moreover, triple bonds connected to the nitrogen (N) are assigned to the silver-colored cluster.

Figure 4: **Graph generation results on MolGAN.** Along with the results of Figure 8 in the main paper, we additionally report the performance of the combination of MolGAN, EHGNN, and GMPool. Solid lines denote the mean, and shaded areas denote the standard deviation of 3 different runs.

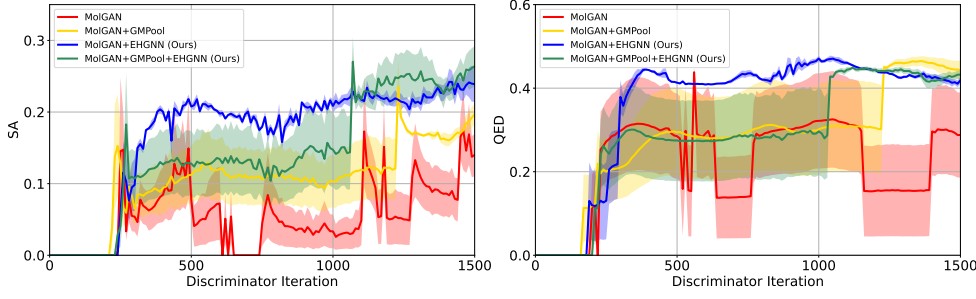

Table 4: **Graph generation results on MARS including all evaluation metrics.** The results are the mean and standard deviation of 3 runs.

| Datasets | Metrics | MARS | MARS + EHGNN (Ours) |
|---|---|---|---|
| ZINC15 | Success Rate | $59.53 \pm 2.11$ | $\textbf{64.30} \pm 1.54$ |
| | QED ($\geq 0.67$) | $95.71 \pm 0.09$ | $\textbf{96.36} \pm 0.49$ |
| | SA ($\geq 0.67$) | $\textbf{99.99} \pm 0.01$ | $\textbf{99.99} \pm 0.02$ |
| | GSK3$\beta$ ($\geq 0.6$) | $86.52 \pm 1.67$ | $\textbf{90.63} \pm 2.57$ |
| | JNK3 ($\geq 0.6$) | $71.52 \pm 4.15$ | $\textbf{73.60} \pm 1.29$ |
| ChEMBL | Success Rate | $56.64 \pm 5.79$ | $\textbf{58.25} \pm 6.07$ |
| | QED ($\geq 0.67$) | $91.01 \pm 2.79$ | $\textbf{91.13} \pm 4.84$ |
| | SA ($\geq 0.67$) | $99.99 \pm 0.01$ | $\textbf{100.00} \pm 0.00$ |
| | GSK3$\beta$ ($\geq 0.6$) | $87.45 \pm 1.73$ | $\textbf{90.34} \pm 2.65$ |
| | JNK3 ($\geq 0.6$) | $\textbf{70.57} \pm 4.75$ | $70.01 \pm 4.83$ |

Table 5: **Graph generation results on MARS under the setting of original success thresholds.** The results are the mean and standard deviation of 3 runs.

| Datasets | Metrics | MARS | MARS + EHGNN (Ours) |
|---|---|---|---|
| ZINC15 | Success Rate | $95.65 \pm 0.90$ | $\textbf{97.28} \pm 1.14$ |
| | QED ($\geq 0.6$) | $99.07 \pm 0.29$ | $\textbf{99.45} \pm 0.15$ |
| | SA ($\geq 0.67$) | $\textbf{99.99} \pm 0.01$ | $\textbf{99.99} \pm 0.02$ |
| | GSK3$\beta$ ($\geq 0.5$) | $99.13 \pm 0.12$ | $\textbf{99.52} \pm 0.23$ |
| | JNK3 ($\geq 0.5$) | $97.33 \pm 1.30$ | $\textbf{98.21} \pm 0.89$ |
| ChEMBL | Success Rate | $\textbf{92.03} \pm 3.83$ | $91.88 \pm 3.50$ |
| | QED ($\geq 0.6$) | $\textbf{96.76} \pm 1.44$ | $96.43 \pm 2.83$ |
| | SA ($\geq 0.67$) | $99.99 \pm 0.01$ | $\textbf{100.00} \pm 0.00$ |
| | GSK3$\beta$ ($\geq 0.5$) | $99.19 \pm 0.31$ | $\textbf{99.39} \pm 0.23$ |
| | JNK3 ($\geq 0.5$) | $95.83 \pm 2.30$ | $\textbf{95.85} \pm 0.92$ |

## D.2 Graph generation

**MolGAN** Since the EHGNN framework can be jointly used with the node-level representation learning methods, we can further combine the EHGNN framework with the node pooling method, for obtaining holistic graph-level representation from both node and edge representations. Thus, we additionally couple the MolGAN + EHGNN with the state-of-the-art node pooling method, namely GMPool. As shown in Figure 4, compared to the large performance gain obtained by our EHGNN, the performance gain obtained from using both GMPool and EHGNN is relatively small, and also the training using both architectures is unstable. This might be because, we can already obtain the effective graph-level representation only with the combination of MolGAN and EHGNN, and additionally using more layers makes the training of the MolGAN architecture difficult since this scheme also increases the number of parameters. On the other perspective, since the original MolGAN architecture is already able to utilize the node representations, albeit, by simple R-GCN, the remaining performance gain comes from the explicit edge representations via our EHGNN.

**MARS** Here, we provide the additional experimental results using the MARS architecture on the ChEMBL dataset, where we used the available data[4] from Xie et al. [40]. As shown in Table 4, MARS equipped with our EHGNN outperforms the baseline model, showing the same tendency as in the results on the ZINC15 dataset. Also, the original MARS and the MARS with EHGNN models successfully generate the high-quality molecules in terms of SA, and there is not much significant difference between those two models on this metric. However, the performance gain with our EHGNN against the naive MARS comes from other metrics, such as QED and GNK3$\beta$, resulting in the successful generation of molecules having all desired properties.

On the other hand, we also report the success rate with individual evaluation metrics according to thresholds used in the MARS paper [40] in Table 5. As shown in Table 5, our MARS + EHGNN model still outperforms the baseline on most of the metrics, and the performance tendency is highly similar to the result of different thresholds in Table 4. Those two results demonstrate that accurate learning of edge representation is important to generate desirable molecules.

---

[4]https://github.com/yutxie/mars

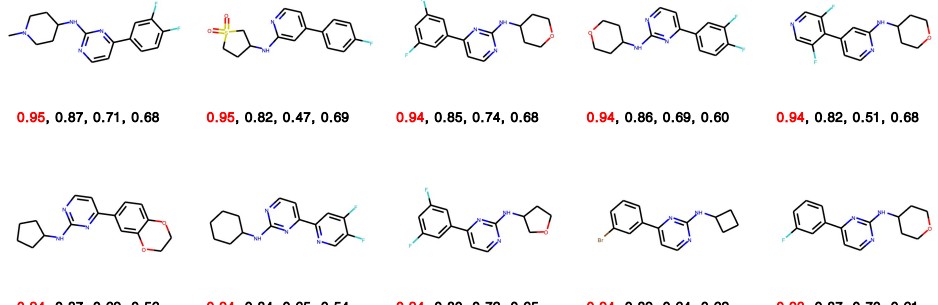

0.95, 0.87, 0.71, 0.68    0.95, 0.82, 0.47, 0.69    0.94, 0.85, 0.74, 0.68    0.94, 0.86, 0.69, 0.60    0.94, 0.82, 0.51, 0.68

0.94, 0.87, 0.69, 0.56    0.94, 0.84, 0.65, 0.54    0.94, 0.80, 0.72, 0.65    0.94, 0.89, 0.64, 0.62    0.93, 0.87, 0.70, 0.61

Figure 5: **10 generated molecules with the highest QED scores.** The numbers are QED, SA, GSK3$\beta$, and JNK3 scores, respectively. We highlight the QED score in red among four different scores.

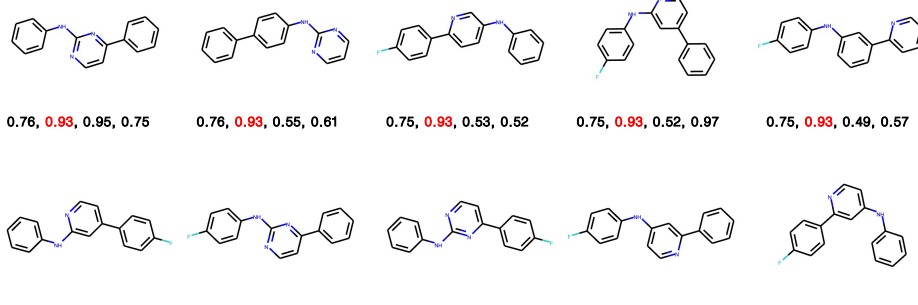

0.76, 0.93, 0.95, 0.75    0.76, 0.93, 0.55, 0.61    0.75, 0.93, 0.53, 0.52    0.75, 0.93, 0.52, 0.97    0.75, 0.93, 0.49, 0.57

0.75, 0.93, 0.56, 0.95    0.78, 0.93, 0.90, 0.83    0.78, 0.92, 0.92, 0.73    0.75, 0.92, 0.58, 0.66    0.75, 0.92, 0.63, 0.59

Figure 6: **10 generated molecules with the highest SA scores.** The numbers are QED, SA, GSK3$\beta$, and JNK3 scores, respectively. We highlight the SA score in red among four different scores.

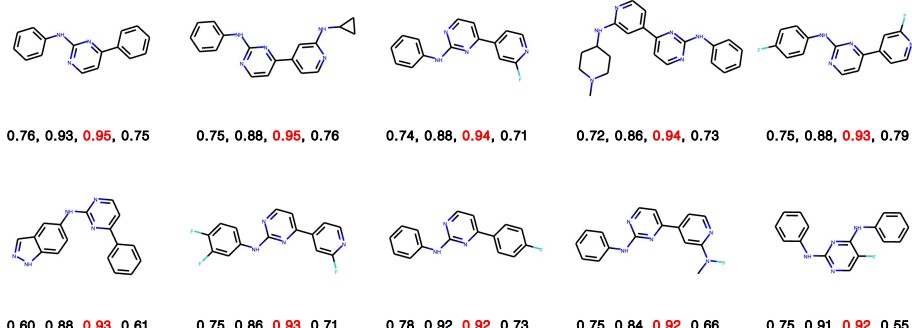

0.76, 0.93, 0.95, 0.75    0.75, 0.88, 0.95, 0.76    0.74, 0.88, 0.94, 0.71    0.72, 0.86, 0.94, 0.73    0.75, 0.88, 0.93, 0.79

0.60, 0.88, 0.93, 0.61    0.75, 0.86, 0.93, 0.71    0.78, 0.92, 0.92, 0.73    0.75, 0.84, 0.92, 0.66    0.75, 0.91, 0.92, 0.55

Figure 7: **10 generated molecules with the highest GSK3$\beta$ scores.** The numbers are QED, SA, GSK3$\beta$, and JNK3 scores, respectively. We highlight the GSK3$\beta$ score in red among four different scores.

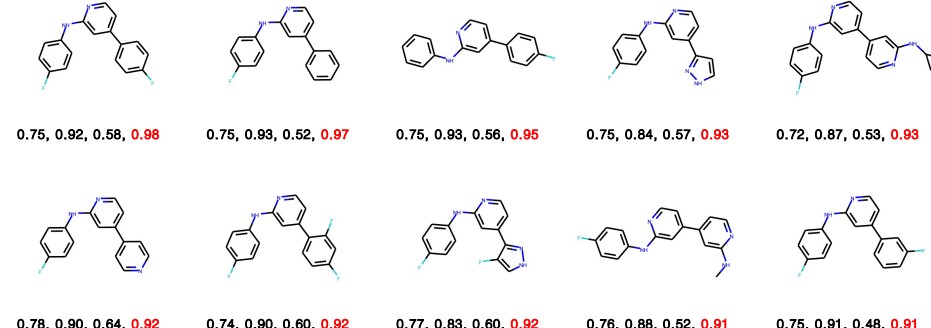

0.75, 0.92, 0.58, 0.98    0.75, 0.93, 0.52, 0.97    0.75, 0.93, 0.56, 0.95    0.75, 0.84, 0.57, 0.93    0.72, 0.87, 0.53, 0.93

0.78, 0.90, 0.64, 0.92    0.74, 0.90, 0.60, 0.92    0.77, 0.83, 0.60, 0.92    0.76, 0.88, 0.52, 0.91    0.75, 0.91, 0.48, 0.91

Figure 8: **10 generated molecules with the highest JNK3 scores.** The numbers are QED, SA, GSK3$\beta$, and JNK3 scores, respectively. We highlight the JNK3 score in red among four different scores.

**Visualization of the generated molecular graphs**    We further provide the examples of generated molecules using our EHGNN on MARS in Figure 5, 6, 7, and 8. We hope that these examples are to be helpful for the chemists to get an insight into the molecules generated with our framework.

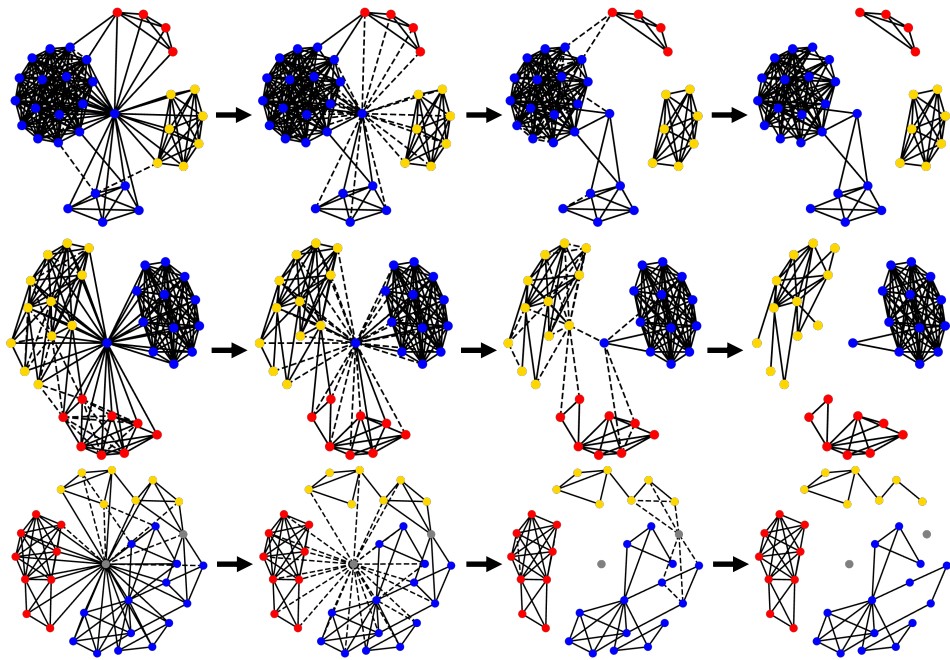

Figure 9: **Edge pooling results on the COLLAB dataset.** Each row represents the pooling process of a graph. Colors denote connected components.

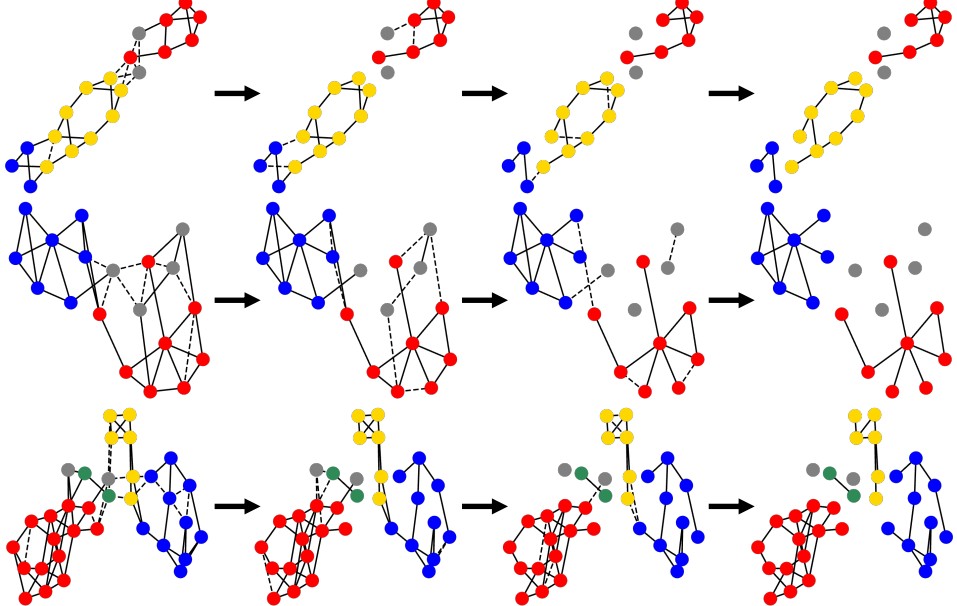

Figure 10: **Edge pooling results on the PROTEINS dataset.** Each row represents the pooling process of a graph. Colors denote connected components.

## D.3 Graph classification

**Additional examples of HyperDrop process** We provide additional examples of HyperDrop processes on the COLLAB and PROTEINS datasets in Figure 9 and Figure 10, respectively. Colors represent the resulting connected components in the final graph after dropping edges, and we represent isolated nodes as gray. Arrows indicate the layer-wise progressive pooling processes. We can see that by dropping unnecessary edges, a large graph is divided into smaller connected components, which we assume to be effective for message-passing between the relevant nodes.

# E  Limitations and Potential Societal Impacts

In this section, we discuss the limitations and potential societal impacts of our work.

**Limitations**   In this work, we propose to learn edge representations with hypergraphs, using the dual hypergraph transformation that allows us to apply off-the-shelf node-level message-passing schemes designed for node representation learning to edges. While we can learn accurate edge representations using the proposed framework, we need two separate GNNs to learn node and edge representations independently. Combining these two GNNs into one, by learning node and edge representations jointly using a single GNN, may be more effective for learning graph representations, while saving the memory as well. We leave this as future work.

**Potential societal impacts**   The system for generating target molecules is significantly important to our society, since it can be used to generate vaccines or drugs for diseases, even for the newly emerged severe acute respiratory syndrome coronavirus 2 (SARS-CoV-2). However, the conventional development of beneficial molecules requires a huge amount of time and resources with a significant number of trial-and-error processes, before actually applying the generated molecules, since we have to check potential outcomes those molecules can have.

In this paper, we show that the proposed edge representation learning framework can accurately represent the edges of the given graph, for the holistic graph-level representation learning, which has been extensively validated on graph generation and classification tasks with biochemical molecules. Therefore, this approach can meaningfully aid the development of target molecules in the following ways. First, the generation system described in Section 4.2 of the main paper is effective for generating molecules with desirable properties, since it can generate more drug-like molecules that can effectively inhibit multiple target proteins. Also, the classification system described in Section 4.3 of the main paper is beneficial for examining the toxicity of generated molecules, which is an essential step before human clinical trials or being deployed on a commercial scale. Therefore, our method allows us to reduce time and resources for generating and validating target molecules, for example in the domain of de novo drug design compared to synthesizing drugs by trial-and-error.

As described above, while our method has huge potential impacts for discovering novel molecules in our real-life, anyone can maliciously use our system, aiming to develop harmful compounds for humans, such as synthesizing toxic or addictive substances. Thus, we strongly hope that our method would not be applied for generating harmful molecules that may have negative impacts on our society.