# OpenReview forum: "Edge Representation Learning with Hypergraphs"
_NeurIPS.cc/2021/Conference — NeurIPS 2021 Poster_

### Official Review · Reviewer_5K8S · 2021-06-25

**Rating:** 6
**Confidence:** 4

**Summary:**

The paper proposes a new approach for computing edge-level representations. The core idea of the paper is to realize message passing not just in the original graph (the primal) that is provided in input to the model, but on the dual hypergraph (where nodes are edges and hyperedges are nodes in the primal) where additional processing such as clustering or dropping can be applied in order to respectively compute graph-wise edge representations or improve the quality of message passing on the primal as irrelevant edges are hopefully dropped on the dual. The authors validate the quality of the model on graph/edge reconstruction tasks on synthetic and molecular datasets (ZINC12), graph generation and graph classification tasks.

**Limitations And Societal Impact:**

Yes.

**Main Review:**

Overall I believe the paper is on the acceptance side despite there are a number of issues I would like the authors to clarify in the rebuttal.

First of all, the authors refer to the dual of the original graph (the primal) as an hypergraph (as multiple nodes are connected by same edges in the dual), however they don’t exploit at all the hypergraph nature of the dual and realize message passing on it as if it was a simple graph. This idea is not novel and was already exploited in https://arxiv.org/abs/1806.00770 for computing edge level representations that can be used for refining an attention mechanism on the primal. Additionally, the complexity of processing the dual as described in the paper - Table 1 seems to be wrongly stated. The authors mention that alternative methods for constructing the adjacency matrix of the dual require O(m^2) operations (with m = number of edges), however because of the sparse nature of the adjacency matrix (if I’m not mistaken) this can actually be realized resorting to edge lists and dictionaries (one mapping edges to incident nodes and one mapping nodes to incident edges) with a complexity equal to O(m*d_max) (with d_max the maximum node degree), which is significantly cheaper and actually the same complexity required by the method of the authors for processing the dual. In this sense, the authors mention in the paper that they can just transpose the incidence matrix of the primal for obtaining the incidence matrix of the dual hypergraph, but they don’t provide any detail on how this can avoid the construct of an edge list in the dual which I believe is still required for realizing message passing therein. I would be happy if the authors could clarify this detail in their rebuttal.

While the idea of processing the dual graph is not novel per se, to the best of my knowledge using message passing on the dual to identify not so relevant or noisy edges in the primal (and thus realize some sort of learnable rewiring operation) is a new contribution to the field (and personally what I found to be the most interesting bit of the paper). However, the justification for why one would want to use this seems to be rather weakly justified in the paper. The authors in particular stress that for graph classification tasks, dropping edges (and not nodes) allows to compute lossless graph-wise representations as all nodes are preserved in the final aggregation process while not so relevant edges are removed. However, one could argue the same for the edges of the dual graph and that, as some edges might be noisy in the primal, some edges might be noisy in the dual which would correspond to dropping nodes in the original graph (in this case loosing some information in the node-wise final aggregation process would actually be beneficial for graph classification problems). Hence, while I believe there is some merit in the idea of pruning the original graph, learning to discard some edges if needed, the justification presented in the paper doesn’t seem to be well argued and I would recommend the authors to rephrase this in terms of rewiring the original graph. I would like the authors to additionally reply on this in their rebuttal as well.

Similarly to the above, the idea of coupling a graph-wise representation based on node features with one explicitly aimed at representing edges I believe is a new (despite not particularly original) contribution that seems to favour graph generation and reconstruction tasks in particular.

On the experimental side, I believe the experiments are overall good and extensive, however I think there might be room for improvement. In particular, for the edge and graph reconstruction tasks, the authors compare with previous approaches that only implicitly use edge features for refining node representations. However, while the importance of computing edge representations seems clear from the experiment, I believe comparing with https://dl.acm.org/doi/abs/10.1145/3394486.3403389 / https://www.ijcai.org/Proceedings/2019/0366.pdf  that explicitly compute edge representations in the message passing procedure coupled with a suitable pooling (e.g. even a simple mean/sum/max on the added colored nodes representing the edges) would have greatly improved the reader understanding of the benefits of processing the dual with respect to an alternative simple solution.

Other comments on the experiments I would like to clarify with the authors:

1. the authors propose just a qualitative analysis on their synthetic dataset for edge reconstruction, I believe showing some quantitative results on this as well would improve the strength of the experimental section.

2. The graph compression experiment in the paper is confusing. What do the authors mean with the relative size of the full graph in Figure 7?

In conclusion, while the presentation of the core ideas of the paper could be improved and there are a series of items that I would like the authors to address in the final manuscript, I think the manuscript does present some contribution to the field and I’m thus (even if weakly) in favor of its acceptance provided that the authors clarify the issues I highlighted.

Other Minor comments:


1) line 92/94, this seems an over statement. As mentioned above, potentially we can simply compute edge-level features by adding for each edge e, a new colored node which is connected to the end points of e and run message passing on this expanded graph. I would rephrase softening the claim.
2) Figure 3 / 5, compression ratio is never defined in the paper, what do the authors mean with this?
3) Figure 5, why did the authors show the reconstruction performance of nodes and edges together when they already showed edge reconstruction performance in Figure 3?

I believe it would be clearer if the authors were showing the reconstruction performance of nodes and edges separately to understand the performance of the model  (e.g. in term of accuracy in figure 5 we see an improvement of the baselines wrt figure 3, while the proposed model seem to worsen, is the node-wise accuracy of the baselines at a node level better than the one of the proposed solution? If yes, why?)

4) Citation to https://arxiv.org/abs/1907.10903 is missing, despite the authors compare with it to showing the robustness of the model wrt oversmoothing.
5) What kind of pooling do the authors use for Hyperdrop without GMT in table 3? I was not able to find that detail anywhere in the paper nor supplementary material. Are the authors using HyperCluster as well in that experiment?


**Time Spent Reviewing:**

6

---

> ### Author Response · Authors · 2021-08-10
> **Initial Response (3/3) to Reviewer 5K8S**
>
> **Question 6:** The authors propose just a qualitative analysis on their synthetic dataset for edge reconstruction, I believe showing some quantitative results on this as well would improve the strength of the experimental section.
>
> **Answer:** Thank you for your suggestions, and we report the quantitative results of the edge reconstruction task (Figure 4; reconstruction of synthetic two-moon graphs) in the table below. As shown in the table, our HyperCluster significantly outperforms baselines on quantitative evaluations as well. We will include this result in the subsequent revision.
>
> | Model | Mean Squared Error (MSE) Loss |
> | --- | --- |
> | EGCN + GMPool | 0.1068 |
> | MPNN + GMPool | 302.8307 |
> | R-GCN + GMPool | 0.0932 |
> | EGNN + GMPool | 0.1067 |
> | **HyperCluster (Ours)** | **0.0525** |
>
> ---
> **Question 7:** The graph compression experiment in the paper is confusing. What do the authors mean with the relative size of the full graph in Figure 7?
>
> **Answer:** We apologize for the confusion. The relative size in Figure 7 corresponds to the relative memory size of the compressed graphs against the size for representing the entire graph, where the memory size is measured by the size of node features, edge features, and connectivity patterns. The compressed graph denotes the resulting graph after pooling node and edge features. We will add detailed explanations of the relative size in the subsequent revision.
>
> Figure 7 exhibits the effectiveness of our method on the task of efficient and accurate graph compression. It shows that, as the number of edges in the graph increases, compressing only the node features is not memory efficient, while our method effectively compresses the entire graph features with sufficiently higher reconstruction accuracy (95%) due to the compression of both node and edge features.
>
> ---
> **Question 8:** line 92/94, this seems an over statement. As mentioned above, potentially we can simply compute edge-level features by adding for each edge e, a new colored node which is connected to the end points of e and run message passing on this expanded graph.
>
> **Answer:** In lines 92-94, we pointed out the limitation of the previous edge-aware GNNs (Gilmer et al., 2017; Gong & Cheng, 2019; Yang & Li, 2020), not the work for line graphs nor the edge converting scheme you mentioned. However, we agree that the lines 92-94 might mislead the reader to understand that all previous work cannot represent edges with node-level message passing, thus we will soften the claim.
>
> ---
> **Question 9:** Figure 3 / 5, compression ratio is never defined in the paper, what do the authors mean with this?
>
> **Answer:** Compression ratio denotes the **ratio of the number of nodes (edges) in the compressed graphs to the number of nodes (edges) in the original graph**, which is equal to the pooling ratio. Intuitively, a compression ratio close to 0 means the number of edges after pooling is small. We will clarify this definition in the subsequent revision.
>
> ---
> **Question 10:** Figure 5, why did the authors show the reconstruction performance of nodes and edges together when they already showed edge reconstruction performance in Figure 3? I believe it would be clearer if the authors were showing the reconstruction performance of nodes and edges separately to understand the performance of the model.
>
> **Answer:** We aim to verify the effectiveness on **learning of graph-level representations including both nodes and edges**, not solely on nodes as existing works nor on edges as ours in Figure 3. Thus, by reconstructing both the nodes and edges in Figure 5, we show that representing only the nodes is deficient for accurately learning an entire graph.
>
> From the experimental point of view,
> * Figure 3 focuses on the representation power of our methods on edges, whereas
> * Figure 5 points out that explicit learning of both nodes and edges with their message passing is substantially effective in holistic graph-level learning.
>
> ---
> **Question 11:** In terms of accuracy in figure 5, we see an improvement of the baselines wrt figure 3, while the proposed model seems to worsen. Is the node-wise accuracy of the baselines at a node level better than the one of the proposed solution? If yes, why?
>
> **Answer:** This is a misunderstanding of experimental settings and metrics in Figure 3 and Figure 5. For the edge reconstruction (Figure 3), we only reconstruct the edge features and evaluate the performance on edges, whereas for the graph reconstruction (Figure 5), we reconstruct both node and edge features and evaluate both.
>
> Therefore, the performance drop in our method in Figure 5 comes from the lower accuracy on node reconstruction compared to our edge reconstruction, while the accuracy improvement in baselines comes from the higher accuracy on node reconstruction compared to their edge reconstruction.
>
> In particular, the node-wise accuracy of the baselines should be similar to us, as we use the same architecture of node reconstruction for all models, which is described in lines 275-276 of our main paper.
>
> ---
> **Question 12:** Citation to Rong et al. (2020) is missing, despite the authors compare with it to showing the robustness of the model wrt oversmoothing.
>
> **Answer:** Thank you for pointing this out. We will include it (Rong et al., 2020) in the subsequent revision.
>
> ---
> **Question 13:** What kind of pooling do the authors use for Hyperdrop without GMT in table 3? Are the authors using HyperCluster as well in that experiment?
>
> **Answer:** Thank you for pointing out the missing details of HyperDrop (without GMT) on the node pooling scheme. We use the simple sum/mean/max over node representations to obtain the one graph representation, following the widely used global pooling scheme (Cangea et al., 2018; Lee et al., 2019).
>
> In contrast to this, HyperDrop with GMT denotes the model that uses GMT over node representations instead of sum/mean/max, showing that HyperDrop can be orthogonally used with existing node pooling schemes. We will clarify this in the subsequent revision.
>
> We also compared the result on HyperCluster in the ablation study (Table 4), and we observed that HyperDrop is more appropriate on the graph classification tasks, which is conceptually explained in lines 58-60 of our paper.
>
> ---
> **References**
> * Gilmer et al. Neural message passing for quantum chemistry, ICML 2017.
> * Gong & Cheng, Exploiting edge features for graph neural networks, CVPR 2019.
> * Yang & Li, NENN: incorporate node and edge features in graph neural networks, ACML 2020.
> * Rong et al. DropEdge: Towards Deep Graph Convolutional Networks on Node Classification. ICLR 2020.
> * Cangea et al. Towards Sparse Hierarchical Graph Classifiers. R2L Workshop, NIPS 2018.
> * Lee et al. Self-Attention Graph Pooling. ICML 2019.

---

> ### Author Response · Authors · 2021-08-10
> **Initial Response (2/3) to Reviewer 5K8S**
>
> **Question 3-1:** The complexity of processing the dual as described in the paper - Table 1 seems to be wrongly stated. The authors mention that alternative methods for constructing the adjacency matrix of the dual require $O(m^2)$ operations (with $m$ = number of edges), however, this could be cheaper because of the sparse nature of the adjacency matrix.
>
> **Answer:** The complexities in Table 1 are correct, as they are computed based on the dense matrix, such as the adjacency matrix for baselines and the incident matrix for ours, not on the sparse matrix. This is described in lines 149-150 (“the transformation of the node adjacency to the edge adjacency”), Section A.2 of the supplementary file which contains further analytical details, and in the caption of Table 1. However, we will further clarify this in the revision.
>
> In the case when an edge list -- a sparse implementation of the connectivity between nodes -- is given, using our transformation is still more efficient. This is because, the transformation cost from the edge list to the hyperedge list (i.e., the edge list of the dual hypergraph) is negligible as we simply apply the **reshaping operation**to the edge list (see Section A.3 of the supplementary file for details), while the cost of transforming an edge list to the edge adjacency is $O(m \cdot d_{\text{max}})$ as you mentioned. Thus, when the number of edges $m$ or $d_{\text{max}}$ in the original graph increases, our method using the dual hypergraph become **even more efficient**than the methods using the line graph, which we experimentally verified by measuring the transformation cost of a random graph with varying number of edge as in the table in the answer to the above question.
>
> ---
> **Question 3-2:** The authors mention in the paper that they can just transpose the incidence matrix of the primal for obtaining the incidence matrix of the dual hypergraph, but they don’t provide any detail on how this can avoid the construct of an edge list in the dual which I believe is still required for realizing message passing therein.
>
> **Answer:** This seems like a misunderstanding between dense and sparse implementations of the connectivity patterns of the graph. If the given graphs are densely represented with the incidence matrix, we need $O(m)$ costs for acquiring the neighboring node set from the incidence matrix to realize the message-passing, which is described in lines 198-201 of the main paper. However, as described in line 188 of the main paper, the **Dual Hypergraph Transformation can be implemented with sparse matrices**, and furthermore, the datasets used in the graph-related tasks were mostly provided in the form of edge lists. Thus, we implemented the DHT with the **sparse hyperedge list**, and we perform message passing over this sparse implementation. As described in the question above, the cost for sparse DHT is negligible, and the details of constructing hyperedge lists are fully explained in Section A.3 of the supplementary file.
>
> ---
> **Question 4:** Using message passing on the dual to identify not so relevant or noisy edges in the primal (and thus realize some sort of learnable rewiring operation) is a new contribution to the field. However, the justification for why one would want to use this seems to be rather weakly justified in the paper. One can further consider identifying noisy edges in the dual, which is the nodes in the primal.
>
> **Answer:** Thank you for thoughtfully acknowledging our contribution with the term “learnable rewiring operation”. However, we have already justified the necessity of pruning edges instead of nodes with the high performance gain of our HyperDrop against node drop methods on graph classification tasks in Table 3, and also with the application of HyperDrop on node classification in Figure 10.
>
> Specifically, we found that such removal of nodes of the original graph substantially underperforms our HyperDrop, as shown in the poor performance of the node drop baselines (e.g., TopKPool and SAGPool). Therefore, losing the node information in the node-wise final aggregation is not desirable, and the previous work (Errica et al., 2020) supports the observation that node information is more dominant than edge information for graph classification.
>
> In addition, our HyperDrop has an additional advantage compared to node-based pooling methods, that we can apply HyperDrop on the node classification task as explained in lines 250-251 of our paper.
>
> As you thoughtfully recommended, we agree that the term “graph rewiring” may help readers to understand the mechanism of our method HyperDrop better, and thus will further strengthen it with the above justifications along with your suggestions in the subsequent revision.
>
> ---
> **Question 5:** On the experimental side, I believe the experiments are overall good and extensive, however I believe comparing with Yi & Park. (2020); Jiang et al. (2019) that explicitly compute edge representations in the message passing procedure coupled with a suitable pooling (e.g. even a simple mean/sum/max on the added colored nodes representing the edges) would have greatly improved the reader understanding of the benefits of processing the dual with respect to an alternative simple solution.
>
> **Answer:** We thank you for your suggestion of the related works on hypergraph representation learning. However, there seems to be a misunderstanding of the contribution of our work, since the suggested works are orthogonal to ours, and a direct comparison is not possible. The suggested two works (Yi & Park, 2020; Jiang et al., 2019) propose general learning schemes on hypergraphs, and does not address edge representation learning nor dual hypergraphs. Please note that the general hypergraph representation learning is not our main focus, but the edge representation learning is. Moreover, we have already compared with the baselines using similar methods for representing the edges in Figure 3 and 5. We further discuss the specific differences between our work and the mentioned works below:
>
> * The first work (Yi & Park, 2020) aims to represent the time-series network of the hypergraph structure, by representing the given hypergraph with recurrent GNNs over it, to model the high-order interactions among temporal nodes with their time dependency. However, this is completely different from ours, except that both use hypergraphs. First, Yi & Park (2020) do not consider the hypergraph duality for transforming edges to nodes, but we propose to represent edges of the original graph as the nodes of the dual hypergraph. Also, they use node features to represent edges, however we found such schemes are suboptimal for representing entire graphs as we showed them in our substantial experiments (Figure 3, 4, 5, 6), thus we represent edges with their message passing schemes.
>
> * Jiang et al. (2019) target the construction of a dynamic hypergraph for learning high-order interactions among vertices of the given graph, where they use the k-NN and k-means clustering schemes to construct the hypergraph. Also, they further apply convolution on both nodes and hyperedges of the constructed graph, on which the vertex-level convolution first aggregates vertex features to hyperedges, and then the hyperedge-level convolution aggregates hyperedge features to the vertex again, using simple MLPs. However, the work of Jiang et al. (2019) also does not learn edge representations with edge-level message passing and solely resorts to nodes, which is clearly orthogonal to our objective of learning edge representations via the hypergraph duality.
>
> * Furthermore, as you pointed out, the suggested previous works (Yi & Park, 2020; Jiang et al., 2019) use the node features to make the edge features with simple mean/sum/max over representations of nodes, which we already compared in our experiments (baselines such as EGCN + GMPool in Figure 3 and Figure 5). Specifically, we obtain the final edge representation of baselines by averaging two representations of an edge’s incident nodes (see Section C.1 of the supplementary file; lines 208-212), and we found such schemes are highly deficient on learning edge representations, compared to our edge-level message passing over edges.
>
> We will include the discussion of the two papers you suggested in the related work section of the revision.
>
> ---
> **References**
> * Errica et al. A fair comparison of graph neural networks for graph classification. ICLR 2020.
> * Yi & Park. Hypergraph Convolutional Recurrent Neural Network. KDD 2020.
> * Jiang et al. Dynamic Hypergraph Neural Networks. IJCAI 2019.

---

> ### Author Response · Authors · 2021-08-10
> **Initial Response (1/3) to Reviewer 5K8S**
>
> We sincerely thank you for your constructive and helpful comments. We deeply appreciate your comments that the methods of identifying unnecessary edges and of representing graphs with learning nodes and edges are novel contributions, and the experiments are good and extensive. We address all your concerns in below:
>
> ---
> **Question 1:** The authors refer to the dual of the original graph (the primal) as a hypergraph (as multiple nodes are connected by the same edges in the dual), however they don’t exploit at all the hypergraph nature of the dual and realize the message passing on it as if it was a simple graph.
>
> **Answer:** We did exploit the hypergraph nature of the dual hypergraph in the transformation and its implementation.
> * Specifically, we utilize the fact that the incidence matrix of the dual hypergraph is equal to the transpose of the incidence matrix of the original graph, which yields no cost for obtaining node and hyperedge features of the dual hypergraph from its original graph.
> * Moreover, the sparse matrix of our Dual Hypergraph Transformation (DHT) is implemented with the simple tensor reshape operations, by exploiting the property that the dual hypergraph is 2-regular, as described in Section A.3 of the supplementary file.
>
> ---
> **Question 2:** This idea is not novel and was already exploited in https://arxiv.org/abs/1806.00770 (Monti et al., 2018) for computing edge level representations that can be used for refining an attention mechanism on the primal.
>
> **Answer:** This is a factual misunderstanding since our novelty is in utilizing dual hypergraph for edge representation learning, and Monti et al. (2018) does not use a hypergraph. The edge to node transformation in our work is done in a completely different manner from Monti et al. (2018). Monti et al. (2018) utilize a line graph (i.e., dual graph) for edge representation learning, which is a **conventional graph (not a hypergraph)** where each node corresponds to an edge of the original graph, while we utilize the **dual hypergraph** for a given graph. This is the main novelty of our work which differentiates it from existing edge representation learning methods. Further, the approach by Monti et al. (2018) is highly limited since the line graph construction 1) is not injective, 2) results in the loss of node information during transformation, and 3) has $O(m \times d_{\text{max}})$ complexity for sparse implementation and $O(m^2)$ for dense implementation (see Table 1), where $m$ is the number of edges and $d_{\text{max}}$ is the maximum number of degrees of the node in the original graph. We list the **main differences between our Dual Hypergraph Transformation (DHT) and the line graph construction** in Monti et al. (2018) below:
>
> 1) Our DHT is **bijective** as shown in lines 180-184, and in Figure 2 in our main paper. However the line graph transformation from Monti et al. (2018) is **not injective**. This means that two different (non-isomorphic) graphs may be transformed into the same line graph, in which case a graph could be reconstructed into another graph, or two different graphs are considered as isomorphic.
>
> 2) Our DHT **does not incur any loss of node information**. However, with line graphs, since the nodes of the original graph do not correspond one-to-one to the edges, **the node information may be lost**during the transformation. This is a fatal flaw for tasks such as graph reconstruction and classification. To tackle this problem, Monti et al. (2018) store both the original graph and the line graph, but this greatly increases the memory complexity.
>
> 3) While our DHT can be done with **almost no cost** by applying simple transpose operations to the incident matrices, the line graph transformation is **not scalable**. First, the line graph construction itself requires at least $O(m)$ complexity where $m$ is the number of edges of the original graph. We further show the inefficiency of line graph construction, with the experimental results in the table below. Also, the constructed line graphs may have excessively large edges if the original graphs have very large degrees (e.g., scale-free graphs and community graphs). Specifically, the number of edges of the line graph has the size of $O(d^2_{\text{max}})$ where $d_{\text{max}}$ is the maximum degree of the original graph.
>
> Further, in contrast to Monti et al. (2018), our work focuses on **edge representation learning for general graph-level tasks**, including graph classification, graph reconstruction, and graph generation, which none of the existing methods tackle. Moreover, the method of Monti et al. (2018) cannot find **task-relevant edges** or **cluster similar edges into one** to obtain more meaningful edge representations, while ours achieve this goal with HyperDrop and HyperCluster.
>
> In the following table, we compare the transformation cost of our DHT (in seconds) with the line graph transformation, on graphs with 1,000 nodes with varying the number of edges, generated from the Erdos-Renyi graph. The results are the average of over 100 different runs.
>
> | $\text{num of edges} \over \text{num of nodes}$ of the original graph | 2 | 4 | 8 | 16 | 32 | 64 |
> | --- | --- | --- | --- | --- | --- | --- |
> | Line graph | 32.78 | 65.93 | 131.36 | 260.92 | 527.44 | 1071.31 |
> | Dual Hypergraph (Ours) | **0.13** | **0.18** | **0.26** | **0.45** | **0.81** | **1.37** |
>
> We believe that the above comparison against existing line-graph transformation for edge representation learning will be a valuable addition to help understanding the novelty and the advantage of our work, and will include it in the paper in the revision. We thank you for your helpful suggestion.
>
> ---
> **References**
> * Monti et al. Dual-Primal Graph Convolutional Networks. arXiv preprint, 2018.

---

> ### Author Response · Authors · 2021-08-28
> **The end of the discussion phase is approaching**
>
> Dear Reviewer 5K8S,
>
> We sincerely appreciate your positive comments on our contributions in methodological and experimental sides: our method using message passing on the dual to identify not so relevant or noisy edges in the primal (and thus realize some sort of learnable rewiring operation) is a new contribution to the field; the idea of coupling a graph-wise representation based on node features with one explicitly aimed at representing edges is a new contribution; the experiments are overall good and extensive. We have made every effort to faithfully address all your comments in the responses, and briefly summarize the main points of our response below:
>
> - We have clarified that the main novelty of our work comes from **edge representation learning using Dual Hypergraph Transformation** which has not been done before. Furthermore, we have shown the **importance of edge representation learning for general graph-related tasks** which has been largely overlooked.
>
> - We have explained the difference of our framework to existing edge-representation learning methods using the line graphs, and showed that our method is extremely efficient in the transformation and does not suffer from the loss of node information, unlike the line graph-based methods.
>
> - We have clarified the complexities in Table 1 and further described the sparse implementation of our proposed DHT.
>
> - We have provided additional qualitative results for the edge reconstruction on the synthetic dataset, and clarified details for the graph reconstruction experiments.
>
> - We have clarified the definition of compression ratio and added the missing citation.
>
> We thank you again for your time and efforts in reviewing our paper, and sincerely appreciate your insightful and constructive comments. We believe that including the new discussions, clarification, and additional qualitative results into the paper will significantly strengthen it.  Please let us know if you have any further questions.
>
> Best regards, Authors

---

> ### Author Response · Authors · 2021-09-03
> **Thank you**
>
> Dear Reviewer 5K8S,
>
> We sincerely appreciate your positive review on our paper. We did our best to faithfully address all comments from you and other reviewers during the long discussion period, and believe that our answers successfully cleared all concerns and questions. We thank you again for your time and efforts in reviewing our paper.
>
> Best regards,
> Authors

---

### Official Review · Reviewer_EuYn · 2021-06-29

**Rating:** 5
**Confidence:** 5

**Summary:**

--------------------------------------------------------------------------------------------------------------------------

* Most graph representation learning methods focus on representing nodes or entire graphs but not edges.
* The paper proposes a dual hypergraph transformation of the input graph (nodes become hyperedges while edges become nodes) to easily learn representations of edges in the graph (the key idea is to use a node representation learning method on the dual).
* Experiments on graph reconstruction and graph generation, two tasks in which edges are crucial, demonstrate the benefits of the proposed edge representation learning method.

**Limitations And Societal Impact:**

Adequately addressed.

**Main Review:**

--------------------------------------------------------------------------------------------------------------------------

### **Strengths**

--------------------------------------------------------------------------------------------------------------------------

1. **Quality of Submission**

The methods used (Dual Hypergraph, HyperCluster, HyperDrop) are appropriate for the tasks (graph reconstruction, generation, and classification).
The submission is a complete piece of work (i.e., not a work in progress).
The authors have also evaluated the weaknesses (albeit briefly) of their work (page 14 of supplementary/appendix).

--------------------------------------------------------------------------------------------------------------------------

2. **Clarity of Presentation**

The submission is well-organised and well-written.
The submission contains a detailed supplementary material that includes code, details of additional experiments, and descriptions of baselines, datasets, tasks, and hyperparameters.
The code seems well-organised and can be easily adapted by others without much effort.

--------------------------------------------------------------------------------------------------------------------------

### **Weaknesses**
The submission can be strengthened by improving the originality, significance of the contributions and positioning with prior work better.

--------------------------------------------------------------------------------------------------------------------------

1. **Originality of Contributions**

The tasks are not new and the methods used are well-known.
The hyperedges of the dual hypergraph are not treated any differently from pairwise edges: the key idea is to run a known graph-based method on the transpose of the incidence matrix (a straightforward idea) as discussed in line 207.
HyperCluster and HyperDrop are also straightforward adoption of known techniques from prior graph pooling models [e.g., (i) Hierarchical graph representation learning with differentiable pooling, NeurIPS'18, (ii) Self-attention graph pooling, ICML'19] to nodes of the dual hypergraph.

--------------------------------------------------------------------------------------------------------------------------

2. **Significance of Experimental Results**

The graph generation results in Table 2 reveal higher variances for the proposed method than baselines.
The experimental methodology could be strengthened by using statistical tests to validate the significance of performance gains.
The authors should clarify how a simple Multi-layer Perceptron (MLP) baseline (without graph connectivity information) performs on graph reconstruction, generation tasks (this would strengthen claims made in lines 37--42).

--------------------------------------------------------------------------------------------------------------------------

3. **Positioning with Prior Work**

Learning representations for both nodes and hyperedges in a hypergraph has been a recent topic of research interest which includes the following publications:
* Be More with Less: Hypergraph Attention Networks for Inductive Text Classification, In EMNLP'20
* Session-based Recommendation with Hypergraph Attention Networks, In SDM'21.

The submission can be strengthened by positioning (and comparing) with hypergraph attention networks since they can be applied easily to a dual hypergraph.
The results on oversmoothing (Figure 10) are not surprising since dropping nodes and/or edges from a graph is  well-known to alleviate oversmoothing issues, e.g., Bayesian Graph Neural Networks with Adaptive Connection Sampling, ICML'20.

--------------------------------------------------------------------------------------------------------------------------

**Time Spent Reviewing:**

5 hours

---

> ### Author Response · Authors · 2021-08-08
> **Initial Response (2/2) to Reviewer EuYn**
>
> **Question 3-1:** Learning representations for both nodes and hyperedges in a hypergraph has been a recent topic of research interest which includes the following publications:
> * (Ding et al., 2020) Be More with Less: Hypergraph Attention Networks for Inductive Text Classification. EMNLP 2020.
> * (Wang et al., 2021) Session-based Recommendation with Hypergraph Attention Networks. SDM 2021.
> The submission can be strengthened by positioning (and comparing) with hypergraph attention networks since they can be applied easily to a dual hypergraph.
>
> **Answer:** We thank you for your suggestion of the related works on Hypergraph Attention Networks. However, there seems to be a critical misunderstanding of the contribution of our work, since the suggested works are orthogonal to ours, and a direct comparison between the two is not possible. Neither of the two works, or any works on hypergraph representation learning, deal with **dual hypergraphs** nor focus on **edge representation learning**. Please note that the general hypergraph representation learning is not the focus of our method, but the edge representation learning is. We can replace Hypergraph GCN used in our experiments with Hypergraph Attention Networks, but a specific choice of hypergraph message passing algorithm is irrelevant to our edge representation learning framework. We further discuss the difference between each work and ours below:
>
> The first work (Ding et al., 2020) aims to classify a document with interactions among words in the document. To model the interaction between words (nodes), they use hypergraphs to model the high-order interactions among nodes by grouping a set of edges into one hyperedge. Then they utilize an attention-based GNN similar to GAT (Veličković et al., 2018) for message passing on that hypergraph. However, this work is completely different from ours, except that both use hypergraphs. First, while Ding et al. (2020) utilizes the hypergraphs to model high-order interactions among words, ours considers the **dual hypergraph** to represent edges of the conventional graphs as the nodes. Also, they use the representation of edges only to augment the node representations. However, we found such schemes to be highly suboptimal for graph representation learning in our experiments (Figure 3, 4, 5, 6), compared with our method which learns a separate set of explicit edge representations.
>
> Similarly, the second work (Wang et al., 2021) proposes to represent items (nodes) in the session with hypergraphs for considering high-order interactions among nodes, to predict the next preferable items in the hypergraph. They also use an attention-based graph neural network similar to GAT (Veličković et al., 2018), for message passing on the hypergraph. However, it also does not learn edge representations and solely resorts to nodes, which is highly different from our objective of learning edge representations via utilizing hypergraph duality.
>
> Nonetheless, we will compare Hypergraph Attention Networks with Hypergraph GCN for message passing as suggested, and include the results in the revision.
>
> ---
> **Question 3-2:** The results on oversmoothing (Figure 10) are not surprising since dropping nodes and/or edges from a graph is well-known to alleviate oversmoothing issues, e.g., Bayesian Graph Neural Networks with Adaptive Connection Sampling, ICML'20.
>
> **Answer:** This is a misunderstanding of our intention. We did not provide the results as a novel and surprising finding of how node/edge drop alleviates oversmoothing. Instead, this is a task to show the effectiveness of our HyperDrop edge pooling method on **node classification tasks**, which is not possible with conventional node pooling methods since they inevitably drop or collapse nodes.
>
> Further, HyperDrop is significantly more effective than stochastic edge drop methods, such as DropEdge (Rong et al., 2020) and GDC (Hasanzadeh et al., 2020) since it identifies **task-relevant edges** and preserve them, while the sampling-based methods randomly drop edges with either fixed (Rong et al., 2020) or learnable (Hasanzadeh et al., 2020) drop rates. The results in Figure 10 clearly show the effectiveness of HyperDrop to baselines, in alleviating the over-smoothing issue.
>
> ---
> **References**
> * Ding et al. Be More with Less: Hypergraph Attention Networks for Inductive Text Classification. EMNLP 2020.
> * Wang et al. Session-based Recommendation with Hypergraph Attention Networks. SDM 2021.
> * Veličković et al. Graph Attention Networks. ICLR 2018.
> * Rong et al. DropEdge: Towards Deep Graph Convolutional Networks on Node Classification. ICLR 2020.
> * Hasanzadeh et al. Bayesian Graph Neural Networks with Adaptive Connection Sampling. ICML 2020.

---

> > ### Comment · Reviewer_EuYn · 2021-08-09
> > **Thanks for the Response**
> >
> > ------
> >
> > Thanks to the authors for the response. I appreciate the discussion on positioning with prior work, statistical tests, and oversmoothing. While I agree with the authors that the proposed edge pooling mechanism is relevant for the paper, the idea has been explored previously, e.g., Learning Dual-Pooling Graph Neural Networks for Few-shot Video Classification, IEEE Transactions on Multimedia 2020.
> >
> > ------
> >
> > The concerns on originality and MLP comparison on edge-crucial tasks such as graph generation, reconstruction remain. The other reviewers have also pointed out that the hypergraph nature of the dual hypergraph is not exploited. In addition to the papers pointed out by Reviewer 5K8S, the following papers also employ the idea of dual hypergraphs for (hyper)edge-level tasks:
> > * Classification in biological networks with hypergraphlet kernels, Bioinformatics 2020
> > * Link Prediction in Hypergraphs using Graph Convolutional Networks, 2018
> > * Multiplex Bipartite Network Embedding using Dual Hypergraph Convolutional Networks, WebConf'21
> >
> > ------

---

> > > ### Author Response · Authors · 2021-08-09
> > > **Second Response to Reviewer EuYn**
> > >
> > > Thank you for your response. We appreciate your quick response, and further address your remaining concerns below:
> > >
> > > ---
> > > **Question 1:** The idea (edge pooling mechanism) has been explored previously, e.g., Learning Dual-Pooling Graph Neural Networks for Few-shot Video Classification, IEEE Transactions on Multimedia 2020. (Hu et al., 2020)
> > >
> > > **Answer:** Since there still seems to remain a misunderstanding of the main novelty of our work, we briefly recap it again. Our main contributions are as follow: 1) proposal of Dual Hypergraph Transformation for **edge representation learning**, 2) proposal of the edge pooling methods for learning **explicit graph-level edge representations**, and 3) demonstrating the effectiveness of edge representation learning on **general tasks with graph-structured data**, such as graph classification, reconstruction, and generation.
> > >
> > > - Hu et al. 20, although it has a mechanism to drop unimportant edges, its goal is removing noisy edges. Specifically, it first constructs a fully-connected graphs between frames of a video, and removes the edges across the videos for a better **node-level message passing**, and the work has nothing to do with edge representation learning.
> > > - Moreover, the meaning of “dual” in Hu et al. (2020) does not refer to the dual hypergraph, nor any hypergraphs, but simply denotes the usage of **two** separate poolings on nodes and edges. Thus, it has nothing to do with hypergraph duality.
> > > - Finally, please note that our main motivation is that edge information is important for many general graph tasks, and thus it is necessary to learn explicit edge representations. Hu et al. (2020) is targeting a specific video classification task, and thus its motivation and goal is completely different from ours.
> > > - As Reviewer 5K8S has acknowledged, using message passing on the dual to identify not so relevant or noisy edges in the primal and thus realize some sort of learnable graph rewiring operation is a new contribution to the field. To summarize, Hu et al. (2020) does not 1) learn edge representations, nor 2) works with dual hypergraphs, and 3) is targeting a specific edge-related task rather than general graph-level tasks.
> > >
> > > ---
> > > **Question 2:** MLP comparisons on edge-crucial tasks such as graph generation, reconstruction remain.
> > >
> > > **Answer:** Simple MLPs are not meaningful baselines on edge-crucial tasks, since they simply **ignore** the connectivity information and treat the nodes as sets. Nevertheless, to address your concerns, we show the performance of the MLPs on graph generation and reconstruction tasks as you suggested.
> > > - Please note that we already compared against the **MLP model for edge representation learning on graph generation tasks** in Table 2, as the baseline MARS uses the MLPs to represent edges for the deletion actions as formally denoted in Eq. 18, and described in Line 302-305 of the supplementary file.
> > > - In addition, we further experimented the MLP baseline to the MolGAN experiment in Figure 8. We construct the discriminator and the rewarder in MolGAN, which are originally the R-GCNs, as the MLPs. We observe that the MolGAN generator **fail to generate any valid molecules** in such a setting, and thus we cannot evaluate the generated molecules on chemical metrics, such as SA and QED in Figure 8.
> > > - Finally, we further experimented with the MLPs on the reconstruction tasks. As shown in the table below (the former one is for Exact Match; the latter one is for Accuracy), The MLP baseline results in poor exact match score, and low accuracy, while our HyperCluster obtains impressive scores that are sometimes near-perfect.
> > >
> > > Exact Match:
> > >
> > > | Compression Ratio (%) | 5 | 10 | 15 |
> > > | --- | --- | --- | --- |
> > > | MLPs | 4.77 $\pm$ 0.51 | 5.62 $\pm$ 0.50 | 6.26 $\pm$ 0.25 |
> > > | HyperCluster (Ours) | 95.79 $\pm$ 4.45 | 99.11 $\pm$ 1.27 | 98.09 $\pm$ 1.04 |
> > >
> > > Accuracy:
> > >
> > > | Compression Ratio (%) | 5 | 10 | 15 |
> > > | --- | --- | --- | --- |
> > > | MLPs | 79.11 $\pm$ 0.32 | 79.44 $\pm$ 0.23 | 79.59 $\pm$ 0.20 |
> > > | HyperCluster (Ours) | 99.72 $\pm$ 0.25 | 99.94 $\pm$ 0.08 | 99.90 $\pm$ 0.05 |
> > >
> > > We will include the new experimental results with the simple MLPs in the revision.
> > >
> > > ---
> > > **Question 3:** The other reviewer (5K8S) has also pointed out that the hypergraph nature of the dual hypergraph is not exploited.
> > >
> > > **Answer:** This is a misunderstanding about our Dual Hypergraph Transformation (DHT). Our DHT exploits the nature of the dual hypergraph in both the transformation and its implementation:
> > > 1) We utilize the fact that the incidence matrix of the dual hypergraph is equal to the transpose of the incidence matrix of the original graph, which yields no cost for obtaining node and hyperedge features of the dual hypergraph from its original graph.
> > > 2) Moreover, the sparse matrix of our DHT can be easily implemented with simple tensor reshape operations, using the property that the dual hypergraph is a 2-regular hypergraph. The specific details are described in Section A.3 of the supplementary file.
> > >
> > > ---
> > > **Question 4:** In addition to the papers pointed out by Reviewer 5K8S, the following papers also employ the idea of dual hypergraphs for (hyper)edge-level tasks:
> > > * Classification in biological networks with hypergraphlet kernels, Bioinformatics 2020. (Lugo-Martinez and Radivojac, 2017)
> > > * Link Prediction in Hypergraphs using Graph Convolutional Networks, arXiv 2018. (Yadati et al., 2018)
> > > * Multiplex Bipartite Network Embedding using Dual Hypergraph Convolutional Networks, WebConf'21. (Xue et al., 2021)
> > >
> > > **Answer:**
> > > We thank you for the suggestion, but they are only related to ours either in one of the two aspects **1) edge representation learning**, and **2) exploiting hypergraph duality**. We already acknowledge existing works on both topics, in the related work. Thus they are orthogonal to our contribution of **edge representation learning with hypergraphs**, which is simple yet effective, and yields largely enhanced performance on general graph tasks.
> > > We discuss the specific differences between our work and the three prior works below:
> > > - The work of Xue et al. (2021) is clearly irrelevant to the dual hypergraphs used in our paper. They propose to embed nodes in the bipartite graphs with two distinct sets of nodes, by constructing two sets of hypergraphs corresponding to the two sets of nodes, to promote information exchange within and across two different node-domains. Thus, the word dual in the mentioned paper simply denotes **two hypergraphs** and has nothing to do with the dual hypergraph for edges.
> > > - Lugo-Martinez and Radivojac (2017) propose graph kernels that exploit hypergraph duality for specific edge-related tasks, and thus there is no edge-level message passing or edge pooling done for graph-level edge representation learning.
> > > - Yadati et al. (2018) exploit hypergraph duality, but its goal is not learning graph-level edge representations, and do not perform message passing or graph pooling with a dual hypergraph. Rather, they construct a regular graph similar to the line graph for the edges, by utilizing “clique expansion” to approximate the dual hypergraph into planar graphs by replacing every hyperlink of size $s$ with a $s$-clique. Thus this method does not perform message passing on the **dual hypergraph**. Further, this approximation to the clique expansion has a clear disadvantage, as it not only results in the loss of the edge information (i.e., some of two different hyperedges are merged into one edge) but also fails to capture high-order relationships among nodes due to transformation to the planar graph. Contrarily, our method directly learns the edge representation on the dual hypergraph, which does not require any cost.
> > > Finally, we want to emphasize that all suggested works focus on a specific edge-level task, while we tackle general graph-related tasks, and report impressive performance on all, which shows the importance of edge-level representations even for non-edge specific tasks, that have been relatively overlooked.

---

> > > > ### Comment · Reviewer_EuYn · 2021-08-10
> > > > **Thanks for the response**
> > > >
> > > > ---
> > > >
> > > > Thanks to the authors again for the response and I appreciate the MLP comparison. The clarity of presentation is certainly a strength of the paper and there were never any misunderstandings of the main claims made in the paper (nor the word "dual" appearing in paper titles). In authors' words, the main contributions are as follows:
> > > > 1) proposal of Dual Hypergraph Transformation for edge representation learning,
> > > > 2) proposal of the edge pooling methods for learning explicit graph-level edge representations,
> > > > 3) demonstrating the effectiveness of edge representation learning on general tasks with graph-structured data, such as graph classification, reconstruction, and generation.
> > > >
> > > > ---
> > > >
> > > > While I agree with the authors that the contributions have not been explored before and the paper is the first paper to put these contributions together with relevant experiments, the key ideas and the advantages of all the ideas are well-known which have been explored previously in different settings. After carefully reading the author's response, I can increase the confidence score but I still tend to marginally lean towards rejection (though accepting the paper would not be bad).
> > > >
> > > > ---

---

> > > > > ### Author Response · Authors · 2021-08-12
> > > > > **A novel combination of well-known techniques could be novel, according to the NeurIPS 2021 reviewer guideline.**
> > > > >
> > > > > We sincerely appreciate your timely responses. During the rebuttal period, we have done our best to address your concerns and correct your misunderstandings, including from your original comments: significance of results; positioning with prior work; to your follow-up comments: MLP baselines; comparison against work on either one of edge learning or hypergraph duality. While we believe that most of your concerns have been resolved through the interaction with you, there still seems to be a misunderstanding of the key ideas of our work, which we address below:
> > > > >
> > > > > ---
> > > > > **Question:**  The key ideas and the advantages of all the ideas are well-known which have been explored previously in different settings.
> > > > >
> > > > > **Answer:**
> > > > > We sincerely appreciate your acknowledgement of the contributions of our work in three aspects. As you mentioned, "the contributions have not been explored before and the paper is the first paper to put these contributions together with relevant experiments." However, you still seem to be worried about the originality of each individual component of our framework, based on your comment "the key ideas and the advantages of all the ideas are well-known which have been explored previously in different settings". However, we do not believe that we need to reinvent the wheel for every component of our method. Hypergraph construction and message passing are only tools, rather than our focus, and we do not claim that ours is the first work on edge representation learning, as we clearly discussed existing works on both hypergraphs and edge representation learning in the related work section of our paper.
> > > > >
> > > > > Also, the NeurIPS 2021 **Reviewer Guidelines (https://neurips.cc/Conferences/2021/Reviewer-Guidelines)** clearly states that, **a novel combination of well-known techniques should not be automatically considered as trivial and can be valuable**. We strongly believe that our work is certainly such a “novel combination” as you kindly recognized, and also is a “valuable” contribution in the field, as our edge representation learning is simple to implement, computationally efficient, highly general as it allows the application of existing message passing and pooling methods, and may have large practical impact as it allows to achieve remarkable performances on various graph-level tasks.
> > > > >
> > > > > We further want to introduce self-attention as an example of such a novel combination of known ideas. The attention mechanism is a well-known concept, but the idea of using a sequence to attend to itself, to learn the relations between the elements, is considered as a novel contribution for learning the representations of sequences. This is because the effectiveness of the self-attention mechanism for sequence representation learning cannot have automatically emerged from the concept of attention, and could be only proposed as a combination of an existing idea (attention) and an existing problem (sequence representation learning). Dual Hypergraph Transformation for edge representation learning with hypergraph message passing and hypergraph pooling, as you acknowledged, is our novel contribution. Therefore we politely insist that our work should be interpreted as a whole, indiscerptible framework, rather than the sum of its components. To faithfully clarify your misconception about originality of our work once more, we refresh our key ideas and our advantages in two folds:
> > > > >
> > > > > **Comparison with the works related to (hyper)edge representation learning**
> > > > > - Lugo-Martinez and Radivojac. Classification in biological networks with hypergraphlet kernels, Bioinformatics 2021.
> > > > > - Yadati et al. Link Prediction in Hypergraphs using Graph Convolutional Networks. arXiv preprint. 2018.
> > > > >
> > > > > These existing works do not perform message passing nor graph pooling using a dual hypergraph. Further, their goal is not learning graph-level edge representations,
> > > > >
> > > > > **Comparison with the works related to hypergraph**
> > > > > - Ding et al. Be More with Less: Hypergraph Attention Networks for Inductive Text Classification. EMNLP 2020.
> > > > > - Wang et al. Session-based Recommendation with Hypergraph Attention Networks. SDM 2021.
> > > > > - Xue et al. Multiplex Bipartite Network Embedding using Dual Hypergraph Convolutional Networks, WebConf'21.
> > > > >
> > > > > These works on hypergraphs are irrelevant to edge representation learning or the dual hypergraphs. Furthermore, the term “dual hypergraphs” used in Xue et al. (2021) denotes “two hypergraphs”.
> > > > >
> > > > > **Comparison with the works related to (edge) pooling**
> > > > > - Ying et al. Hierarchical graph representation learning with differentiable pooling. NeurIPS 2018.
> > > > > - Lee et al. Self-attention graph pooling. ICML 2019.
> > > > > - Hu et al. Learning Dual-Pooling Graph Neural Networks for Few-shot Video Classification. IEEE Transactions on Multimedia 2020.
> > > > >
> > > > > The works of Ying et al. (2018) and Lee et al. (2019) are well known node-based pooling methods, which have nothing to do with the hypergraphs nor dual hypergraphs, and are clearly different from our specific edge pooling methods. Moreover, the term “dual” simply denotes that they perform “two” separate poolings.
> > > > >
> > > > > ---
> > > > >
> > > > > We now discuss the novel advantages of our key idea. In contrast to your argument that the advantages of key ideas are well-known, the benefits of our edge representation learning with hypergraphs have been **largely unknown**:
> > > > >
> > > > > - The idea of applying the hypergraph duality, to realize message passing between edges, has never been used before, which the Reviewer 5K8S has agreed by the comment **“using the message passing on the dual to identify not so relevant or noisy edges in the primal is new”**. Compared to existing works related to edge representation learning our method **does not require specially designed schemes for realizing the message passing between edges**. Also, the transformation to dual hypergraph is simple and intuitive, which is more efficient than other edges-to-nodes transformation using graphs (e.g., line graph, bipartite graph, or clique expansion).
> > > > >
> > > > > - Obtaining graph-level edge representations is another key idea, which was not considered in the previous works. By applying the pooling methods on the dual hypergraph, we can cluster similar edges to get holistic graph representations, or identify task-relevant edges for effective message passing, while at the same time achieving lossless compression of the nodes. Reviewer 5K8S has pointed out that our **learnable graph rewiring operation is a new contribution**to the field, and no other works have recognized these advantages.
> > > > >
> > > > > - Exploiting well-known methods does not diminish our contribution, but rather shows the **versatility and generality of our methods**. This is because with our method, there is no need to reinvent the wheels for edge representation learning. The main strength is that we can apply any node-based message passing or graph pooling methods to edges. Further, we have clearly demonstrated the importance of edge representation learning, with impressive performance on general graph-related tasks.
> > > > >
> > > > > - The **advantage of edges-to-nodes transformation for graph-level representation learning has been largely undiscovered**, and our work is the first to tackle such unknown suboptimality on graph-level tasks (reconstruction, generation, and classification) with our simple yet effective solution. In particular, the experimental setups and results are favorably acknowledged by other reviewers: JU4W, xkvv, and 5K8S. Our work is the first to explore such overlooked advantages, and further verified it with extensive experiments.
> > > > >
> > > > > ---
> > > > >
> > > > > We believe that we have faithfully addressed all your remaining concerns, thus politely ask you to recognize the novelty in the combination of the components following the NeurIPS reviewer guideline, and revise the score and the review accordingly. We thank you again for your time and efforts in reviewing our paper.

---

> > > > > > ### Comment · Reviewer_EuYn · 2021-08-12
> > > > > > **A combination of well-known techniques could also not be novel if it is not non-trivial**
> > > > > >
> > > > > > Thanks for the discussion.
> > > > > >
> > > > > > The construction of a dual hypergraph is trivial and easy to see, in general, from any bipartite graph with say, $V_1$, $V_2$ as the two partitions. If one side of the partition is treated as nodes of a primal hypergraph (say $V_1$), the other side would become the hyperedges ($V_2$) (this is essentially the idea used to embed hyperedges through message-passing in most of the publications discussed in the rebuttal). The dual of this primal is when $V_2$ is viewed as nodes and $V_1$ as hyperedges (message passing on the dual can be done through a role interchange of $V_1$ and $V_2$, or, as this submission says, through the transpose of the incidence matrix).
> > > > > >
> > > > > > The dual hypergraph of a pairwise graph $(V, E)$ considered in this submission is obtained by setting $V_2=E$ (nodes) and $V_1=V$ (hyperedges). Given a large body of work on message passing on bipartite graphs (with two different vertex types) and hypergraphs, I do not believe the ideas of (i) constructing the dual hypergraph of a pairwise graph, (ii) message passing on the dual, (iii) pooling with edges treated as nodes (of the dual) are significantly novel contributions for NeurIPS. The other contributions and advantages could be valuable but given the high acceptance bar of NeurIPS, I tend to lean towards marginal rejection.

---

> > > > > > > ### Author Response · Authors · 2021-08-14
> > > > > > > **The task-level, framework-level, and component level novelty of our work**
> > > > > > >
> > > > > > > We appreciate your quick response, and are happy to respond to your comments.
> > > > > > >
> > > > > > > ---
> > > > > > > **Question:** I do not believe the ideas of (i) constructing the dual of a graph (with pairwise edges), (ii) message passing on the dual, (iii) pooling with edges as nodes are significantly novel contributions.
> > > > > > >
> > > > > > > **Answer:** It seems that there still exist misunderstandings about the novelty of our work, and thus we describe our novelty in the following three aspects: **task-level, framework-level, and component-level**.
> > > > > > >
> > > > > > > ## Task-level novelty
> > > > > > >
> > > > > > > - We first want to emphasize that our main contribution is indeed the focus on edge representation learning, which has been **largely overlooked**. However, we find it **crucial to the success of graph-level representation learning**. In other words, we target the limitation of current GNNs that focus on nodes and using the edges only as auxiliary information for node representation learning, by pointing out the importance of explicit representation learning for the edges, and demonstrating this through experimental studies on four different tasks. This problem-side novelty is **acknowledged by most reviewers** -- “this paper studies an interesting and useful problem to learn the representation of edges” (Reviewer xkvv); “the authors work on an interesting problem of edge representation learning” (Reviewer JU4W); “the idea of coupling a graph-wise representation based on node features with one explicitly aimed at representing edges I believe is a new contribution” (Reviewer 5K8S) -- and we strongly believe that this task-level novelty should not be dismissed and further be highlighted, for a fair evaluation of our work.
> > > > > > >
> > > > > > > - To tackle this important but largely undiscovered problem, we found that the combination of edge representation learning and hypergraph duality is very **simple and versatile, yet effective and useful**, which we concretely explain in the next section.
> > > > > > >
> > > > > > > ---
> > > > > > > ## Framework-level novelty
> > > > > > >
> > > > > > > - Furthermore, we want to highlight the novelty of our proposed framework. While we have tried our best to clarify the key idea of our work is the **edge representation learning with hypergraph duality**, in our paper and the past responses, it seems that you still interpret our work as a mere sum of its components, thus here we restate our **framework-level novelty** more concretely.
> > > > > > >
> > > > > > > - **The novelty we claim is neither (i) nor (ii)**, but the **novel framework which puts them togethe for edge representation learning**. We are confident that no existing works have proposed such integration, although this seemingly straightforward framework achieves **remarkable performance** in a vast range of graph-level tasks. (iii) that you mentioned is at last made possible under the complete framework (i)+(ii) we suggested, and it is just one of many advantages our framework has. Most of these advantages are not the properties of the individual components, but can only emerge when they are **put together as a single framework**. Dissecting our edge learning framework into individual components, and saying each of them is not novel, is exactly the same as saying that the self-attention mechanism is not novel since 1) it utilizes an existing attention mechanism, and 2) sequence learning has been done before, and 3) it is a straightforward idea to use the attention mechanism for sequence learning. However, this is not the novelty of the self-attention learning framework, but the idea of utilizing the attention mechanism for sequence representation learning is. Similarly, neither edge representation learning nor hypergraph message passing is our novelty, but edge representation learning on the dual hypergraph is, and we believe this is a sufficiently novel idea but has not been explored because existing works overlooked the importance of explicit edge representation learning. If this is such a trivial idea, why hasn't anybody proposed to use them, while they help achieve such remarkable performances?
> > > > > > >
> > > > > > > While we already discussed the advantages of our combination in the previous response, we would like to restate them below:
> > > > > > >
> > > > > > > **Versatility** Our edge representation learning framework eliminates the need to design tailored message passing and even pooling schemes for edges, since the Dual Hypergraph Transformation interchanges the structural role of nodes and edges without costs. The combination of these two components, (i) dual hypergraph transformation and (ii) edge-level message passing for edge representation learning, should be considered as an indiscerptible framework.
> > > > > > >
> > > > > > > **Utility** Our edge representation learning framework has numerous utilities on various graph-level tasks, and we show which architectural variation is appropriate for each task throughout our experiments, as follows:
> > > > > > >
> > > > > > > - In the **graph reconstruction** task, we show that the graph-level edge representation obtained by clustering similar edges, is highly beneficial to compress edge representations compared to various edge-aware GNNs (Figure 3, 4, 5, 6, and 7). 2) In the **graph generation** task, we show that accurately learning edge representations is useful to generate molecules with desired chemical properties than other edge-aware GNNs, since ours can precisely distinguish different molecules with accurate edge information (Figure 8 and Table 2).
> > > > > > >
> > > > > > > - In the **graph classification** task, we present that the hierarchical edge drop scheme using the obtained edge representations from message-passing outperforms existing hierarchical node-based pooling methods (Table 3), by dropping uninformative edges while preserving the node information. We also present qualitative evidence of our effective edge drop with visual examples in Figure 9. Further, we verify that our method is orthogonal to global pooling methods, and using both methods together can significantly improve the performance.
> > > > > > >
> > > > > > > - In the **node classification** task, we validate that identifying task-relevant edges with our method is helpful to alleviate the over-smoothing problem of GNNs (Figure 10), better than the random edge dropping scheme. Previous node-based pooling methods cannot be applied to such tasks due to the inevitable loss of node information.
> > > > > > >
> > > > > > > The above advantages of our edge representation learning framework clearly and persuasively support that it can be extensively used for various graph-related tasks, where the importance of edge representations has been largely overlooked. To sum up, we believe that the significance of our work not only comes from our **effective and efficient framework** of learning edge representations, but also from our **extensive experiments** on substantial graph-level tasks, in the framework-level.
> > > > > > >
> > > > > > > ---
> > > > > > > ## Component-level novelty
> > > > > > >
> > > > > > > We believe that the above explanations of the task- and the framework-level of our novelty sufficiently clear up your concerns on (i), (ii), and (iii). However, even putting aside the task- and framework-level novelty for edge representation learning, our work has distinct points in the component-level comparison.
> > > > > > >
> > > > > > > 1) We have never claimed the novelty for “(i) constructing dual hypergraphs” as we have clearly explained in our initial response, which we have also clearly acknowledged the hypergraph duality (Berge, 1973; Scheinerman & Ullman, 2011) and the relevant works in the Related Work.
> > > > > > >
> > > > > > > 2) The idea (ii), applying the hypergraph duality to realize message passing between edges for learning explicit edge representations with GNNs, **has never been exploited before**, including other settings. The Reviewer 5K8S has agreed by the comment “using the message passing on the dual to identify not so relevant or noisy edges in the primal is new”.
> > > > > > >
> > > > > > > 3) The idea (iii) you mentioned is not our key idea for the proposed HyperCluster and HyperDrop, and is a clear misunderstanding of our methods, since our focus is not on the removal of unnecessary edges. The key idea for our novel edge pooling methods is **obtaining graph-level edge representations** to solve the downstream tasks, which we approach by pooling nodes in the dual hypergraph. Further, to the best of our knowledge, **no other work has considered using pooling in the dual hypergraph for edge representation learning**. Reviewer 5K8S also has pointed out that our “learnable graph rewiring operation is a new contribution to the field”, and no other works have recognized these advantages.
> > > > > > >
> > > > > > > ---
> > > > > > > We have faithfully responded to the originality of our work, and have answered to the difference with the provided references. If you have further concerns in regard to the novelty of our work even in the task-, framework-, and the componenet-level, we are happy to resolve them in further discussions. We hope the above responses satisfactorily address your points.
> > > > > > >
> > > > > > > Thanks, Authors.
> > > > > > >
> > > > > > > ---
> > > > > > > **References**
> > > > > > > * Berge. Graphs and hypergraphs. 1973.
> > > > > > > * Scheinerman & Ullman. Fractional graph theory: a rational approach to the theory of graphs. Courier Corporation. 2011.

---

> > > > > > > > ### Comment · Reviewer_EuYn · 2021-08-14
> > > > > > > > **Thanks for the Response**
> > > > > > > >
> > > > > > > > Let us politely agree to disagree on the novelty and positioning of the paper. There are merits in the paper that are very clear from the presentation. The description for the overall score in reviewer guidelines perfectly summarises my review of the paper:
> > > > > > > >
> > > > > > > > `5: Marginally below the acceptance threshold.`
> > > > > > > >
> > > > > > > > `I tend to vote for rejecting this submission, but accepting it would not be that bad.`

---

> ### Author Response · Authors · 2021-08-08
> **Initial Response (1/2) to Reviewer EuYn**
>
> We sincerely thank you for your constructive and helpful comments. We appreciate your positive comments that the submission quality is a complete piece of work that is well-organized and -written, and the methods we used are appropriate for substantial graph tasks. We initially address all your concerns below:
>
> ---
> **Question 1-1:** The tasks are not new and the methods used are well-known. The hyperedges of the dual hypergraph are not treated any differently from pairwise edges: the key idea is to run a known graph-based method on the transpose of the incidence matrix (a straightforward idea) as discussed in line 207.
>
> **Answer:** This is a critical misunderstanding of the contribution of our work. We did not claim that we have invented or discovered hypergraph duality (Berge, 1973; Scheinerman & Ullman, 2011), which is a well-known concept from graph theory. Our novelty is rather on the proposal of **Dual Hypergraph Transformation for edge representation learning, which exploits hypergraph duality**. Although hypergraph duality has been well known, it **has not been applied for edge representation learning**for graphs, and ours is the **first work** that exploits it.
>
> Exploiting hypergraph duality for edge representation learning is **not a straightforward idea**since none of the existing works have done so. It indeed is conceptually very simple to understand, but this does not automatically mean that coming up with the idea is simple. Message passing for node representation learning is already done in WL-test, or graph kernel learning, but still its application to graph representation learning with GNNs is a meaningful contribution. If exploiting hypergraph duality is such a simple idea, then why hasn't anyone proposed it before, while it yields impressive performance on a variety of graph-related tasks, such as graph classification, reconstruction, and generation? We believe that the main reason why such a “simple” idea has not been explored thus far, is because researchers have overlooked the needs of learning explicit representations for edges. We have shed a fresh light on the problem, demonstrating its importance on graph classification, reconstruction, and generation, which is another very important contribution of this work.
>
> The ability to apply well-known message passing and graph pooling methods for nodes to edges, handling the edges in the same manner as nodes, **does not show the lack of originality**of our work, but is rather the **main strength**of our method. This is because with our method, there is no need to reinvent the wheels for edge representation learning. Previous edge representation learning methods (Gong & Cheng, 2019; Yang & Li, 2020) do not allow to treat the edges in the same manner as the nodes, and thus cannot utilize existing message passing and pooling methods for nodes.
>
> ---
> **Question 1-2:** HyperCluster and HyperDrop are also straightforward adoption of known techniques from prior graph pooling models [e.g., (i) Hierarchical graph representation learning with differentiable pooling, NeurIPS'18, (ii) Self-attention graph pooling, ICML'19] to nodes of the dual hypergraph.
>
> **Answer:** While HyperCluster and HyperDrop are straightforward adoptions of conventional graph pooling methods, their **effects are completely different from conventional graph pooling methods**, since they can remove edges or cluster edges which is not possible with the conventional pooling methods. With a similar argument, we can also say that Drop Connect (Wan et al., 2013) or Stochastic Layers (Huang et al., 2016) are straightforward adaptations of Dropout, but they are considered as meaningful contributions since their effects are different. Ours is the **very first work that proposed edge pooling**, and analyzed its effectiveness coming from the lossless compression of node information, and we believe that this is an important contribution.
>
> ---
> **Question 2-1:** The graph generation results in Table 2 reveal higher variances for the proposed method than baselines.
>
> **Answer:** This is a **misinterpretation of the results**. The main reason that the variances are high for the results in Table 2, is because the base model, MARS, shows high variances due to its use of MCMC sampling. The magnitude of the variances of our method is **similar to**the magnitude of the variances of MARS (i.e., the mean of all the standard deviations is **2.9 for MARS and 3.0 for ours**). Moreover, as we can see in the results on MolGAN (Figure 8), our method leads to notably more stable results, which is a further evidence that the high variances in the MARS experiment did not come from the application of our method.
>
> ---
> **Question 2-2:** The experimental methodology could be strengthened by using statistical tests to validate the significance of performance gains.
>
> **Answer:** This seems like a misinterpretation as well, since there is no overlap between the reported performance of the baseline and ours on the ZINC dataset except for JNK3 metric, when considering the standard deviations. However, as suggested, we have performed the t-test to validate the significance of performance gains. The success rate of ours, which summarizes all the metrics on graph generation, is statistically significant against the baseline MARS, from the t-test, with p-value 0.017 that is significantly smaller than significance level 0.05, on the ZINC dataset.
>
> ---
> **Question 2-3:** The authors should clarify how a simple Multi-layer Perceptron (MLP) baseline (without graph connectivity information) performs on graph reconstruction, generation tasks (this would strengthen claims made in lines 37--42).
>
> **Answer:** We apologize for the confusion. The description in lines 37-42 only corresponds to the graph classification tasks, as stated in the previous works (Errica et al., 2020; Huang et al., 2021), and as we empirically show by reporting DeepSet’s performance on graph classification tasks in Table 3. We will clarify this in the subsequent revision.
>
> ---
> **References**
> * Berge. Graphs and hypergraphs. 1973.
> * Scheinerman & Ullman. Fractional graph theory: a rational approach to the theory of graphs. Courier Corporation, 2011.
> * Liyu Gong & Qiang Cheng. Exploiting edge features for graph neural networks. CVPR 2019.
> * Yulei Yang & Dongsheng Li. NENN: incorporate node and edge features in graph neural networks. ACML 2020.
> * Wan et al. Regularization of Neural Networks using DropConnect. ICML 2013.
> * Huang et al. Deep Networks with Stochastic Depth. ECCV 2016.
> * Errica et al. A fair comparison of graph neural networks for graph classification. ICLR 2020.
> * Huang et al. Combining label propagation and simple models out-performs graph neural networks. ICLR 2021.

---

> ### Author Response · Authors · 2021-08-28
> **The end of the discussion phase is approaching**
>
> Dear Reviewer EuYn,
>
> We appreciate your positive comments that our submission is a complete piece of work that is well organized and well written. Moreover, we sincerely appreciate your follow-up comments and your acknowledgements about the main contributions and novelty of our work -- the merits of our paper are very clear from the presentation whose contributions and advantages could be valuable -- during the interactions with us. We have made every effort to faithfully address all your comments in the responses. Please let us know if you have anything else we should address.
>
> We thank you again for your time and efforts in reviewing our paper, and thank you for engaging with us.
>
> Best regards, Authors

---

> ### Author Response · Authors · 2021-09-03
> **Thank you**
>
> Dear Reviewer EuYn,
>
> The long discussion period is over, and we would like to thank you for actively engaging in the discussions with us. During the discussion period, we did our best to faithfully address all concerns from you and other reviewers, and we believe that incorporating these discussions into our paper will significantly strengthen it. We also sincerely appreciate your positive comments on our work, and acknowledgement of the main contributions of our works in three aspects. Thank you again for your time and efforts in reviewing our paper.
>
> Best Regards,
> Authors

---

### Official Review · Reviewer_xkvv · 2021-07-17

**Rating:** 6
**Confidence:** 4

**Summary:**

This paper proposes a new edge representation learning model by hypergraph transformation and message passing. Two edge pooling schemes are designed to learn a holistic representation. The proposed model is validated on graph reconstruction, generation, and classification tasks, and the experimental results show the proposed model outperforms the baseline models on several datasets.

**Limitations And Societal Impact:**

The authors adequately addressed the potential negative societal impact of their work.

**Main Review:**

This paper studies an interesting and useful problem, to learn the representation of edge. The proposed idea and method are intuitive. The experiment results are impressive. However, I have a few concerns about this paper.
(1) Transforming edges to nodes is not very novel, and the strategy is adopted by many existing works, e.g. edge clustering in network community detection. What is the main difference between this paper and the combination of the above transformation and a GCN?
(2) Does the transformation increases the complexity of message passing, especially for the scale-free networks (few hubs with very high degree)?
(3) The descriptions of the tasks in the experiments are not clear and may cause misunderstanding. For example, edge reconstruction is to reconstruct the feature of edge actually.

**Time Spent Reviewing:**

4.5 hours

---

> ### Author Response · Authors · 2021-08-09
> **Initial Response to Reviewer xkvv**
>
> We sincerely thank you for your constructive and helpful comments. We appreciate your positive comments that our work studies an interesting and useful problem, the idea and method are intuitive, and the experiment results are impressive. We initially address all your concerns below:
>
> ---
> **Question 1-1:** Transforming edges to nodes is not very novel, since the transformation strategy is adopted by many existing works, e.g. edge clustering in network community detection.
>
> **Answer:** This is a critical misunderstanding. We do not claim to have proposed an edge representation learning for the first time, and acknowledge existing works on the same topic in the introduction and the related work section. The main novelty of our work comes from the **edge representation learning using Dual Hypergraph Transformation**which converts a graph into its dual form with simple transpose operations applied to incident matrices, which has not been done before. Also, problem-wise, we shed new light on the **importance of edge representation learning for general graph-related tasks** (e.g., graph classification, reconstruction, and generation), which has been relatively overlooked due to the complexity of the edge representation learning. We discuss the main differences between existing works on edge representation learning and ours below:
>
> **Existing works on edge representation learning**
> 1) they require a unique construction of an edge graph which prevents the application of existing message passing and graph pooling methods to the edges.
> 2) The construction of such an edge graph is usually computationally costly.
> 3) They target a specific edge-related task, such as edge clustering, or use the edge representations only to augment the node representations.
>
> **Our edge representation learning with Dual Hypergraph Transformation**
> 1)  It allows to apply conventional message passing and graph pooling methods for nodes to edges, without any change of the algorithm
> 2) It performs node to edge transformation with a single transpose operation without any extra cost.
> 3) We show that edge representation learning can enhance the performance of any general graph-related tasks, such as graph classification, reconstruction, and generation tasks, thanks to the simplicity of our Dual Hypergraph Transformation and its compatibility with conventional message passing and graph pooling.
>
> ---
> **Question 1-2:** What is the main difference between this paper (edge clustering in network community detection) and the combination of the above transformation and a GCN?
>
> **Answer:** The edge to node transformation in our work is done in a completely different manner from the existing works, such as Chen et al. (2019) you mentioned. They utilize a line graph for edge representation learning, which is a **conventional graph (not a hypergraph)** where each node corresponds to an edge of the original graph, while we utilize the **dual hypergraph** for a given graph. This is the main novelty of our work which differentiates it from existing edge representation learning methods, as we clearly stated in Related Work section. Further, the approach by Chen et al. (2019) is highly limited since the line graph construction is 1) not injective, 2) results in the loss of node information during transformation, and 3) has $O(m \times d_{\text{max}})$ complexity for sparse implementation and $O(m^2)$ for dense implementation (see Table 1), where $m$ is the number of edges and $d_{\text{max}}$ is the maximum number of degrees of the node in the original graph. We list the **main differences between our Dual Hypergraph Transformation (DHT) and the line graph construction** in Chen et al. (2019) below:
>
> 1) Our DHT is **bijective** as shown in lines 180-184, and in Figure 2 in our main paper. However the line graph transformation from Chen et al. (2019) is **not injective**. This means that two different (non-isomorphic) graphs may be transformed into the same line graph, in which case a graph could be reconstructed into another graph, or two different graphs are considered as isomorphic.
>
> 2) Our DHT **does not incur any loss of node information**. However, with line graphs, since the nodes of the original graph do not correspond one-to-one to the edges, **the node information may be lost**during the transformation. This is a fatal flaw for tasks such as graph reconstruction and classification. To tackle this problem, Chen et al. (2019) store both the original graph and the line graph, but this greatly increases the memory complexity.
>
> 3) While our DHT can be done with **almost no cost** by applying simple transpose operations to the incident matrices, the line graph transformation is **not scalable**. First, the line graph construction itself requires at least $O(m)$ complexity where $m$ is the number of edges of the original graph. We further show the inefficiency of line graph construction, with the experimental results in the table below. Also, the constructed line graphs may have excessively large edges if the original graphs have very large degrees (e.g., scale-free graphs and community graphs). Specifically, the number of edges of the line graph has the size of $O(d^2_{\text{max}})$ where $d_{\text{max}}$ is the max degree of the original graph.
>
> Further, all existing works on edge representation learning, including Chen et al. (2019) simply use them to **augment the node representation learning for specific edge-related downstream tasks**. For instance, Chen et al. (2019) utilize the line graph to capture higher-order interactions among nodes for network community detection. Contrarily, our work focuses on **edge representation learning for general graph tasks**, including graph classification, graph reconstruction, graph generation, and node classification, which none of the existing methods tackle. Further, none of the existing methods can find **task-relevant edges** or **cluster similar edges into one** to obtain more meaningful edge representations, while ours achieve this goal with HyperDrop and HyperCluster.
>
> In the following table, we compare the transformation cost of our DHT (in seconds) with the line graph transformation, on graphs with 1,000 nodes with varying the number of edges, generated from the Erdos-Renyi graph. The results are average over 100 different runs.
>
> | $ \text{num of edges} \over \text{num of nodes} $ of the original graph | 2 | 4 | 8 | 16 | 32 | 64 |
> | --- | --- | --- | --- | --- | --- | --- |
> | Line graph | 32.78 | 65.93 | 131.36 | 260.92 | 527.44 | 1071.31 |
> | Dual Hypergraph (Ours) | **0.13** | **0.18** | **0.26** | **0.45** | **0.81** | **1.37** |
>
> We believe that this comparison against existing line-graph transformation for edge representation learning will be a valuable addition to better understand the novelty and the advantage of our work, and will include it to the paper in the revision. We thank you for your helpful suggestion.
>
> ---
> **Question 2:**  Does the transformation increase the complexity of message passing, especially for the scale-free networks (few hubs with very high degree)?
>
> **Answer:** The complexity of message passing on the dual hypergraph **does not depend on the type of networks** (e.g., scale-free networks), as long as the number of edges of the original graph is fixed, unlike the line graph. This is because the **complexity for message-passing is $O(m)$ for both the original graph and the dual hypergraph**, where $m$ is the number of edges. To be specific, since each node in the dual hypergraph has a degree of two (see Section A.3 of the supplementary file; line 62-64), the hyperedge list of the dual hypergraph has a dimensionality of $2m$, thus the message passing complexity on the dual hypergraph using the hyperedge list is $O(m)$ regardless of the type of a given graph. This complexity is the same for the original graph using the edge list.
>
> Although the analytic complexity of message passing is identical, we further empirically validate the computational complexity, by measuring the time cost for message passing (in seconds), on both random (Erdos-Renyi) and scale-free (Barabasi-Albert) networks with $3,000$ nodes and $11,984$ edges following the densification law (i.e., $m \propto n^{1.18}$, where $n$ and $m$ is the number of nodes and edges, respectively) (Leskovec et al., 2005) of the internet graph. As shown in the table below, the message passing time for nodes and edges are **almost identical**on two different types of graphs.
>
> | Graph | Time for node-level message passing (sec) | Time for edge-level message passing (sec) |
> | --- | --- | --- |
> | Erdos-Renyi | 0.0031 | 0.0034 |
> | Barabasi-Albert | 0.0029 | 0.0032 |
>
> ---
>
> **Question 3:**  The descriptions of the tasks in the experiments are not clear and may cause misunderstanding. For example, edge reconstruction is to reconstruct the feature of edge actually.
>
> **Answer:** Please note that each task is described in detail in the **Section C of the supplementary file**. We have clearly described that edge reconstruction is a task of reconstructing the features of the edges, in lines 189-190 in Section C.1 of the supplementary file. However we will also clarify this in the main paper as suggested.
>
>
> ---
>
> **References**
> * Chen et al. Supervised Community Detection with Line Graph Neural Networks. ICLR 2019.
> * Leskovec et al. Graphs over time: densification laws, shrinking diameters and possible explanations. KDD 2005.

---

> > ### Comment · Reviewer_xkvv · 2021-08-13
> > **Thanks for the Response**
> >
> > Thanks to the authors for the response. I appreciate the discussion, but I do not agree with the answer to question 2.
> >
> > Taking a star in graph theory as an example, the star is a graph and it has one hub node and $n$ trivial nodes connected to the hub. In my opinion, it requires $2n$ message passing to aggregate in one layer on the original graph. While it requires $n(n-1)$ message passing to aggregate on the dual hypergraph. The time complexity is increased obviously.

---

> > > ### Author Response · Authors · 2021-08-13
> > > **On the complexity of  message passing with a star graph**
> > >
> > > We sincerely appreciate your comments, and would be happy to address your concern below.
> > >
> > > ---
> > > **Question:** Taking a star in graph theory as an example, the star is a graph and it has one hub node and $n$ trivial nodes connected to the hub. In my opinion, it requires $2n$ message passing to aggregate in one layer on the original graph. While it requires $n(n-1)$ message passing to aggregate on the dual hypergraph. The time complexity is increased obviously.
> > >
> > > **Answer:** This is a misunderstanding. The complexity of the message passing for such a star graph, will definitely increase when you use a **line graph** to convert the edges to nodes, but not with our **dual hypergraph**. Please note that the complexity of the message passing is asymptotically the same for both the primal and the dual hypergraph, with our dual hypergraph transformation.
> > >
> > > Given a star graph $G$ with one hub and $n$ nodes connected to the hub, the message passing complexities of the line graph constructed out of $G$, and the dual hypergraph of $G$, are as follows:
> > >
> > > **The complexity of the message passing with a line graph**
> > > - A line graph of a star graph $G$ is a clique with $n$ nodes ($K_n$), which have $\frac{n(n-1)}{2}$ edges. Thus, as you correctly point out, the aggregation on this clique requires $n(n-1)$ message passing, which is clearly larger than the $2n$ message passing used on the original graph. Therefore, your argument is indeed correct in the case of line graphs. However, our dual hypergraph does not yield any increase in the computational complexity for the message passing, as we describe below.
> > >
> > > **The complexity of the message passing with a dual hypergraph**
> > > - A dual hypergraph of a star graph $G$ is a hypergraph with $n$ nodes connected by a **single hyperedge**, where each node has a single loop. Thus the total number of (hyper)edges is $n+1$, **which is exactly the same as the number of the nodes in the original graph**. Note that a hyperedge of a hypergraph could be associated with multiple nodes, and thus this results in the dual hypergraph having a hyperedge associated with all nodes of the star graph. We can use an incident matrix to represent such one to many association between an edge and the nodes (see Figure 2 for an example of such a hypergraph and the dual hypergraph transformation). Thus the aggregation on the dual hypergraph requires $2(n+1)$ message passing, which is asymptotically the same with the complexity of the message passing on the original graph, $2n$.
> > >
> > > In short, this seems like a **misunderstanding coming from confusing our dual hypergraph with a line graph**, in which a single node is split into multiple edges. As shown in our previous response for Question 2, the complexity of message passing is almost the same for the primal and the dual hypergraph, with our dual hypergraph transformation.
> > >
> > > To figure out the difference between the dual hypergraph and the line graph with an example, consider the **input graph in Figure 1** of our paper, which has four nodes and four edges. The dual hypergraph $G^{\ast}$ has a hyperedge $B$ connecting three nodes ({$1, 2, 3$}). However, the line graph cannot represent a set of edges as one edge, thus hyperedge $B$ of $G^{\ast}$ is inevitably splitted into three edges ({$1, 2$}, {$2, 3$}, and {$1,3$}). This subsequently increases the number of edges, compared to the dual hypergraph.

---

> > > > ### Comment · Reviewer_xkvv · 2021-08-13
> > > > **Thanks for the Response**
> > > >
> > > > Thanks for the response. I understand that the total number of (hyper)edges is $n+1$.  According to Eq. 4, aggregating of one node in the dual hypergraph requires $n$ message passing. And there are $n$ nodes in the dual hypergraph. Thus, the computational complexity of the aggregation becomes $O(n(n-1))$ in total.

---

> > > > > ### Author Response · Authors · 2021-08-13
> > > > > **The complexity of the message passing is indeed $O(m)$ with the sparse implementation.**
> > > > >
> > > > > We sincerely appreciate your timely response. We respond to your comment as below:
> > > > >
> > > > > ---
> > > > > **Question:** According to Eq. 4, aggregating of one node in the dual hypergraph requires $n$ message passing. And there are $n$ nodes in the dual hypergraph. Thus, the computational complexity of the aggregation becomes $O(n(n-1))$ in total.
> > > > >
> > > > >
> > > > > **Answer:** This is a misunderstanding. **The message passing on the dual hypergraph has exactly the same asymptotic complexity as the message passing on the primal(original) hypergraph, regardless of the type of the graph**. In case of the **sparse implementation**, the message passing complexity (i.e., the whole operational complexities to realize the Equation (1) and (4)) on the dual hypergraph is **$O(m)$** ($m$ is the number of edges in the original graph), which is identical to the complexity on the original graph. The complexity you explained is obtained from the **dense implementation**of the message passing on Equation (1) and (4); however, in this case, the complexity on the original graph should also be $O(n^2)$. We provide more detailed explanations of how we obtain these complexities below.
> > > > >
> > > > >
> > > > > **Complexity of Dense Implementation of Equation (4)**
> > > > >
> > > > > The cost of Equation (4) (message passing on the dual hypergraph) is consistent with the cost of Equation (1) (message passing on the original graph), since they have the same form. When it comes to considering the dense implementation, we first need to find the neighboring nodes for each node which requires $O(n)$ for each node (as described in line 199), and then perform aggregations with the obtained neighborhood indices for all nodes. **Therefore, given the star graph, densely implemented message passing on the dual hypergraph has the complexity of $O(n^2)$, which is exactly the same for the original graph $O(n^2)$**.
> > > > >
> > > > >
> > > > > **Complexity of our Sparse Implementation of Equation (4)**
> > > > >
> > > > > However, we can indeed avoid such inefficiency of the dense implementation using the **sparse matrices**(i.e., edge list, hyperedge list). As sparse matrices already contain the neighbor information in the form of two-dimensional lists, it eliminates the need to find the neighboring nodes, thus making the whole message passing more efficient. Formally, the entire message passing in the original graph using the sparse matrices has $O(m)$ complexity.
> > > > >
> > > > > Thus we **sparsely implemented the message passing on the dual hypergraph**, as clearly described in lines 186-189 and 200-201, and also in the answer to Question 2 of the first response (Please read our initial response to your review). Formally, by using the hyperedge list of the dual hypergraph, we can achieve **$O(m)$** complexity for message passing.
> > > > >
> > > > > To be specific, since each node in the dual hypergraph has a degree of two (see Section A.3 of the supplementary file; lines 62-64), the hyperedge list of the dual hypergraph has a dimensionality of $2m$, thus the message passing complexity on the dual hypergraph using the hyperedge list is $O(m)$. Therefore, using the sparse implementation, the complexity of message passing on the dual hypergraph is $O(m)$, not $O(n^2)$.
> > > > >
> > > > > To support the above analytic complexity of message passing for the original graph and the dual hypergraph, we have already measured their actual running-time on the real-world graphs in the previous response (Question 2), and we again bring the empirical results of the message passing time, to emphasize that the complexities for the original and the dual are the same.
> > > > >
> > > > > **Experimental validation**
> > > > >
> > > > > Although the analytic complexity of message passing is identical, we further empirically validate the computational complexity, by measuring the time cost for message passing (in seconds), on both random (Erdos-Renyi) and scale-free (Barabasi-Albert) networks with $3,000$ nodes and $11,984$ edges following the densification law (i.e., $m \propto n^{1.18}$, where $n$ and $m$ is the number of nodes and edges, respectively) (Leskovec et al., 2005) of the internet graph. As shown in the table below, the message passing time for the original graph and the dual hypergraph is **almost identical**on two different types of graphs.
> > > > >
> > > > > | Graph | Message passing on original graph (sec) | Message passing on dual hypergraph (sec) |
> > > > > | --- | --- | --- |
> > > > > | Erdos-Renyi | 0.0031 | 0.0034 |
> > > > > | Barabasi-Albert | 0.0029 | 0.0032 |
> > > > >
> > > > >
> > > > > ---
> > > > > **References**
> > > > > * Leskovec et al. Graphs over time: densification laws, shrinking diameters and possible explanations. KDD 2005.

---

> > > > > > ### Comment · Reviewer_xkvv · 2021-08-14
> > > > > > **Thanks for the response**
> > > > > >
> > > > > > I read the lines you mentioned, but there are few descriptions to help readers understand why the number of massage passing will decrease by using the sparse Implementation.
> > > > > >
> > > > > > Could you please provide a complexity analysis. How many messages passing are conducted for a node in the hypergraph using the sparse Implementation? And how many nodes are required to conduct aggregation in the hypergraph using the sparse Implementation?

---

> > > > > > > ### Author Response · Authors · 2021-08-14
> > > > > > > **Complexity analysis of the sparse hypergraph implementation**
> > > > > > >
> > > > > > > We sincerely thank you for your timely response. We provide the answers to your new question below:
> > > > > > >
> > > > > > > ---
> > > > > > > **Question:** Could you please provide a complexity analysis. How many messages passing are conducted for a node in the hypergraph using the sparse Implementation? And how many nodes are required to conduct aggregation in the hypergraph using the sparse Implementation?
> > > > > > >
> > > > > > > **Answer:** We want to clarify that the message passing for the sparse implementation focuses on the **edges**, rather than the nodes. In other words, the message is passed from $x_i$ to $x_j$ for each $(i,j)$ in the edge list, and not for every pair of $x_i$ and $x_j$ in the node set, and thus the complexity of the message passing depends on the number of edges.
> > > > > > >
> > > > > > > Specifically, as the (hyper)edge list **already contains the information of the neighboring nodes**, there is **no need to pass the messages for every pair of nodes**. The messages of the nodes are first **aggregated to the (hyper)edges**, and then **passed to each incident node**. Thus this sparse implementation costs $O(m)$ complexity, where $m$ is the number of (hyper)edges. We further explain the complexity of sparse implementation with specific examples below.
> > > > > > >
> > > > > > > **Example of sparse implementation of message passing on the graph and the hypergraph**
> > > > > > >
> > > > > > > - We first consider the message passing complexity of the original graph with sparse implementation. Suppose that we are given a clique with four nodes, where the nodes are A, B, C, and D. Then, the undirected edges of this clique are defined as following six edges [[A, B], [A, C], [A, D], [B, C], [B, D], [C, D]], which is indeed the two-dimensional edge list (we omit the reverse orders, such as B to A, for simplicity). The message passing denotes the information propagation from the start node to the end node, which first propagates the information from the start node to the incidence edge (for example, A to [A, B] for the propagation of A to B), and then further propagates the information of the intermediate edge to the end node (for example, [A, B] to B for the propagation of A to B). Therefore, **we do not have to consider the number of message passing conducted for each node**. In other words, **we only consider the number of message passing conducted for all edges** to count the number of the aggregation operations, from which we can approximate the complexity of message passing as $O(m)$.
> > > > > > >
> > > > > > > - The message passing complexity of the (dual) hypergraph with sparse implementation, is almost the same to as that of the original graph. We now suppose that we are given a hypergraph with four nodes 1, 2, 3, and 4 connected to one hyperedge A. Then, the hyperedge list is denoted as follows: [[1, A], [2, A], [3, A], [4, A]] which indicates the incidence between nodes and edges. Similar to the case of the original graph, the message passing on the hypergraph first propagates the information from start nodes to intermediate hyperedges (for example, {1, 2, 3, 4} to A), and then propagates the aggregated information on the intermediate hyperedges to the end nodes (for example, A to {1, 2, 3, 4}). Thus, the number of aggregation operations is linear to **the number of elements in the hyperedge list**.
> > > > > > >
> > > > > > > - To sum up, with our dual hypergraph case, as we repeatedly describe that the dual hypergraph of the original graph has $2m$ elements in the hyperedge list, the **message passing complexity of the dual hypergraph has exactly the same asymptotic complexity as the message passing on the original graph**, which is $O(m)$.
> > > > > > >
> > > > > > > - We will further clarify the complexity of message passing for the original graph and the hypergraph across dense and sparse implementations in the revision, and we thank you for the valuable discussion.
> > > > > > >
> > > > > > > ---
> > > > > > > We sincerely appreciate your comments as well as your responsiveness during the discussion period. We hope the above responses satisfactorily address your points, and please let us know if you have anything else we should address.
> > > > > > >
> > > > > > > Thanks, Authors.

---

> > > > > > > ### Author Response · Authors · 2021-08-17
> > > > > > > **A gentle reminder**
> > > > > > >
> > > > > > > Dear Reviewer xkvv,
> > > > > > >
> > > > > > > I know that you must be very busy, but could you please check our response below, which includes the complexity analysis of the sparse hypergraph you requested? We showed that the complexity of the message passing on the hypergraph is $O(m)$, where $m$ is the number of edges.
> > > > > > >
> > > > > > > Thanks,
> > > > > > > Authors

---

> > > > > > > > ### Comment · Reviewer_xkvv · 2021-08-30
> > > > > > > > **Thanks for the response**
> > > > > > > >
> > > > > > > > Get it. Thanks for the clarification.

---

> > > > > > > > > ### Author Response · Authors · 2021-08-31
> > > > > > > > > **Any other concerns?**
> > > > > > > > >
> > > > > > > > > Dear Reviewer xkvv,
> > > > > > > > >
> > > > > > > > > Do you have any other concerns remaining other than the complexity of the message passing? We would like to politely ask you to revise the review and the score, if you found our responses satisfactory. We strongly believe that the detailed explanation of the complexity of message passing we provided in the response will further strengthen our paper. We thank you again for your constructive comments.
> > > > > > > > >
> > > > > > > > > Thanks,
> > > > > > > > > Authors.

---

> > > > > > > ### Author Response · Authors · 2021-08-24
> > > > > > > **A gentle reminder**
> > > > > > >
> > > > > > > Dear Reviewer xkvv,
> > > > > > >
> > > > > > > Could you please check our response on the sparse hypergraph implementation, and tell us if you need more explanations? We hope that your concern on the increased computational complexity is cleared away with the explanations we provided, and you understand that the computational complexity of the message passing is the same for the primal (original) and the dual hypergraph, with our dual hypergraph transformation. We thank you again for your time and efforts in reviewing our paper.
> > > > > > >
> > > > > > > Thanks,
> > > > > > > Authors

---

> ### Author Response · Authors · 2021-09-02
> **The discussion period ends today**
>
> Dear Reviewer xkvv,
>
> We sincerely appreciate your positive comments that we study an interesting and useful problem to learn the representations of edges, the proposed idea and method are intuitive, and the experiment results are impressive. We have made every effort to faithfully address all your comments in the responses. Here, we briefly summarize the main points of our responses below:
>
> - We have clarified that the main novelty of our work comes from edge representation learning using Dual Hypergraph Transformation which is **novel, simple yet effective, and allows lossless compression of the node information**. Furthermore, we pointed out the **importance of edge representation learning for general graph-related tasks**, which has been largely overlooked.
> - We have explained that the **complexity of the message passing on the dual hypergraph is the same as the complexity of the message passing on the original graph** regardless of the type of the input graph. Moreover, we have provided the complexity analysis as well as the intuitive example of the message passing on the dual hypergraph.
>
> We sincerely appreciate your insightful and constructive comments, and thank you again for your time and efforts in reviewing our paper. Please let us know if you have any further questions.
>
> Best regards, Authors

---

> ### Author Response · Authors · 2021-09-03
> **Thank you**
>
> Dear Reviewer xkvv,
>
> We would like to thank you for actively engaging in the discussion with us, which allowed us to address all your comments. We believe that incorporating your constructive comments and our discussion on them into our paper will significantly strengthen it. We thank you again for your time and efforts in reviewing our paper.
>
> Best regards,
> Authors

---

### Official Review · Reviewer_JU4W · 2021-07-25

**Rating:** 7
**Confidence:** 4

**Summary:**

This paper is in the area of graph representation learning, and more specifically, they focus on edge representation learning. The proposed method is an edge representation learning framework that is based on Dual Hypergraph Transformation (DHT), that transforms the edges of the original graph into the nodes of a hypergraph and nodes to hyperedges. The dual hyper graph construction allows to apply any off-the-shelf message passing techniques designed for node representations to edge representation. Then, they proposed two novel graph pooling methods for the hypergraph to obtain compact graph-level edge representations; to cluster (HyperCluster) or drop (HyperDrop) edges to obtain holistic graph-level edge representations. They validate the edge representations on various synthetic and molecular datasets for graph representation and generation downstream tasks. When tested on the graph classification task, the proposed method outperforms the state-of-the-art comparison graph polling methods.

-- Edit after the discussion phase: I acknowledge that I have read the authors' response and the other reviews.

**Ethical Concerns:**

None.

**Limitations And Societal Impact:**

The authors discuss the limitations of this work in the paper, they also include a section for this purpose in the Appendix, and there are no concerns for potential negative societal impact.

**Main Review:**

The paper is well-written and well-structured. The authors work on an interesting problem of edge representation learning and more specifically the propose a framework to construct a hypergraph and apply off-the-shelf message-passing methods from node to edge representations. The related work is dense and covers most important related works. The methodology section (Section 3) contains all details for the proposed edge representation learning method, and justifications with references when needed. The experimental setup is nicely done, it has a good structure with all the necessary details and analyses (including results summarization in Tables and Figures). The authors include edge reconstruction results in the ZINC molecule and synthetic datasets, as well as graph reconstruction results in the ZINC molecule and some examples. They have added the graph compression results figures that shows the relative size of compressed graphs from the proposed method to the node pooling method (GMT). The results on the graph generation task against MolGAN and MARS show that the proposed method outperforms the others. The graph classification results are shown against various datasets and various pooling methods. Overall, the paper discusses some nice ideas regarding the edge representation learning and how to apply message passing techniques for node representation learning to edge representation learning. Also, the authors propose two edge pooling methods and they provide a convincing analysis of an in-depth experimentation they have done that shows the performance improvements.

**Time Spent Reviewing:**

8

---

> ### Author Response · Authors · 2021-08-10
> **Initial Response to Reviewer JU4W**
>
> We sincerely appreciate your thoughtful acknowledgments of the novelty of the problem as well as an idea to tackle it, the strength of the proposed method, the quality of the writing, and the extensiveness of the experiments.
>
> As you pointed out, our paper proposes an intuitive and effective idea to solve the interesting problem of edge representation learning by exploiting the hypergraph duality, to overcome the lack of sub-optimality of edge representation learning with GNNs in previous works on the topic. The proposed Edge HyperGraph Neural Network (EHGNN) and the two novel graph pooling methods (HyperCluster and HyperDrop) together serve the purpose of learning explicit task-relevant edge representations, which is a problem that has been relatively overlooked despite its importance in general graph-related tasks.
>
> Further, as mentioned, our proposed methods are not only powerful as verified by the extensive experiments on a variety of graph-related tasks, but also is surprisingly simple and is universal, as they eliminate the need to design special message passing or pooling methods specifically tailored for edges. We strongly agree with your review, and believe that our work will be a valuable contribution for graph representation learning with GNNs, which will not only have high impact due to its practical effectiveness, but also open a new door for the relatively overlooked problem of edge representation learning.

---

> ### Author Response · Authors · 2021-08-28
> **The end of the discussion phase is approaching**
>
> Dear Reviewer JU4W,
>
> We appreciate your positive comments about the novelty of the problem, the strength of our proposed methods, the quality of the writing, and the extensiveness of the experiments. We have done our best to respond to other reviewers’ comments in the discussion phase, which we believe have made our work more solid. Please let us know if you have any further questions.
>
> We thank you again for your time and efforts in reviewing our paper, as well as your constructive comments.
>
> Best regards, Authors

---

> ### Author Response · Authors · 2021-09-03
> **Thank you**
>
> Dear Reviewer JU4W,
>
> We sincerely appreciate your positive review of our work. During the long discussion period, we did our best to faithfully address all comments from the reviewers, and we believe that incorporating these discussions and new results will further strengthen our work. We thank you again for your time and efforts in reviewing our paper.
>
> Best regards,
> Authors

---

### Author Response · Authors · 2021-08-17
**Recap of our main novelty and contributions**

We sincerely appreciate your time and effort in reviewing our paper, as well as the constructive comments and valuable suggestions from all reviewers. To clarify the originality and positioning of our work more precisely, here we summarize the main contributions of our work in the following three aspects: **task-level, framework-level, and component-level**. For other points besides the main contributions, please refer to our responses to each question on the comments for each reviewer.

---
## Task-level novelty

- We first want to emphasize that our main contribution is indeed the focus on edge representation learning, which has been **largely overlooked**. However, we find it **crucial to the success of graph-level representation learning** in our work. In other words, we target the limitation of current GNNs that focus on nodes and using the edges only as auxiliary information for node representation learning, by pointing out the importance of explicit representation learning for the edges, and demonstrating this through experimental studies on four different tasks. We strongly believe that this task-level novelty should not be dismissed and further be highlighted, and thank you for thoughtfully acknowledging our problem-side novelty as “this paper studies an interesting and useful problem to learn the representation of edges”, “the authors work on an interesting problem of edge representation learning”, and “the idea of coupling a graph-wise representation based on node features with one explicitly aimed at representing edges is a new contribution”.

- To tackle this important but largely undiscovered problem, we found that the combination of edge representation learning and hypergraph duality is very **simple and versatile, yet effective and useful**, which we concretely explain in the next section.

---
## Framework-level novelty

- We now highlight the novelty of our proposed framework. The key idea of our work is the “edge representation learning with hypergraph duality”, and this is not a mere sum of its components, such as (i) transformation of edges to nodes with hypergraph duality, (ii) message passing on the hypergraph, and (iii) pooling edges as nodes. We restate our **framework-level novelty** more concretely below.

- **The novelty we claim is neither (i) nor (ii)**, but the **novel framework which puts them together for edge representation learning**. We are confident that no existing works have proposed such integration, although this seemingly straightforward framework achieves **remarkable performance** in a vast range of graph-level tasks. (iii) is at last made possible under the complete framework (i)+(ii) we suggested, and it is just one of many advantages our framework has. Most of these advantages are not the properties of the individual components, but can only emerge when they are **put together as a single framework**. Dissecting our edge learning framework into individual components, and saying each of them is not novel, is exactly the same as saying that the self-attention mechanism is not novel since 1) it utilizes an existing attention mechanism, 2) sequence learning has been done before, and 3) it is a straightforward idea to use the attention mechanism for sequence learning. However, this is not the novelty of the self-attention learning framework, but the idea of utilizing the attention mechanism for sequence representation learning is. Similarly, neither edge representation learning nor hypergraph message passing is our novelty, but edge representation learning on the dual hypergraph is, and we believe this is a sufficiently novel idea but has not been explored because existing works overlooked the importance of explicit edge representation learning. If this is such a trivial idea, why hasn't anybody proposed to use them, while they help achieve such remarkable performances?

Here we itemize the framework-level advantages of our combination below:

**Versatility** Our edge representation learning framework eliminates the need to design tailored message passing and even pooling schemes for edges, since the Dual Hypergraph Transformation (DHT) interchanges the structural role of nodes and edges without costs. The combination of these two components, **dual hypergraph transformation** and **edge-level message passing** for edge representation learning, should be considered as an indiscerptible framework.

**Utility** Our edge representation learning framework has numerous utilities on various graph-level tasks, and we show which architectural variation is appropriate for each task throughout our experiments, as follows:

- In the **graph reconstruction** task, we show that the graph-level edge representation obtained by clustering similar edges, is highly beneficial to compress edge representations compared to various edge-aware GNNs (Figure 3, 4, 5, 6, and 7). In the **graph generation** task, we show that accurately learning edge representations is useful to generate molecules with desired chemical properties than other edge-aware GNNs, since ours can precisely distinguish different molecules with accurate edge information (Figure 8 and Table 2).

- In the **graph classification** task, we present that the hierarchical edge drop scheme using the obtained edge representations from message-passing outperforms existing hierarchical node-based pooling methods (Table 3), by dropping uninformative edges while preserving the node information. We also present qualitative evidence of our effective edge drop with visual examples in Figure 9. Further, we verify that our method is orthogonal to global pooling methods, and using both methods together can significantly improve the performance.

- In the **node classification** task, we validate that identifying task-relevant edges with our method is helpful to alleviate the over-smoothing problem of GNNs (Figure 10), better than the random edge dropping scheme. Previous node-based pooling methods cannot be applied to such tasks due to the inevitable loss of node information.

The above advantages of our edge representation learning framework clearly and persuasively support that it can be extensively used for various graph-related tasks, where the importance of edge representations has been largely overlooked. To sum up, we believe that the significance of our work not only comes from our **effective and efficient framework** of learning edge representations, but also from our **extensive experiments** on substantial graph-level tasks, in the framework-level.

---
## Component-level novelty

We believe that the above explanations of the task- and the framework-level of our novelty sufficiently demonstrate the novelty of our work. However, even putting aside the task- and framework-level novelty for edge representation learning, our work has distinct points in the component-level comparison.

1) We have never claimed the novelty for “constructing dual hypergraphs” as we have clearly explained in our initial response, which we have also clearly acknowledged the hypergraph duality (Berge, 1973; Scheinerman & Ullman, 2011) and the relevant works in the Related Work.

2) The idea of applying the hypergraph duality to realize message passing between edges for learning explicit edge representations with GNNs, **has never been exploited before**, including other settings.

3) The key idea for our novel edge pooling methods is **obtaining graph-level edge representations** to solve the downstream tasks, which we approach by pooling nodes in the dual hypergraph. To the best of our knowledge, **no other work has considered using pooling in the dual hypergraph for edge representation learning**.

---
We hope the above responses make our contributions more clear. We thank you again for reviewing our work, and the new discussions and experimental results are highly valuable to us, which we will include in the subsequent revision.

Thanks, Authors.

---
**References**
* Berge. Graphs and hypergraphs. 1973.
* Scheinerman & Ullman. Fractional graph theory: a rational approach to the theory of graphs. Courier Corporation. 2011.

---

### Decision · Program_Chairs · 2021-09-28

**Decision:**

Accept (Poster)

**Comment:**

This paper proposes a new edge representation learning model through Dual Hypergraph Transformation.
Finding good representation of edges remains an under-studied problem and the results of this paper can be considered as progress towards the problem.
 The proposed model outperforms
baselines on several tasks such asl graph reconstruction, generation, and classification tasks.
However, the authors should incorporate the salient points of the discussion in the final submission.


**Consistency Experiment:**

NeurIPS has a long history of experimentation. In 2014, NeurIPS ran an experiment in which 10% of submissions were reviewed by two independent committees to quantify the randomness in the review process. This year, we repeated a variant of this experiment to see how the quality of the review process has changed over time.  This paper was part of the experiment and was therefore assigned to two committees (consisting of reviewers, an Area Chair, and a Senior Area Chair) that reached independent decisions.  If both committees made the same recommendation, this recommendation was followed. If a single committee recommended acceptance, the paper was accepted (with the exception of a few cases in which the other committee identified what we considered a fatal flaw, e.g., an error in a key result).

This copy’s committee reached the following decision: **Accept (Poster)**

The other committee assigned to the paper recommended **Reject**.  You can find the other set of reviews, along with any follow up discussion with the authors here:
https://openreview.net/forum?id=vwgsqRorzz